# The Global Methane Budget 2000-2017

Marielle Saunois[1], Ann R. Stavert[2], Ben Poulter[3], Philippe Bousquet[1], Josep G. Canadell[2], Robert B. Jackson[4], Peter A. Raymond[5], Edward J. Dlugokencky[6], Sander Houweling[7,8], Prabir K. Patra[9,10], Philippe Ciais[1], Vivek K. Arora[11], David Bastviken[12], Peter Bergamaschi[13], Donald R. Blake[14], Gordon Brailsford[15], Lori Bruhwiler[6], Kimberly M. Carlson[16,17], Mark Carrol[3], Simona Castaldi[18,19,20], Naveen Chandra[9], Cyril Crevoisier[21], Patrick M. Crill[22], Kristofer Covey[23], Charles L. Curry[24], Giuseppe Etiope[25,26], Christian Frankenberg[27,28], Nicola Gedney[29], Michaela I. Hegglin[30], Lena Höglund-Isaksson[31], Gustaf Hugelius[32], Misa Ishizawa[33], Akihiko Ito[33], Greet Janssens-Maenhout[13], Katherine M. Jensen[34], Fortunat Joos[35],Thomas Kleinen[36], Paul B. Krummel[37], Ray L. Langenfelds[37], Goulven G. Laruelle[38], Licheng Liu[39], Toshinobu Machida[33], Shamil Maksyutov[33], Kyle C. McDonald[34], Joe McNorton[40], Paul A. Miller[41], Joe R. Melton[42], Isamu Morino[33], Jurek Müller[35], Fabiola Murguia-Flores[43], Vaishali Naik[44], Yosuke Niwa[33,45], Sergio Noce[20], Simon O'Doherty[46], Robert J. Parker[47], Changhui Peng[48], Shushi Peng[49], Glen P. Peters[50], Catherine Prigent[51], Ronald Prinn[52], Michel Ramonet[1], Pierre Regnier[38], William J. Riley[53], Judith A. Rosentreter [54], Arjo Segers[55], Isobel J. Simpson[14], Hao Shi[56], Steven J. Smith[57,58], L. Paul Steele[37], Brett F. Thornton[22], Hanqin Tian[56], Yasunori Tohjima[33], Francesco N. Tubiello[59], Aki Tsuruta[60] Nicolas Viovy[1], Apostolos Voulgarakis[61,62], Thomas S. Weber[63], Michiel van Weele[64], Guido R. van der Werf[8], Ray F. Weiss[65], Doug Worthy[66], Debra Wunch[67], Yi Yin[1,27], Yukio Yoshida[33], Wenxin Zhang[41], Zhen Zhang[68], Yuanhong Zhao[1], Bo Zheng[1], Qing Zhu[53], Qiuan Zhu[69], and Qianlai Zhuang[39]

[1]Laboratoire des Sciences du Climat et de l'Environnement, LSCE-IPSL (CEA-CNRS-UVSQ), Université Paris-Saclay 91191 Gif-sur-Yvette, France

[2]Global Carbon Project, CSIRO Oceans and Atmosphere, Aspendale, VIC 3195, and Canberra, ACT 2601, Australia

[3]NASA Goddard Space Flight Center, Biospheric Science Laboratory, Greenbelt, MD 20771, USA

[4]Department of Earth System Science, Woods Institute for the Environment, and Precourt

Institute for Energy, Stanford University, Stanford, CA 94305-2210, USA

[5]School of Forestry and Environmental Studies, Yale University, New Haven, CT 06511, USA.

[6]NOAA ESRL, 325 Broadway, Boulder, CO 80305, USA

[7]SRON Netherlands Institute for Space Research, Sorbonnelaan 2, 3584 CA Utrecht, the Netherlands

[8]Vrije Universiteit Amsterdam, Department of Earth Sciences, Earth and Climate Cluster, VU Amsterdam, Amsterdam, the Netherlands

[9]Research Institute for Global Change, JAMSTEC, 3173-25 Showa-machi, Kanazawa, Yokohama, 236-0001, Japan

[10] Center for Environmental Remote Sensing, Chiba University, Chiba, Japan

[11]Canadian Centre for Climate Modelling and Analysis, Climate Research Division, Environment and Climate Change Canada, Victoria, BC, V8W 2Y2, Canada

[12]Department of Thematic Studies – Environmental Change, Linköping University, 581 83 Linköping, Sweden

[13]European Commission Joint Research Centre, Via E. Fermi 2749, 21027 Ispra (Va), Italy

[14]Department of Chemistry, University of California Irvine, 570 Rowland Hall, Irvine, CA 92697, USA

[15]National Institute of Water and Atmospheric Research, 301 Evans Bay Parade, Wellington, New Zealand

[16] Institute on the Environment, University of Minnesota, Saint Paul, Minnesota 55108, USA

[17] Department of Natural Resources and Environmental Management, University of Hawai'i, Honolulu, Hawai'i 96822, USA.

[18]Dipartimento di Scienze Ambientali, Biologiche e Farmaceutiche, Università degli Studi della Campania Luigi Vanvitelli, via Vivaldi 43, 81100 Caserta, Italy

[19]Department of Landscape Design and Sustainable Ecosystems, RUDN University, Moscow,

Russia

[20]Impacts on Agriculture, Forests, and Ecosystem Services Division, Centro Euro-Mediterraneo sui Cambiamenti Climatici, Via Augusto Imperatore 16, 73100 Lecce, Italy

[21]Laboratoire de Météorologie Dynamique, LMD-IPSL, Ecole Polytechnique, 91120 Palaiseau, France

[22]Department of Geological Sciences and Bolin Centre for Climate Research, Svante Arrhenius väg 8, 106 91 Stockholm, Sweden

[23]Program in Environmental Studies and Sciences, Skidmore College, Saratoga Springs, NY 12866, USA

[24]School of Earth and Ocean Sciences, University of Victoria, P.O. Box 1700 STN CSC, Victoria, BC, Canada V8W 2Y2

[25]Istituto Nazionale di Geofisica e Vulcanologia, Sezione Roma 2, via V. Murata 605 00143 Rome, Italy

[26] Faculty of Environmental Science and Engineering, Babes Bolyai University, Cluj-Napoca, Romania

[27] Division of Geological and Planetary Sciences, California Institute of Technology, Pasadena,

CA, United States

[28]Jet Propulsion Laboratory, California Institute of Technology, Pasadena, CA, United States

[29]Met Office Hadley Centre, Joint Centre for Hydrometeorological Research, Maclean Building, Wallingford OX10 8BB, UK

[30]Department of Meteorology, University of Reading, Earley Gate, Reading RG6 6BB, United

Kingdom

[31]Air Quality and Greenhouse Gases Program (AIR), International Institute for Applied Systems Analysis (IIASA), 2361 Laxenburg, Austria

[32]Department of Physical Geography and Bolin Centre for Climate Research, Stockholm University, 106 91 Stockholm, Sweden

[33]Center for Global Environmental Research, National Institute for Environmental Studies (NIES), Onogawa 16-2, Tsukuba, Ibaraki 305-8506, Japan

[34]Department of Earth and Atmospheric Sciences, City College of New York, City University of New York, New York, NY 10031, USA

[35]Climate and Environmental Physics, Physics Institute and Oeschger Centre for Climate Change
Research, University of Bern, Sidlerstr. 5, 3012 Bern, Switzerland

[36]Max Planck Institute for Meteorology, Bundesstraße 53, 20146 Hamburg, Germany

[37]Climate Science Centre, CSIRO Oceans and Atmosphere, Aspendale, Victoria 3195, Australia

[38]Department Geoscience, Environment & Society, Université Libre de Bruxelles, 1050-Brussels, Belgium

[39]Department of Earth, Atmospheric, Planetary Sciences, Department of Agronomy, Purdue University, West Lafayette, IN 47907, USA

[40]Research Department, European Centre for Medium-Range Weather Forecasts, Reading, UK

[41]Department of Physical Geography and Ecosystem Science, Lund University, Sölvegatan 12, 223 62, Lund, Sweden

[42]Climate Research Division, Environment and Climate Change Canada, Victoria, BC, V8W 2Y2, Canada

[43]School of Geographical Sciences, University of Bristol, Bristol, BS8 1SS, UK

[44]NOAA/Geophysical Fluid Dynamics Laboratory (GFDL), 201 Forrestal Rd., Princeton, NJ 08540, USA

[45]Meteorological Research Institute (MRI), Nagamine 1-1, Tsukuba, Ibaraki 305-0052, Japan

[46]School of Chemistry, University of Bristol, Cantock's Close, Clifton, Bristol BS8 1TS, UK

[47]National Centre for Earth Observation, University of Leicester, Leicester, LE1 7RH, UK

[48]Department of Biology Sciences, Institute of Environment Science, University of Quebec at Montreal, Montreal, QC H3C 3P8, Canada

[49]Sino-French Institute for Earth System Science, College of Urban and Environmental Sciences, Peking University, Beijing 100871, China.

[50]CICERO Center for International Climate Research, Pb. 1129 Blindern, 0318 Oslo, Norway

[51]CNRS, Sorbonne Université, Observatoire de Paris, Université PSL, Lerma, Paris, France

[52]Department of Earth, Atmospheric and Planetary Sciences, Massachusetts Institute of
Technology (MIT), Building 54-1312, Cambridge, MA 02139, USA

[53]Climate and Ecosystem Sciences Division, Lawrence Berkeley National Lab, 1 Cyclotron Road, Berkeley, CA 94720, US

[54]Centre for Coastal Biogeochemistry, School of Environment, Science and Engineering, Southern Cross University, Lismore, NSW 2480, Australia

[55]TNO, dep. of Climate Air & Sustainability, P.O. Box 80015, NL-3508-TA, Utrecht, The Netherlands

[56]International Center for Climate and Global Change Research, School of Forestry and Wildlife Sciences, Auburn University, 602 Duncan Drive, Auburn, AL 36849, USA

[57]Joint Global Change Research Institute, Pacific Northwest National Lab, College Park, MD, USA

[58]Department of Atmospheric and Oceanic Science, University of Maryland, College Park, MD, USA

[59]Statistics Division, Food and Agriculture Organization of the United Nations (FAO), Viale delle Terme di Caracalla, Rome 00153, Italy

[60]Finnish Meteorological Institute, P.O. Box 503, FI-00101, Helsinki, Finland

[61]Department of Physics, Imperial College London, London SW7 2AZ, UK

[62]School of Environmental Engineering, Technical University of Crete, Chania, Greece

[63]Department of Earth and Environmental Sciences, University of Rochester, Rochester, NY 14627, USA

[64]KNMI, P.O. Box 201, 3730 AE, De Bilt, the Netherlands

[65]Scripps Institution of Oceanography (SIO), University of California San Diego, La Jolla, CA 92093, USA

[66]Environnement Canada, 4905, rue Dufferin, Toronto, Canada

[67]Department of Physics, University of Toronto, 60 St. George Street, Toronto, Ontario, Canada

[68]Department of Geographical Sciences, University of Maryland, United States of America

[69]College of Hydrology and Water Resources, Hohai University, Nanjing, 210098, China

*Correspondence to*: Marielle Saunois (marielle.saunois@lsce.ipsl.fr)

**Abstract.** Understanding and quantifying the global methane ($CH_4$) budget is important for assessing realistic

pathways to mitigate climate change. Atmospheric emissions and concentrations of $CH_4$ continue to increase, making $CH_4$ the second most important human-influenced greenhouse gas in terms of climate forcing, after carbon dioxide ($CO_2$). The relative importance of $CH_4$ compared to $CO_2$ depends on its shorter atmospheric lifetime, stronger warming potential, and variations in atmospheric growth rate over the past decade, the causes of which are still debated. Two major difficulties in reducing uncertainties in the atmospheric growth

rate arise from the variety of geographically overlapping $CH_4$ sources and from the destruction of $CH_4$ by short-lived hydroxyl radicals (OH). To address these difficulties, we have established a consortium of multi-disciplinary scientists under the umbrella of the Global Carbon Project to synthesize and stimulate new research aimed at improving and regularly updating the global methane budget. Following Saunois et al. (2016), we present here the second version of the living review paper dedicated to the decadal methane

budget, integrating results of top-down studies (atmospheric observations within an atmospheric inverse-modelling framework) and bottom-up estimates (including process-based models for estimating land surface emissions and atmospheric chemistry, inventories of anthropogenic emissions, and data-driven extrapolations).

For the 2008-2017 decade, global methane emissions are estimated by atmospheric inversions (a top-down

approach) to be 576 Tg $CH_4$ $yr^{-1}$ (range 550-594, corresponding to the minimum and maximum estimates of

the model ensemble), of which 359 Tg $CH_4$ $yr^{-1}$ or ~60% are attributed to anthropogenic sources, that is emissions caused by direct human activity (range 336-376 Tg $CH_4$ $yr^{-1}$ or 50-65%). The updated total emission is 29 Tg $CH_4$ $yr^{-1}$ larger than our estimate for the period 2000-2009 and 24 Tg $CH_4$ $yr^{-1}$ larger than the one reported in the previous budget for 2003-2012 (Saunois et al., 2016). Since 2012, global $CH_4$ emissions have been tracking the warmest scenarios assessed by the Intergovernmental Panel on Climate Change. Bottom-up methods suggest almost 30% larger global emissions (737 Tg $CH_4$ $yr^{-1}$, range 594-881) than top-down inversion methods. Indeed, bottom-up estimates for natural sources such as natural wetlands, other inland water systems, and geological sources are higher than top-down estimates. The atmospheric constraints on the top-down budget suggest that at least some of these bottom-up emissions are overestimated. The latitudinal distribution of atmospheric observation-based emissions indicates a predominance of tropical emissions (~65% of the global budget, <30°N) compared to mid (~30%, 30°N-60°N) and high-northern latitudes (~4%, 60°N-90°N). The most important source of uncertainty in the methane budget is attributable to natural emissions, especially those from wetlands and other inland waters.

Some of our global source estimates are smaller than those in previously published budgets (Saunois et al. 2016; Kirschke et al. 2013), particularly for wetland emissions that are about 35 Tg $CH_4$ $yr^{-1}$ lower due to efforts to better partition wetlands and other inland waters. Emissions from geological sources are also found to be smaller by 7 Tg $CH_4$ $yr^{-1}$, and from wild animals by 8 Tg $CH_4$ $yr^{-1}$. However, the overall discrepancy between bottom-up and top-down estimates has been reduced by only 5% compared to Saunois et al. (2016), due to a higher estimate of non-vegetated wetland, i.e. inland waters, emissions resulting more detauiled research on emission factors. Priorities for improving the methane budget include: i) a global, high-resolution map of water-saturated soils and inundated areas emitting methane based on a robust classification of different types of emitting habitats; ii) further development of process-based models for inland-water emissions; iii) intensification of methane observations at local scales (e.g., FLUXNET-$CH_4$ measurements and urban-scale monitoring to constrain bottom-up land surface models, and at regional scales (surface networks and satellites) to constrain atmospheric inversions; iv) improvements of transport models and the representation of photochemical sinks in top-down inversions, and v) development of a 3D variational inversion system using isotopic and/or co-emitted species such as ethane to improve source partitioning.

The data presented here can be downloaded from https://doi.org/10.18160/GCP-CH4-2019 (Saunois et al., 2019) and from the Global Carbon Project.

**Table of content**

## 1 Introduction

The surface dry air mole fraction of atmospheric methane (CH$_4$) reached 1857 ppb in 2018 (Fig. 1), approximately 2.6 times greater than its estimated pre-industrial equilibrium value in 1750. This increase is attributable in large part to increased anthropogenic emissions arising primarily from agriculture (e.g., livestock production, rice cultivation, biomass burning), fossil fuel production and use, waste disposal, and alterations to natural methane fluxes due to increased atmospheric CO$_2$ concentrations and climate change

(Ciais et al., 2013). Atmospheric CH$_4$ is a stronger absorber of Earth's emitted thermal infrared radiation than carbon dioxide (CO$_2$), as assessed by its global warming potential (GWP) relative to CO$_2$. For a 100-yr time

horizon and without considering climate feedbacks GWP($CH_4$) = 28 (IPCC AR5, Myhre et al., 2013). Although global anthropogenic emissions of $CH_4$ are estimated at around 366 Tg $CH_4$ $yr^{-1}$ (Saunois et al., 2016), representing only 3% of the global $CO_2$ anthropogenic emissions in units of carbon mass flux, the

increase of atmospheric $CH_4$ concentrations has contributed ~23% (~0.62 W $m^{-2}$) to the additional radiative forcing accumulated in the lower atmosphere since 1750 (Etminan et al., 2016). Changes in other chemical compounds (such as nitrogen oxides ($NO_x$) or carbon monoxide (CO)) also influence the forcing of atmospheric $CH_4$ through changes to its atmospheric lifetime. From an emission perspective, the total radiative forcing attributable to anthropogenic $CH_4$ emissions is currently about 0.97 W $m^{-2}$ (Myhre et al.,

2013). Emissions of $CH_4$ contribute to the production of ozone, stratospheric water vapour, and $CO_2$, and most importantly affect its own lifetime (Myhre et al., 2013; Shindell et al., 2012). $CH_4$ has a short lifetime in the atmosphere (about 9 years for the year 2010 (Prather et al., 2012)) hence a stabilization or reduction of $CH_4$ emissions leads rapidly, in a few decades, to a stabilization or reduction of its atmospheric concentration and therefore its radiative forcing. Reducing $CH_4$ emissions is therefore recognized as an effective option for

rapid climate change mitigation, especially on decadal timescales (Shindell et al., 2012), because of its shorter lifetime than $CO_2$.

Of concern, the current anthropogenic methane emissions trajectory is estimated to lie between the two warmest IPCC-AR5 scenarios (Nisbet et al., 2016, 2019), i.e., the RCP8.5 and RCP6.0, corresponding to temperature increases above 3°C by the end of this century. This trajectory implies that large reductions of

methane emissions are needed to meet the 1.5-2°C target of the Paris Agreement (Collins et al., 2013; Nisbet et al., 2019). Moreover, $CH_4$ is a precursor of important air pollutants such as ozone, and, as such, its emissions are covered by two international conventions: the United Nations Framework Convention on Climate Change (UNFCCC) and the Convention on Long Range Transport of Air Pollution (CLRTAP), another motivation to reduce its emissions.

Changes in the magnitude and temporal variation (annual to inter-annual) of methane sources and sinks over the past decades are characterized by large uncertainties (Kirschke et al., 2013; Saunois et al., 2017; Turner et al., 2019). Also, the decadal budget suggests relative uncertainties (hereafter reported as min-max ranges) of 20-35% for inventories of anthropogenic emissions in specific sectors (e.g., agriculture, waste, fossil fuels), 50% for biomass burning and natural wetland emissions, and reaching 100% or more for other natural

sources (e.g. inland waters, geological sources). The uncertainty in the chemical loss of methane by OH, the predominant sink of atmospheric methane, is estimated around 10% (Prather et al., 2012) to 15% (from bottom-up approaches in Saunois et al. (2016)). This represents, for the top-down methods, the minimum relative uncertainty associated with global methane emissions, as other methane sinks (atomic oxygen and chlorine oxidations, soil uptake) are much smaller and the atmospheric growth rate is well-defined

(Dlugokencky et al., 2009). Globally, the contribution of natural $CH_4$ emissions to total emissions can be

quantified by combining lifetime estimates with reconstructed pre-industrial atmospheric methane concentrations from ice cores (e.g. Ehhalt et al., 2001). Regionally, uncertainties in emissions may reach 40-60% (e.g. for South America, Africa, China and India, see Saunois et al. (2016)).

In order to verify future emission reductions, for example to help conduct Paris Agreement's stocktake, sustained and long-term monitoring of the methane cycle is needed to reach more precise estimation of trends, and reduced uncertainties in anthropogenic emissions (Bergamaschi et al., 2018a; Pacala, 2010) . Reducing uncertainties in individual methane sources and thus in the overall methane budget is challenging for at least four reasons. Firstly, methane is emitted by a variety of processes, including both natural and anthropogenic sources, point and diffuse sources, and sources associated with three different emission classes (i.e., biogenic, thermogenic and pyrogenic). These multiple sources and processes require the integration of data from diverse scientific communities. The fact that anthropogenic emissions result from unintentional leakage from fossil fuel production or agriculture further complicates production of accurate bottom-up emission estimates. Secondly, atmospheric methane is removed by chemical reactions in the atmosphere involving radicals (mainly OH) that have very short lifetimes (typically ~1s). The spatial and temporal distributions of OH are highly variable. Although OH can be measured locally, calculating global $CH_4$ loss through OH measurements would require high resolution OH measurements (typically half an hour to integrate cloud cover, and 1 km spatially to consider OH high reactivity and heterogeneity). As a result, such a calculation is currently possible only through modelling. However, simulated OH concentrations from chemistry climate models still show uncertain spatio-temporal distribution at regional to global scales (Zhao et al., 2019). Thirdly, only the net methane budget (sources minus sinks) is constrained by precise observations of atmospheric growth rates (Dlugokencky et al., 2009), leaving the sum of sources and the sum of sinks more uncertain. One simplification for $CH_4$ compared to $CO_2$ is that the oceanic contribution to the global methane budget is small (~1-3%), making source estimation predominantly a continental problem (USEPA, 2010b). Finally, we lack observations to constrain 1) process models that produce estimates of wetland extent (Kleinen et al., 2012; Stocker et al., 2014) and wetland emissions (Melton et al., 2013; Poulter et al., 2017; Wania et al., 2013) 2) other inland water sources (Bastviken et al., 2011; Wik et al., 2016a) 3) inventories of anthropogenic emissions (Höglund-Isaksson, 2012, 2017; Janssens-Maenhout et al., 2019; USEPA, 2012), and 4) atmospheric inversions, which aim to estimate methane emissions from global to regional scales (Bergamaschi et al., 2013, 2018b; Bohn et al., 2015; Houweling et al., 2014; Kirschke et al., 2013; Saunois et al., 2016; Spahni et al., 2011; Thompson et al., 2017; Tian et al., 2016).

The global methane budget inferred from atmospheric observations by atmospheric inversions relies on regional constraints from atmospheric sampling networks, which are relatively dense for northern mid-latitudes, with a number of high-precision and high-accuracy surface stations, but are sparser at tropical latitudes and in the Southern Hemisphere (Dlugokencky et al., 2011) . Recently the atmospheric observation

density has increased in the tropics due to satellite-based platforms that provide column-average methane mixing ratios. Despite continuous improvements in the precision and accuracy of space-based measurements (e.g. Buchwitz et al., 2016), systematic errors greater than several ppb on total column observations can still limit the usage of such data to constrain surface emissions (Alexe et al., 2015; Bousquet et al., 2018; Chevallier et al., 2017; Locatelli et al., 2015). The development of robust bias corrections on existing data

can help overcome this issue (e.g. Inoue et al., 2016) and satellite-based inversions have been suggested to reduce global and regional flux uncertainties compared to surface-based inversions (e.g. Fraser et al., 2013). The Global Carbon Project (GCP) seeks to develop a complete picture of the carbon cycle by establishing common, consistent scientific knowledge to support policy debate and actions to mitigate greenhouse gas emissions to the atmosphere (www.globalcarbonproject.org). The objective of this paper is to analyse and

synthesize the current knowledge of the global methane budget, by gathering results of observations and models in order to better understand and quantify the main robust features of this budget, its remaining uncertainties, and to make recommendations. We combine results from a large ensemble of bottom-up approaches (e.g., process-based models for natural wetlands, data-driven approaches for other natural sources, inventories of anthropogenic emissions and biomass burning, and atmospheric chemistry models),

and top-down approaches (including methane atmospheric observing networks, atmospheric inversions inferring emissions and sinks from the assimilation of atmospheric observations into models of atmospheric transport and chemistry). The focus of this work is on decadal budgets and on the update of the previous assessment made for the period 2003-2012 to the more recent 2008-2017 decade. More in-depth analysis of trends and year-to-year changes are left to future publications. The regional budget is further discussed in

Stavert et al. (2020). Our current paper is a living review, published at about three-year intervals, to provide an update and new synthesis of available observational, statistical and model data for the overall $CH_4$ budget and its individual components.

Kirschke et al. (2013) was the first $CH_4$ budget synthesis and was followed by Saunois et al. (2016). Kirschke et al. (2013) reported decadal mean $CH_4$ emissions and sinks from 1980 to 2009 based on bottom-up and top-

down approaches. Saunois et al. (2016) reported methane emissions for three time periods: 1) the last calendar decade (2000-2009), 2) the last available decade (2003-2012), and 3) the last available year (2012) at the time. Here, we update reporting methane emissions and sinks for 2000-2009 decade, for the most recent 2008-2017 decade where data are available, and for the year 2017, reducing the time lag between the last reported year and analysis. The methane budget is presented here at global and latitudinal scales and data can

be downloaded from https://doi.org/10.18160/GCP-CH4-2019 (Saunois et al., 2019).

Five sections follow this introduction. Section 2 presents the methodology used in the budget: units, definitions of source categories, of regions, data analysis; and discusses the delay between the period of study of the budget and the release date. Section 3 presents the current knowledge about methane sources and sinks

based on the ensemble of bottom-up approaches reported here (models, inventories, data-driven approaches). Section 4 reports atmospheric observations and top-down atmospheric inversions gathered for this paper. Section 5, based on Sections 3 and 4, provides the updated analysis of the global methane budget by comparing bottom-up and top-down estimates and highlighting differences. Finally, Section 6 discusses future developments, missing components, and the most critical remaining uncertainties based on our update to the global methane budget.

## 2 Methodology

### 2.1 Units used

Unless specified, fluxes are expressed in teragrams of $CH_4$ per year (1Tg $CH_4$ yr$^{-1}$=$10^{12}$ g$CH_4$ yr$^{-1}$), while atmospheric concentrations are expressed as dry air mole fractions, in parts per billion (ppb), with atmospheric methane annual increases, $G_{ATM}$, expressed in ppb yr$^{-1}$. In the tables, we present mean values and ranges for the two decades 2000-2009, and 2008-2017, together with results for the most recent available year (2017). Results obtained from previous syntheses (i.e. Saunois et al., 2016) are also given for the decade 2000-2009. Following Saunois et al. (2016) and considering that the number of studies is often relatively small for many individual source and sink estimates, uncertainties are reported as minimum and maximum values of the available studies, in brackets. In doing so, we acknowledge that we do not consider the uncertainty of the individual estimates, and we express uncertainty as the range of available mean estimates, i.e., differences across measurements/methodologies considered. These minimum and maximum values are those presented in Section 2.5 and exclude identified outliers.

The $CH_4$ emission estimates are provided with up to three digits, for consistency across all budget flux components and to ensure the accuracy of aggregated fluxes. Nonetheless, given the values of the uncertainties in the methane budget, we encourage the reader to consider not more than two digits as significant.

### 2.2 Period of the budget and availability of data

The bottom-up estimates rely on global anthropogenic inventories, surface land models for wetlands emissions and published literature for other natural sources. The global gridded anthropogenic inventories are updated irregularly, generally every 3 to 5 years. The last reported years of available inventories were 2012, 2014 or 2016 when we started this study. For this budget, in order to cover the reported period (2000-2017), it was necessary to extrapolate some of these datasets as explained in Sect. 3.1.1. The surface land models were run over the full period 2000-2017 using dynamical wetland areas (Sect. 3.2.1).

For the top-down estimates, we use atmospheric inversions covering 2000-2017. The simulations run until mid-2018, but the last year of reported inversion results is 2017, which represents a two and a half year lag with the present, a two-year shorter lag than for the last release (Saunois et al., 2016). Satellite observations are linked to operational data chains and are generally available days to weeks after the recording of the spectra. Surface observations can lag from months to years because of the time for flask analyses and data checks in (mostly) non-operational chains. The final six months of inversions are generally ignored (spin down) because the estimated fluxes are not constrained by as many observations as the previous periods.

## 2.3 Definition of regions

Geographically, emissions are reported globally and for three latitudinal bands (90°S-30°N, 30-60°N, 60-90°N, only for gridded products). When extrapolating emission estimates forward in time (see Sect. 3.1.1), and for the regional budget presented by (Stavert et al., 2020), a set of 19 regions (oceans and 18 continental regions, see supplementary Fig. S1) were used. As anthropogenic emissions are often reported by country, we define these regions based on a country list (Table S1). This approach was compatible with all top-down and bottom-up approaches considered. The number of regions was chosen to be close to the widely used TransCom inter-comparison map (Gurney et al., 2004) but with subdivisions to separate the contribution from important countries or regions for the methane cycle (China, South Asia, Tropical America, Tropical Africa, USA, and Russia). The resulting region definition is the same as used for the GCP $N_2O$ budget (Tian et al., 2019).

## 2.4 Definition of source categories

Methane is emitted by different processes (i.e., biogenic, thermogenic, or pyrogenic) and can be of anthropogenic or natural origin. Biogenic methane is the final product of the decomposition of organic matter by methanogenic *Archaea* in anaerobic environments, such as water-saturated soils, swamps, rice paddies, marine sediments, landfills, sewage and wastewater treatment facilities, or inside animal digestive systems. Thermogenic methane is formed on geological time scales by the breakdown of buried organic matter due to heat and pressure deep in the Earth's crust. Thermogenic methane reaches the atmosphere through marine and land geological gas seeps. These methane emissions are increased by human activities, for instance the exploitation and distribution of fossil fuels. Pyrogenic methane is produced by the incomplete combustion of biomass and other organic material. Peat fires, biomass burning in deforested or degraded areas, wildfires and biofuel burning are the largest sources of pyrogenic methane. Methane hydrates, ice-like cages of trapped methane found in continental shelves and slopes and below sub-sea and land permafrost, can be of either biogenic or thermogenic origin. Each of these three process categories has both anthropogenic and natural components.

In the following, we present the different methane sources depending on their anthropogenic or natural origin, which is relevant for climate policy. Here, "natural sources" refer to pre-agricultural emissions even if they are perturbed by anthropogenic climate change, and "anthropogenic sources" are caused by direct human activities since pre-industrial/pre-agricultural time (3000-2000 BC, Nakazawa et al., 1993) including

agriculture, waste management and fossil fuel related activities. Natural emissions are split between "wetland" and "other natural" emissions (e.g., non-wetland inland waters, wild animals, termites, land geological sources, oceanic geological and biogenic sources, and terrestrial permafrost). Anthropogenic emissions contain: "agriculture and waste emissions", "fossil fuel emissions", "biomass and biofuel burning emissions", assuming that all types of fires cause anthropogenic sources, although they are partly of natural

origin (Fig. 6, see also Table 3 and 6).

Our definition of natural/anthropogenic sources does not correspond exactly to the definition used by UNFCCC following the IPCC guidelines (IPCC, 2006), where, for pragmatic reasons, all emissions from managed land are reported as anthropogenic, which is not the case here. For instance, we consider all wetlands as natural emissions, despite some wetlands being managed and their emissions being partly

reported in UNFCCC national communications. The human induced perturbation of climate, atmospheric $CO_2$, and nitrogen and sulfur deposition may cause changes in the sources we classified as natural. Following our definition, emissions from wetlands, inland water or thawing permafrost will be accountable in "natural" emissions, even though, we acknowledge that climate change – a human perturbation – may cause increasing emissions from these sources. Methane emissions from reservoirs are considered as natural even though

reservoirs are human-made, and since the 2019 refinement to the IPCC guidelines (IPCC, 2006, 2019) emissions from reservoirs and other flooded lands are considered anthropogenic by UNFCCC.

Following Saunois et al. (2016), we report anthropogenic and natural methane emissions for five main source categories for both bottom-up and top-down approaches.

Bottom-up estimates of methane emissions for some processes are derived from process-oriented models

(e.g., biogeochemical models for wetlands, models for termites), inventory models (agriculture and waste emissions, fossil fuel emissions, biomass and biofuel burning emissions), satellite-based models (large scale biomass burning), or observation-based upscaling models for other sources (e.g., inland water, geological sources). From these bottom-up approaches, it is possible to provide estimates for more detailed source sub-categories inside each main GCP category (see budget in Table 3). However, the total methane emission

derived from the sum of independent bottom-up estimates remains unconstrained.

For atmospheric inversions (top-down approach) the situation is different. Atmospheric observations provide a constraint on the global total source and a reasonable constraint on the global sink derived from methyl chloroform (Montzka et al., 2011; Rigby et al., 2017). The inversions reported in this work solve either for a total methane flux (e.g. Pison et al., 2013), or for a limited number of source categories (e.g. Bergamaschi et

al., 2013). In most of the inverse systems the atmospheric oxidant concentrations are prescribed with pre-optimized or scaled OH fields, and thus the atmospheric sink is not solved. The assimilation of $CH_4$ observations alone, as reported in this synthesis, can help to separate sources with different locations or temporal variations but cannot fully separate individual sources as they often overlap in space and time in some regions. Top-down global and regional methane emissions per source category were obtained directly from gridded optimized fluxes, wherever an inversion had solved for the separate five main GCP categories. Alternatively, if an inversion only solved for total emissions (or for categories other than the main five described above), then the prior contribution of each source category at the spatial resolution of the inversion was scaled by the ratio of the total (or embedding category) optimized flux divided by the total (or embedding category) prior flux (Kirschke et al., 2013). In other words, the prior relative mix of sources at model resolution is kept while updating total emissions with atmospheric observations. The soil uptake was provided separately in order to report total gross surface emissions instead of net fluxes (sources minus soil uptake).

In summary, bottom-up models and inventories are presented for all source processes and for the five main categories defined above globally. Top-down inversions are reported globally and only for the five main emission categories.

## 2.5 Processing of emission maps and box-plot representation of emission budgets

Common data analysis procedures have been applied to the different bottom-up models, inventories and atmospheric inversions whenever gridded products exist. Gridded emissions from atmospheric inversions, land-surface models for wetland or biomass burning were provided at the monthly scale. Emissions from anthropogenic inventories are usually available as yearly estimates. These monthly or yearly fluxes were provided on a 1°x1° grid or re-gridded to 1°x1°, then converted into units of Tg $CH_4$ per grid cell. Inversions with a resolution coarser than 1° were downscaled to 1° by each modeling group. Land fluxes in coastal pixels were reallocated to the neighbouring land pixel according to our 1° land-sea mask, and vice-versa for ocean fluxes. Annual and decadal means used for this study were computed from the monthly or yearly gridded 1°x1° maps.

Budgets are presented as boxplots with quartiles (25%, median, 75%), outliers, and minimum and maximum values without outliers. Outliers were determined as values below the first quartile minus three times the inter-quartile range, or values above the third quartile plus three times the inter-quartile range. Mean values reported in the tables are represented as "+" symbols in the corresponding figures.

 **3 Methane sources and sinks: bottom-up estimates**

For each source category, a short description of the relevant processes, original data sets (measurements, models) and related methodology are given. More detailed information can be found in original publication references and in the supplementary material of this study.

**3.1 Anthropogenic sources**

 **3.1.1 Global inventories gathered**

The main bottom-up global inventory datasets covering anthropogenic emissions from all sectors (Table 1) are from the United States Environmental Protection Agency (USEPA, 2012), the Greenhouse gas and Air pollutant Interactions and Synergies (GAINS) model developed by the International Institute for Applied Systems Analysis (IIASA) (Gomez Sanabria et al., 2018; Höglund-Isaksson, 2012, 2017) and the Emissions

 Database for Global Atmospheric Research (EDGARv3.2.2, Janssens-Maenhout et al., 2019) compiled by the European Commission Joint Research Centre (EC-JRC) and Netherland's Environmental Assessment Agency (PBL). We also used the Community Emissions Data System for historical emissions (CEDS) (Hoesly et al., 2018) developed for climate modelling and the Food and Agriculture Organization (FAO) dataset emission database (Tubiello, 2019), which only covers emissions from agriculture and land use

 (including peatland and biomass fires).

These inventory datasets report emissions from fossil fuel production, transmission and distribution; livestock enteric fermentation; manure management and application; rice cultivation; solid waste and wastewater. Since the level of detail provided by country and by sector varies among inventories, the data were reconciled into common categories according to Table S2. For example, agricultural and waste burning

 emissions treated as a separate category in EDGAR, GAINS and FAO, are included in the biofuel sector in the USEPA inventory and in the agricultural sector in CEDS. The GAINS, EDGAR and FAO estimates of agricultural waste burning were excluded from this analysis (these amounted to 1-3 Tg $CH_4$ $yr^{-1}$ in recent decades to prevent any inadvertent overlap with separate estimates of biomass burning emissions (e.g. GFEDv4.1s). In the inventories used here, emissions for a given region/country and a given sector are usually

 calculated following IPCC methodology (IPCC, 2006), as the product of an activity factor and an emission factor for this activity. An abatement coefficient is used additionally, to account for any regulations implemented to control emissions (see e.g. Höglund-Isaksson et al., 2015). These datasets differ in their assumptions and data used for the calculation; however, they are not completely independent because they follow the same IPCC guidelines (IPCC, 2006), and, at least for agriculture, use the same FAOSTAT activity

 data. While the USEPA inventory adopts emissions reported by the countries to the UNFCCC, other inventories (FAOSTAT, EDGAR and the GAINS model) produce their own estimates using a consistent

approach for all countries. These other inventories compile country-specific activity data and emission factor information or, if not available, adopt IPCC default factors (Höglund-Isaksson, 2012; Janssens-Maenhout et al., 2019; Tubiello, 2019). CEDS takes a different approach starting from pre-existing default emission estimates; for methane, a combination of EDGAR and FAO estimates is used, scaled to match other individual or region-specific inventory values when available. This process maintains the spatial information in the default emission inventories while preserving consistency with country level data. The FAOSTATdataset (hereafter FAO-CH$_4$) was used to provide estimates of methane emissions at country level but is limited to agriculture (enteric fermentation, manure management, rice cultivation, energy usage, burning of crop residues and prescribed burning of savannahs) and land-use (biomass burning). FAO-CH$_4$ uses activity data mainly from the FAOSTAT crop and livestock production database, as reported by countries to FAO (Tubiello et al., 2013), and applies mostly the Tier 1 IPCC methodology for emissions factors (IPCC, 2006), which depend on geographic location and development status of the country. For manure, the necessary country-scale temperature was obtained from the FAO global agro-ecological zone database (GAEZv3. 0, 2012). Although country emissions are reported annually to the UNFCCC by annex I countries, and episodically by non-annex I countries, data gaps of those national inventories do not allow the inclusion of these estimates in this analysis.

In this budget, we use the following versions of these databases (see Table 1):

- EDGARv4.3.2 which provides yearly gridded emissions by sectors from 1970 to 2012 (Janssens-Maenhout et al., 2019),
- GAINS model scenario ECLIPSE v6 (Gomez Sanabria et al., 2018; Höglund-Isaksson, 2012, 2017) which provides both annual sectoral totals by country from 1990 to 2015 and a projection for 2020 (that assumes current emission legislation for the future) and an annual sectorial gridded product from 1990 to 2015,
- USEPA (USEPA, 2012), which provides 5-year sectorial totals by country from 1990 to 2020 (estimates from 2005 onward are a projection), with no gridded distribution available,
- CEDS version 2017-05-18 which provides both gridded monthly and annual country-based emissions by sectors from 1970 to 2014 (Hoesly et al., 2018),
- FAO-CH$_4$ (database accessed in February 2019, (FAO, 2019)) containing annual country level data for the period 1961-2016, for rice, manure, and enteric fermentation; and 1990-2016 for burning savannah, crop residue and non-agricultural biomass burning.

In order to report emissions for the period 2000-2017, we extended and interpolated some of the datasets as explained in Sect. 2.2. The USEPA dataset was linearly interpolated to provide yearly values. The FAO-CH$_4$ dataset, ending in 2016, was extrapolated to 2017 using a linear fit based on 2014-2016 data. The EDGARv4.3.2 was extrapolated to 2017 using the extended FAO-CH$_4$ emissions for enteric fermentation,

manure management and rice cultivation, and using BP statistical review of fossil fuel production and consumption (BP Statistical Review of World Energy, 2019) for emissions from coal, oil and gas sectors. In this extrapolated inventory, called EDGARv4.3.2$_{EXT}$, methane emissions for year $t$ are set equal to the 2012 (last year) of EDGAR emissions ($E_{EDGARv4.3.2}$) times the ratio between FAO-CH4 emissions (or BP statistics) of year $t$ ($E_{FAO-CH4}(t)$) and FAO-CH4 emissions (or BP statistics) of 2012 ($E_{FAO-CH4}(2012)$). For each emission sector, region-specific emissions of EDGARv4.3.2$_{EXT}$ in year $t$ are estimated following Eq. (1):

$$E_{EDGARv4.3.2ext}(t) = E_{EDGARv4.3.2}(2012) \times E_{FAO-CH4}(t)/E_{FAO-CH4}(2012) \tag{1}$$

Transport, industrial, waste and biofuel sources were linearly extrapolated in EDGARv4.3.2$_{EXT}$ based on the last three years of data while other sources were kept constant at the 2012 level. To allow comparisons through 2017, the CEDS dataset has also been extrapolated in an identical method creating CEDS$_{EXT}$. However, in contrast to EDGARv4.3.2 dataset, CEDS dataset provides only a combined oil and gas sector; hence, we extended this sector using the sum of BP oil and gas emissions. The by-country GAINS dataset was linearly projected by sector for each country using the trend between the historical 2015 and projected 2020 values. These by-country projections were aggregated to the 19 global regions (Section 2.3 and Fig. S1) and used to extrapolate the GAINS gridded dataset in a similar manner to that described in Equation 1. Although we only use the extended inventories, in the following the "EXT" suffix will be dropped for clarity.

### 3.1.2 Total anthropogenic emissions

In order to avoid double counting and ensure consistency with each inventory, the range (min-max) and mean values of the total anthropogenic emissions were not calculated as the sum of the mean and range of the three anthropogenic categories ("Agriculture and waste", "Fossil fuels" and "Biomass burning & biofuels"). Instead, we calculated separately the total anthropogenic emissions for each inventory by adding its values for "Agriculture and waste", "Fossil fuels" and "biofuels" with the range of available large-scale biomass burning emissions. This approach was used for the EGDARv4.3.2, CEDS and GAINS inventories, but we kept the USEPA inventory as originally reported because it includes its own estimates of biomass burning emissions. FAO-CH4 was only included in the range reported for the "Agriculture and waste" category. For the latter, we calculated the range and mean value as the sum of the mean and range of the three anthropogenic subcategory estimates "Enteric fermentation and Manure", "Rice", and "Landfills and Waste". The values reported for the upper-level anthropogenic categories ("Agriculture and waste", "Fossil fuels" and "Biomass burning & biofuels") are therefore consistent with the sum of their subcategories, although there might be small percentage differences between the reported total anthropogenic emissions and the sum of the three upper-level categories. This approach provides a more accurate representation of the range of emission estimates, avoiding an artificial expansion of the uncertainty attributable to subtle differences in the definition of sub-sector categorisations between inventories.

Based on the ensemble of databases detailed above, total anthropogenic emissions were 366 [349-393] Tg CH$_4$ yr$^{-1}$ for the decade 2008-2017 (Table 3, including biomass and biofuel burning) and 334 [321-358] Tg CH$_4$ yr$^{-1}$ for the decade 2000-2009. Our estimate for the preceding decade is statistically consistent with Saunois et al. (2016) (338 Tg CH4 yr-1 [329-342]) and Kirschke et al. (2013) (331 Tg CH$_4$ yr$^{-1}$ [304-368]) for the same period. The slightly larger range reported herein with respect to previous estimates is mainly due to a larger range in the biomass burning estimates, as more biomass burning products are included in this update. The range associated with our estimates (~10-12%) is smaller than the range reported in Höglund-Isaksson et al. (2015) (~20%), perhaps because they analysed data from a wider range of inventories and projections, plus this study was referenced to one year only (2005) rather than averaged over a decade, as done here.

Figure 2 (top panels) summarizes global methane emissions of anthropogenic sources (including biomass and biofuel burning) by different datasets between 2000 and 2050. The datasets consistently estimate total anthropogenic emissions of ~300 Tg CH$_4$ yr$^{-1}$ in 2000. The main discrepancy between the inventories is their trend after 2005, with the lowest emissions projected by GAINS and the largest by CEDS. For the Sixth Assessment report of the IPCC, seven main Shared Socioeconomic Pathways (SSPs) were defined for future climate projections in the Coupled Model Intercomparison Project 6 (CMIP6) (Gidden et al., 2019; O'Neill et al., 2016) ranging from 1.9 to 8.5 W m$^{-2}$ radiative forcing by the year 2100 (as shown by the number in the SSP names). The trends in methane emissions from 2010 estimated by current inventories track the pathways with the highest radiative forcing in 2100 (based on the unharmonized scenarios developed by Integrated Assessment Models, top left panel). For the 1970-2015 period, historical emissions used in CMIP6 (Feng et al., 2019) combine anthropogenic emissions from CEDS (Hoesly et al., 2018) and a climatological value from the GFEDv4.1s biomass burning inventory (van Marle et al., 2017). The CEDS anthropogenic emissions estimates, based on EDGARv4.2, are 10-20 Tg higher than the more recent EDGARv4.3.2 (van Marle et al., 2017). Harmonized scenarios used for CMIP6 activities start in 2015 at 388 Tg CH$_4$ yr$^{-1}$. Since methane emissions continue to track scenarios that assume no or minimal climate policies, it may indicate that climate policies, when present, have not yet produced sufficient results to change the emissions trajectory substantially (Nisbet et al., 2019). After 2015, the SSPs span a range of possible outcomes, but current emissions appear likely to follow the higher-emission trajectories over the next decade. This illustrates the challenge of methane mitigation that lies ahead to help reach the goals of the Paris agreement. In addition, estimates of methane atmospheric concentrations from the unharmonized scenarios (Riahi et al., 2017) indicate that observations of global methane concentrations fall well within the range of scenarios (Fig. 2 bottom). The methane concentrations are estimated using a simple exponential decay with inferred natural emissions (Meinshausen et al., 2011), and the emergence of any trend between observations and scenarios needs to be confirmed in the following years. In the future, it will be important to monitor the trends from

year 2015 (the Paris agreement) estimated in inventories and from atmospheric observations, and compare them to various scenarios.

### 3.1.3 Fossil fuel production and use

Most anthropogenic methane emissions related to fossil fuels come from the exploitation, transportation, and usage of coal, oil, and natural gas. Additional emissions reported in this category include small industrial contributions such as production of chemicals and metals, fossil fuel fires (e.g., underground coal mine fires and the Kuwait oil and gas fires), and transport (road and non-road transport). Methane emissions from the
610 oil industry (e.g. refining) and production of charcoal are estimated to be a few Tg $CH_4$ $yr^{-1}$ only and are included in the transformation industry sector in the inventory. Fossil fuel fires are included in the sub-category "Oil & Gas". Emissions from industries and road and non-road transport are reported apart from the two main sub-categories "Oil & Gas" and "Coal", contrary to Saunois et al. (2016); each of theses amounts to about 5 Tg $CH_4$ $yr^{-1}$ (Table 3). The large range (0-12 Tg $CH_4$ $yr^{-1}$) is attributable to difficulties in allocating
some sectors to these sub-sectors consistently among the different inventories (See Table S2). The spatial distribution of methane emissions from fossil fuels is presented in Fig. 3 based on the mean gridded maps provided by CEDS, EDGARv4.3.2 and GAINS for the 2008-2017 decade; USEPA lacks a gridded product. Global mean emissions from fossil fuel related activities, other industries and transport are estimated from the four global inventories (Table 1) to be of 128 [113-154] Tg $CH_4$ $yr^{-1}$ for the 2008-2017 decade (Table 3),
but with large differences in the rate of change during this period across inventories. The sector accounts on average for 35% (range 30-42%) of total global anthropogenic emissions.

**Coal mining.**

During mining, methane is emitted primarily from ventilation shafts, where large volumes of air are pumped into the mine to keep the $CH_4$ mixing ratio below 0.5% to avoid accidental ignition, and from dewatering
operations. In countries of the Organization for Economic Co-operation and Development (OECD), methane released from ventilation shafts is in principle used as fuel, but in many countries, it is still emitted into the atmosphere or flared, despite the efforts for coalmine recovery under the UNFCCC Clean Development Mechanisms (http://cdm.unfccc.int). Methane leaks also occur during post-mining handling, processing, and transportation. Some $CH_4$ is released from coal waste piles and abandoned mines; while emissions from these
sources were believed to be low (IPCC, 2000), recent work has estimated these at 22 billion $m^3$ (against 103 billion $m^3$ from functioning coal mines) in 2010 with emissions projected to increase into the future (Kholod et al., 2020).

In 2017, almost 40% (IEA, 2019b) of the world's electricity is still produced from coal. This contribution grew in the 2000s at the rate of several per cent per year, driven by Asian economic growth where large
reserves exist, but global coal consumption declined since 2014. In 2018, the top ten largest coal producing

nations accounted for ~90% of total world methane emissions for coal mining; among them, the top three producers (China, USA and India) produced almost two thirds (64%) of the world's coal (IEA, 2019a).

Global estimates of $CH_4$ emissions from coal mining show a large range of 29-61 Tg $CH_4$ $yr^{-1}$ for 2008-2017, in part due to the lack of comprehensive data from all major producing countries. The highest value of the range comes from the CEDS inventory while the lowest comes from USEPA. CEDS seems to have overestimated coal mining emissions from China by almost a factor of 2, most likely due to its dependence on the EDGARv4.2 emission inventory. As highlighted by Saunois et al. (2016), a county-based inventory of Chinese methane emissions also confirms the overestimate of about +38% with total anthropogenic emissions estimated at 43±6 Tg $CH_4$ $yr^{-1}$ (Peng et al., 2016). EDGARv4.2 inventory follows the IPCC guidelines and uses European averaged emission factor for $CH_4$ from coal production to substitute missing data for China, which appears to be overestimated by a factor of approximately two. These differences highlight significant errors resulting from the use of emission factors, and that applying "Tier 1" approaches for coal mine emissions is not sufficiently accurate as stated by the IPCC guidelines. The newly released version of EDGARv4.3.2 used here has revised China coal methane emission factors downwards and distributed them to more than 80 times more coal mining locations in China. Coal mining emission factors depend strongly on the type of coal extraction (underground mining emits up to 10 times more than surface mining), the geological underground structure (region-specific), history (basin uplift), and the quality of the coal (brown coal emits more than hard coal). Finally, coal mining is the main source explaining the differences between inventories globally (Fig. 2).

For the 2008-2017 decade, methane emissions from coal mining represent 33% of total fossil fuel related emissions of methane (42 Tg $CH_4$ $yr^{-1}$, range of 29-61). An additional very small source corresponds to fossil fuel fires (mostly underground coal fires, ~0.15 Tg $yr^{-1}$ in 2012, EDGARv4.3.2).

**Oil and natural gas systems.**

This sub-category includes emissions from both conventional and shale oil and gas exploitation. Natural gas is comprised primarily of methane, so both fugitive and planned emissions during the drilling of wells in gas fields, extraction, transportation, storage, gas distribution, end use, and incomplete combustion of gas flares emit methane (Lamb et al., 2015; Shorter et al., 1996). Persistent fugitive emissions (e.g., due to leaky valves and compressors) should be distinguished from intermittent emissions due to maintenance (e.g. purging and draining of pipes). During transportation, fugitive emissions can occur in oil tankers, fuel trucks and gas transmission pipelines, attributable to corrosion, manufacturing and welding faults. According to Lelieveld et al. (2005), $CH_4$ fugitive emissions from gas pipelines should be relatively low, however distribution networks in older cities may have higher rates, especially those with cast-iron and unprotected steel pipelines (Phillips et al., 2013). Measurement campaigns in cities within USA and Europe revealed that significant emissions occur in specific locations (e.g. storage facilities, city gates, well and pipeline

pressurization/depressurization points) along the distribution networks (e.g. Jackson et al., 2014a; McKain et al., 2015; Wunch et al., 2016). However, methane emissions vary significantly from one city to another depending, in part, on the age of city infrastructure and the quality of its maintenance, making urban emissions difficult to scale-up. In many facilities, such as gas and oil fields, refineries and offshore platforms, venting of natural gas is now replaced by flaring with almost complete conversion to $CO_2$; these two processes are usually considered together in inventories of oil and gas industries. Also, single-point failure of natural gas infrastructure can leak methane at high rate for months, such as at the Aliso Canyon blowout in the Los Angeles, CA, basin (Conley et al., 2016) or the recent shale gas well blowout in Ohio (Pandey et al., 2019), thus hampering emission control strategies. Production of natural gas from the exploitation of hitherto unproductive rock formations, especially shale, began in the 1970s in the US on an experimental or small-scale basis, and then, from early 2000s, exploitation started at large commercial scale. The shale gas contribution to total dry natural gas production in the United States reached 62% in 2017, growing rapidly from 40% in 2012, with only small volumes produced before 2005 (EIA, 2019). The possibly larger emission factors from the shale gas as compared to the conventional ones, have been widely debated (e.g. Cathles et al., 2012; Howarth, 2019; Lewan, 2020). However, the latest studies tend to infer similar emission factors in a narrow range of 1-3% (Alvarez et al., 2018; Peischl et al., 2015; Zavala-Araiza et al., 2015), different from the widely spread rates of 3-17% from previous studies (e.g. Caulton et al., 2014; Schneising et al., 2014). Methane emissions from oil and natural gas systems vary greatly in different global inventories (72 to 97 Tg $yr^{-1}$ in 2017, Table 3). The inventories generally rely on the same sources and magnitudes for activity data, with the derived differences therefore resulting primarily from different methodologies and parameters used, including emission factors. Those factors are country- or even site-specific and the few field measurements available often combine oil and gas activities (Brandt et al., 2014) and remain largely unknown for most major oil- and gas-producing countries. Depending on the country, the reported emission factors may vary by two orders of magnitude for oil production and by one order of magnitude for gas production (Table SI-5.1 of Höglund-Isaksson (2017)). The GAINS estimate of methane emissions from oil production, for instance, is twice as high as EDGARv4.3.2. For natural gas, the uncertainty is of a similar order of magnitude. During oil extraction, natural gas generated can be either recovered (re-injected or utilized as an energy source) or not recovered (flared or vented to the atmosphere). The recovery rates vary from one country to another (being much higher in the USA, Europe and Canada than elsewhere), and from one type of oil to another: flaring is less common for heavy oil wells than for conventional ones (Höglund-Isaksson et al., 2015). Considering recovery rates could lead to two-times-higher methane emissions accounting for country-specific rates of generation and recovery of associated gas than when using default values (Höglund-Isaksson, 2012). This difference in methodology explains, in part, why GAINS estimates are higher than those of EDGARv4.3.2.

Most studies (Alvarez et al., 2018; Brandt et al., 2014; Jackson et al., 2014b; Karion et al., 2013; Moore et al., 2014; Olivier and Janssens-Maenhout, 2014; Pétron et al., 2014; Zavala-Araiza et al., 2015), albeit not all (Allen et al., 2013; Cathles et al., 2012; Peischl et al., 2015), suggest that methane emissions from oil and gas industry are underestimated by inventories and agencies, including the USEPA. Zavala-Araiza et al. (2015) showed that a few high-emitting facilities, i.e., super-emitters, neglected in the inventories, dominated USA emissions. These high emitting points, located on the conventional part of the facility, could be avoided through better operating conditions and repair of malfunctions. As USA production increases, absolute methane emissions almost certainly increase, as well USA crude oil production doubled over the last decade and natural gas production rose more than 50% (EIA, 2019). However, global implications of the rapidly growing shale gas activity in the US remains to be determined precisely.

For the 2008-2017 decade, methane emissions from upstream and downstream oil and natural gas sectors are estimated to represent about 63% of total fossil $CH_4$ emissions (80 Tg $CH_4$ yr$^{-1}$, range of 68-92 Tg $CH_4$ yr$^{-1}$, Table 3), with a lower uncertainty range than for coal emissions for most countries.

### 3.1.4 Agriculture and waste sectors

This main category includes methane emissions related to livestock production (i.e., enteric fermentation in ruminant animals and manure management), rice cultivation, landfills, and wastewater handling. Of these, globally and in most countries, livestock is by far the largest source of $CH_4$, followed by waste handling and rice cultivation. Conversely, field burning of agricultural residues is a minor source of $CH_4$ reported in emission inventories. The spatial distribution of methane emissions from agriculture and waste handling is presented in Fig. 3 based on the mean gridded maps provided by CEDS, EDGARv4.3.2 and GAINS over the 2008-2017 decade.

Global emissions from agriculture and waste for the period 2008-2017 are estimated to be 206 Tg $CH_4$ yr$^{-1}$ (range 191-223, Table 3), representing 56% of total anthropogenic emissions.

**Livestock: Enteric fermentation and manure management.** Domestic ruminants such as cattle, buffalo, sheep, goats, and camels emit methane as a by-product of the anaerobic microbial activity in their digestive systems (Johnson et al., 2002). The very stable temperatures (about 39°C) and pH (6.5-6.8) values within the rumen of domestic ruminants, along with a constant plant matter flow from grazing (cattle graze many hours per day), allow methanogenic *Archaea* residing within the rumen to produce methane. Methane is released from the rumen mainly through the mouth of multi-stomached ruminants (eructation, ~87% of emissions) or absorbed in the blood system. The methane produced in the intestines and partially transmitted through the rectum is only ~13%.

The total number of livestock continues to grow steadily. There are currently (2017) about 1.5 billion cattle globally, 1 billion sheep, and nearly as many goats (http://www.fao.org/faostat/en/#data/GE). Livestock

numbers are linearly related to CH$_4$ emissions in inventories using the Tier 1 IPCC approach such as FAOSTAT. In practice, some non-linearity may arise due to dependencies of emissions on total weight of the animals and their diet, which are better captured by Tier 2 and higher approaches. Cattle, due to their large population, large individual size, and particular digestive characteristics, account for the majority of enteric fermentation CH$_4$ emissions from livestock worldwide (Tubiello, 2019), particularly in intensive agricultural systems in wealthier and emerging economies, including the United States (USEPA, 2016). Methane emissions from enteric fermentation also vary from one country to another as cattle may experience diverse living conditions that vary spatially and temporally, especially in the tropics (Chang et al., 2019).

Anaerobic conditions often characterize manure decomposition in a variety of manure management systems globally (e.g., liquid/slurry treated in lagoons, ponds, tanks, or pits), with the volatile solids in manure producing CH$_4$. In contrast, when manure is handled as a solid (e.g., in stacks or dry-lots) or deposited on pasture, range, or paddock lands, it tends to decompose aerobically and to produce little or no CH$_4$. However aerobic decomposition of manure tends to produce nitrous oxide (N$_2$O), which has a larger warming impact than CH$_4$. Ambient temperature, moisture, energy contents of the feed, manure composition, and manure storage or residency time affect the amount of CH$_4$ produced. Despite these complexities, most global datasets used herein apply a simplified IPCC Tier 1 approach, where amounts of manure treated depend on animal numbers and simplified climatic conditions by country.

Global methane emissions from enteric fermentation and manure management are estimated in the range of 99-115 Tg CH$_4$ yr$^{-1}$, for the year 2010, in the GAINS model and CEDS, USEPA, FAO-CH$_4$ and EDGARv4.3.2 inventories. These values are slightly higher than the IPCC Tier 2 estimate of Dangal et al. (2017) (95.7 Tg CH$_4$/yr for 2010) and the IPCC Tier 3 estimates of Herrero et al. (2013) (83.2 Tg CH$_4$ yr$^{-1}$ for 2000), but in agreement with the recent IPCC Tier 2 estimate of Chang et al. (2019) (99± 12 Tg CH$_4$ yr$^{-1}$ for 2012).

For the period 2008-2017, we estimated total emissions of 111 [106-116] Tg CH$_4$ yr$^{-1}$ for enteric fermentation and manure management, about one third of total global anthropogenic emissions.

**Rice cultivation.** Most of the world's rice is grown in flooded paddy fields (Baicich, 2013). The water management systems, particularly flooding, used to cultivate rice are one of the most important factors influencing CH$_4$ emissions and one of the most promising approaches for CH$_4$ emission mitigation: periodic drainage and aeration not only cause existing soil CH$_4$ to oxidize, but also inhibit further CH$_4$ production in soils (Simpson et al., 1995; USEPA, 2016; Zhang, 2016). Upland rice fields are not typically flooded, and therefore are not a significant source of CH$_4$. Other factors that influence CH$_4$ emissions from flooded rice fields include fertilization practices (i.e. the use of urea and organic fertilizers), soil temperature, soil type (texture and aggregated size), rice variety and cultivation practices (e.g., tillage, seeding, and weeding practices) (Conrad et al., 2000; Kai et al., 2011; USEPA, 2011; Yan et al., 2009). For instance, methane

emissions from rice paddies increase with organic amendments (Cai et al., 1997) but can be mitigated by applying other types of fertilizers (mineral, composts, biogas residues) or using wet seeding (Wassmann et al., 2000).

The geographical distribution of rice emissions has been assessed by global (e.g. Janssens-Maenhout et al., 2019; Tubiello, 2019; USEPA, 2012) and regional (e.g. Castelán-Ortega et al., 2014; Chen et al., 2013; Chen and Prinn, 2006; Peng et al., 2016; Yan et al., 2009; Zhang and Chen, 2014) inventories or land surface models (Li et al., 2005; Pathak et al., 2005; Ren et al., 2011; Spahni et al., 2011; Tian et al., 2010, 2011; Zhang, 2016). The emissions show a seasonal cycle, peaking in the summer months in the extra-tropics associated with monsoons and land management. Similar to emissions from livestock, emissions from rice paddies are influenced not only by extent of rice field area (analogous to livestock numbers), but also by changes in the productivity of plants (Jiang et al., 2017) as these alter the $CH_4$ emission factor used in inventories. Nonetheless, the inventories considered herein are largely based on IPCC Tier 1 methods, which largely scale with cultivated areas but include regional specific emission factors.

The largest emissions from rice cultivation are found in Asia accounting for 30 to 50% of global emissions (Fig. 3). The decrease of $CH_4$ emissions from rice cultivation over recent decades is confirmed in most inventories, because of the decrease in rice cultivation area, changes in agricultural practices, and a northward shift of rice cultivation since the 1970s, as in China (e.g. Chen et al., 2013).

Based on the global inventories considered in this study, global methane emissions from rice paddies are estimated to be 30 [25-38] Tg $CH_4$ $yr^{-1}$ for the 2008-2017 decade (Table 3), or about 8% of total global anthropogenic emissions of methane. These estimates are consistent with the 29 Tg $CH_4$ $yr^{-1}$ estimated for the year 2000 by Carlson et al. (2017).

**Waste management.** This sector includes emissions from managed and non-managed landfills (solid waste disposal on land), and wastewater handling, where all kinds of waste are deposited. Methane production from waste depends on the pH, moisture, and temperature of the material. The optimum pH for methane emission is between 6.8 and 7.4 (Thorneloe et al., 2000). The development of carboxylic acids leads to low pH, which limits methane emissions. Food or organic waste, leaves and grass clippings ferment quite easily, while wood and wood products generally ferment slowly, and cellulose and lignin even more slowly (USEPA, 2010a). Waste management was responsible for about 11% of total global anthropogenic methane emissions in 2000 (Kirschke et al., 2013). A recent assessment of methane emissions in the U.S. found landfills to account for almost 26% of total U.S. anthropogenic methane emissions in 2014, the largest contribution of any single $CH_4$ source in the United States (USEPA, 2016). In Europe, gas control has been mandatory on all landfills since 2009, following the ambitious objective raised in the EU Landfill Directive (1999) to reduce landfilling of biodegradable waste to 65% below the 1990 level by 2016. This mitigation is attempted through source separation and treatment of separated biodegradable waste in composts, bio-digesters, and paper recycling.

Wastewater from domestic and industrial sources is treated in municipal sewage treatment facilities and private effluent treatment plants. The principal factor in determining the $CH_4$ generation potential of wastewater is the amount of degradable organic material in the wastewater. Wastewater with high organic content is treated anaerobically, which leads to increased emissions (André et al., 2014). Excessive and rapid urban development worldwide, especially in Asia and Africa, could enhance methane emissions from waste unless adequate mitigation policies are designed and implemented rapidly.

The GAINS model and CEDS and EDGAR inventories give robust emission estimates from solid waste in the range of 29-41 Tg $CH_4$ yr$^{-1}$ for the year 2005, and more uncertain wastewater emissions in the range 14-33 Tg $CH_4$ yr$^{-1}$1.

In our study, the global emission of methane from waste management is estimated in the range of 60-69 Tg $CH_4$ yr$^{-1}$ for the 2008-2017 period with a mean value of 65 Tg $CH_4$ yr$^{-1}$, about 12% of total global anthropogenic emissions.

### 3.1.5 Biomass and biofuel burning

This category includes methane emissions from biomass burning in forests, savannahs, grasslands, peats, agricultural residues, as well as, from the burning of biofuels in the residential sector (stoves, boilers, fireplaces). Biomass and biofuel burning emits methane under incomplete combustion conditions (i.e., when oxygen availability is insufficient for complete combustion), for example in charcoal manufacturing and smouldering fires. The amount of methane emitted during the burning of biomass depends primarily on the amount of biomass, burning conditions, and the specific material burned.

In this study, we use large-scale biomass burning (forest, savannah, grassland and peat fires) from five biomass burning inventories (described below) and the biofuel burning contribution from anthropogenic emission inventories (EDGARv4.3.2, CEDS, GAINS and USEPA). The spatial distribution of emissions from the burning of biomass and biofuel over the 2008-2017 decade is presented in Fig. 3 based on data listed in Table 1.

At the global scale, during the period of 2008-2017, biomass and biofuel burning generated methane emissions of 30 [26-40] Tg $CH_4$ yr$^{-1}$ (Table 3), of which 30-50 % is from biofuel burning.

**Biomass burning.** Fire is an important disturbance event in terrestrial ecosystems globally (van der Werf et al., 2010), and can be of either natural (typically ~10% of fires, ignited by lightning strikes or started accidentally) or anthropogenic origin (~90%, human initiated fires) (USEPA (2010b) chapter 9.1). Anthropogenic fires are concentrated in the tropics and subtropics, where forests, savannahs and grasslands may be burned to clear land for agricultural purposes or to maintain pastures and rangelands. Small fires

associated with agricultural activity, such as field burning and agricultural waste burning, are often not well detected by remote sensing methods and are instead estimated based on cultivated area.

Emission rates of biomass burning vary with biomass loading (depending on the biomes) at the location of the fire, the efficiency of the fire (depending on the vegetation type), the fire type (smoldering or flaming) and emission factor (mass of the considered species / mass of biomass burned). Depending on the approach, these parameters can be derived using satellite data and/or biogeochemical model, or through simpler IPCC default approaches.

In this study, we use five products to estimate biomass burning emissions. The Global Fire Emission Database (GFED) is the most widely used global biomass burning emission dataset and provides estimates from 1997. Here, we use GFEDv4.1s (van der Werf et al., 2017), based on the Carnegie-Ames-Stanford-Approach (CASA) biogeochemical model and satellite derived estimates of burned area (from MODerate resolution Imaging Sensor, MODIS), fire activity and plant productivity. GFEDv4.1s (with small fires) is available at a 0.25° resolution and on a daily basis from 1997 to 2017. One characteristic of the GFEDv4.1s burned area is that small fires are better accounted compared to GFEDv4.1 (Randerson et al., 2012), increasing carbon emissions by approximately 35% at the global scale.

The Quick Fire Emissions Dataset (QFED) is calculated using the fire radiative power (FRP) approach, in which the thermal energy emitted by active fires (detected by MODIS) is converted to an estimate of methane flux using biome specific emissions factors and a unique method of accounting for cloud cover. Further information related to this method and the derivation of the biome specific emission factors can be found in Darmenov and da Silva (Darmenov and da Silva, 2015). Here we use the historical QFEDv2.5 product available daily on a 0.1x0.1 grid for 2000 to 2017.

The Fire Inventory from NCAR (FINN, Wiedinmyer et al., 2011) provides daily, 1km resolution estimates of gas and particle emissions from open burning of biomass (including wildfire, agricultural fires and prescribed burning) over the globe for the period 2002-2018. FINNv1.5 uses MODIS satellite observations for active fires, land cover and vegetation density.

We use v1.3 of the Global Fire Assimilation System (GFAS, Kaiser et al., 2012), which calculates emissions of biomass burning by assimilating Fire Radiative Power (FRP) observations from MODIS at a daily frequency and 0.5° resolution and is available for 2000-2016.

The FAO-$CH_4$ yearly biomass burning emissions are based on the most recent MODIS 6 burned area products, coupled with a pixel level (500m) implementation of the IPCC Tier 1 approach, and are available from 1990 to 2016 (Table 1).

The differences in emission estimates for biomass burning arise from specific geographical and meteorological conditions and fuel composition, which strongly impact combustion completeness and emission factors. The latter vary greatly according to fire type, ranging from 2.2 g $CH_4$ kg$^{-1}$ dry matter burned

for savannah and grassland fires up to 21 g $CH_4$ $kg^{-1}$ dry matter burned for peat fires (van der Werf et al., 2010).

In this study, based on the five aforementioned products, biomass burning emissions are estimated at 17 Tg $CH_4$ $yr^{-1}$ [14-26] for 2008-2017, representing about 5% of total global anthropogenic methane emissions.

**Biofuel burning.** Biomass that is used to produce energy for domestic, industrial, commercial, or transportation purposes is hereafter called biofuel burning. A largely dominant fraction of methane emissions from biofuels comes from domestic cooking or heating in stoves, boilers and fireplaces, mostly in open cooking fires where wood, charcoal, agricultural residues, or animal dung are burned. It is estimated that
more than two billion people, mostly in developing countries, use solid biofuels to cook and heat their homes daily (André et al., 2014), and yet methane emissions from biofuel combustion have received relatively little attention. Biofuel burning estimates are gathered from the CEDS, USEPA, GAINS and EDGAR inventories. Due to the sectoral breakdown of the EDGAR and CEDS inventories the biofuel component of the budget has been estimated as equivalent to the "RCO - Energy for buildings" sector as defined in Worden et al.
(2017) and Hoesly et al. (2018) (See Table S2). This is equivalent to the sum of the IPCC 1A4a_Commercial-institutional, 1A4b_Residential, 1A4c_Agriculture-forestry-fishing and 1A5_Other-unspecified reporting categories. This definition is consistent with that used in Saunois et al. (2016) and Kirschke et al. (2013). While this sector incorporates biofuel use it also includes the use of other combustible materials (e.g. coal or gas) for small scale heat and electricity generation within residential and commercial premises. Data provided
by the GAINS inventory suggests that this approach may overestimate biofuels emissions by between 5 and 50%.

In our study, biofuel burning is estimated to contribute 12 Tg $CH_4$ $yr^{-1}$ [10-14] to the global methane budget, about 3% of total global anthropogenic methane emissions for 2008-2017.

### 3.1.6 Other anthropogenic sources (not explicitly included in this study)

Other anthropogenic sources not included in this study are related to agriculture and land-use management. In particular, increases in global palm oil production have led to the clearing of natural peat forests, reducing natural peatland area and associated natural $CH_4$ emissions. While studies have long suggested that $CH_4$ emissions from peatland drainage ditches are likely to be significant (e.g. Minkkinen and Laine, 2006), $CH_4$ emissions related to palm oil plantations have yet to be properly quantified. Taylor et al. (2014) have
quantified global palm oil wastewater treatment fluxes to be $4 \pm 32$ Tg $CH_4$ $yr^{-1}$ for 2010-2013. This currently represents a small and highly uncertain source of methane but one potentially growing in the future.

## 3.2 Natural sources

Natural methane sources include vegetated wetland emissions and inland water systems (lakes, small ponds, rivers), land geological sources (gas-oil seeps, mud volcanoes, microseepage, geothermal manifestations and volcanoes), wild animals, wildfires, termites, thawing terrestrial and marine permafrost and oceanic sources (biogenic, geological and hydrate). In water-saturated or flooded ecosystems, the decomposition of organic matter gradually depletes most of the oxygen in the soil, resulting in anaerobic conditions and methane production. Once produced, methane can reach the atmosphere through a combination of three processes: (1) diffusive loss of dissolved $CH_4$ across the air-water boundary; (2) ebullition flux from sediments and (3) flux mediated by emergent aquatic macrophytes and terrestrial plants (plant transport). On its way to the atmosphere, in the soil or water columns, methane can be partly or completely oxidized by a group of bacteria called methanotrophs, which use methane as their only source of energy and carbon (USEPA, 2010b). Concurrently, methane from the atmosphere can diffuse into the soil column and be oxidized (See Sect. 3.3.4 on soil uptake).

### 3.2.1 Wetlands

Wetlands are generally defined as ecosystems in which soils or peats are water saturated or where surface inundation (permanent or not) dominates the soil biogeochemistry and determines the ecosystem species composition (USEPA, 2010b). In order to refine such overly broad definition for methane emissions, we define wetlands as ecosystems with inundated or saturated soils or peats where anaerobic conditions lead to methane production (Matthews and Fung, 1987; USEPA, 2010b). Brackish water emissions are discussed separately in Sect. 3.2.6. Our definition of wetlands includes peatlands (bogs and fens), mineral soil wetlands (swamps and marshes), and seasonal or permanent floodplains. It excludes exposed water surfaces without emergent macrophytes, such as lakes, rivers, estuaries, ponds, and reservoirs (addressed in the next section), as well as rice agriculture (see Sect. 3.1.4, rice cultivation paragraph), and wastewater ponds. It also excludes coastal vegetated ecosystems (mangroves, seagrasses, salt marshes) with salinities usually >0.5 psu (See Sect. 3.2.6). Even with this definition, some wetlands could be considered as anthropogenic systems, being affected by human land-use changes such as impoundments, drainage, or restoration (Woodward et al., 2012). In the following we retain the generic denomination "wetlands" for natural and human-influenced wetlands, as discussed in Sect. 2.2.

The three most important factors influencing methane production in wetlands are the spatial and temporal extent of anoxia (linked to water saturation), temperature and substrate availability (Valentine et al., 1994; Wania et al., 2010; Whalen, 2005).

Land-surface models estimate $CH_4$ emissions through a series of processes, including $CH_4$ production, oxidation and transport. The models are then forced with inputs accounting for changing environmental factors (Melton et al., 2013; Poulter et al., 2017; Tian et al., 2010; Wania et al., 2013; Xu et al., 2010). Methane emissions from wetlands are computed as the product of an emission flux density and a methane producing area or surface extent (see Supplementary Material; Bohn et al., 2015; Melton et al., 2013). Wetland extent appears to be a primary contributor to uncertainties in the absolute flux of methane emissions from wetlands, with meteorological response the main source of uncertainty for seasonal and interannual variability (Bohn et al., 2015; Desai et al., 2015; Poulter et al., 2017).

In this work, thirteen land surface models computing net $CH_4$ emissions (Table 2) were run under a common protocol with a 30-year spin-up (1901-1930) followed by a simulation through the end of 2017 forced by CRU-JRA reconstructed climate fields (Harris, 2019). Of the 13 models, 10 previously contributed to Saunois et al. (2016), three models were new to this release (JSBACH, LPJ-GUESS and TEM-MDM) (Table S3). Atmospheric $CO_2$ influencing wetland Net Primary Production (NPP) was also prescribed in the models. In all models, the same remote sensing-based wetland area and dynamics dataset called WAD2M (Wetland Area Dynamics for Methane Modeling) was prescribed. WAD2M provides year to year varying monthly global wetland areas over 2000-2017, partly addressing known issues, such as separation between wetlands and other inland waters (Poulter et al., 2017). WAD2M combines microwave remote sensing data from Schroeder et al. (2015) with various regional inventory datasets to develop a monthly global wetland area dataset, which will be further presented in the near future by Poulter and colleagues. Non-vegetated wetland inland waters (i.e., lakes, rivers and ponds) were subtracted using the Global Surface Waters dataset of Pekel et al. (2016), assuming that permanent waters were those that were present > 50% of the time within a 32-year observing period. Then, wetland inventories for the tropics (Gumbricht et al., 2017), high-latitudes (Hugelius et al., 2014; Widhalm et al., 2015) and temperate regions (Lehner and Döll, 2004) were used to set the long-term annual mean wetland area, to which a seasonal cycle of fractional surface water was added using data from the Surface WAter Microwave Product Series Version 3.2 (SWAMPS) (Jensen and Mcdonald, 2019; Schroeder et al., 2015). Rice agriculture was removed using the MIRCA2000 dataset from circa 2000, as a fixed distribution. The combined remote-sensing and inventory WAD2M product leads to a maximum wetland area of 14.9 $Mkm^2$ during the peak season (8.4 $Mkm^2$ on annual average, with a range of 8.0 to 8.9 $Mkm^2$ from 2000-2017, about 5.5% of the global land surface). The largest wetland areas in WAD2M are in Amazonia, the Congo Basin, and the Western Siberian Lowlands, which in previous studies were underestimated by inventories (Bohn et al., 2015).

The average emission map from wetlands for 2008-2017 built from the 13 models is plotted in Fig. 3. The zones with the largest emissions are the Amazon basin, equatorial Africa and Asia, Canada, western Siberia, eastern India, and Bangladesh. Regions where methane emissions are robustly inferred (defined as regions

where mean flux is larger than the standard deviation of the models) represent 61% of the total methane flux due to natural wetlands. This contribution is 80% lower than found in Saunois et al. (2016) probably due to the different ensemble of models gathered here and the more stringent exclusion of inland waters. The main primary emission zones are consistent between models, which is clearly favoured by the prescribed common wetland extent. However, the different sensitivities of the models to temperature, vapour pressure, precipitation, and radiation can generate substantially different patterns, such as in India. Some secondary (in magnitude) emission zones are also consistently inferred between models: Scandinavia, Continental Europe, Eastern Siberia, Central USA, and tropical Africa.

The resulting global flux range for natural wetland emissions is 101-179 Tg $CH_4$ yr$^{-1}$ for the 2000-2017 period, with an average of 148 Tg $CH_4$ yr$^{-1}$ and a one-sigma standard deviation of 25 Tg $CH_4$ yr$^{-1}$. For the last decade, 2008-2017, the average ensemble emissions were 149 Tg $CH_4$ yr$^{-1}$ with a range of 102-182 (Table 3). Using a prognostic set of simulations, where models used their own internal approach to estimate wetland area and dynamics, the average ensemble emissions were 161 Tg $CH_4$ yr$^{-1}$ with a range of 125-218 for the 2008-2017 period. The greater range of uncertainty from prognostic area models is due to unconstrained wetland area, but generally the magnitude and interannual variability agree between diagnostic and prognostic area approaches. Wetland emissions represent about 20% of the total (natural plus anthropogenic) methane sources estimated by bottom-up approaches. The large range in the estimates of wetland $CH_4$ emissions results from difficulties in defining wetland $CH_4$ producing areas as well as in parameterizing terrestrial anaerobic conditions that drive sources and the oxidative conditions leading to sinks (Melton et al., 2013; Poulter et al., 2017; Wania et al., 2013). The ensemble mean emission using diagnostic wetland extent in the models is lower by ~35 Tg $CH_4$ yr$^{-1}$ than the one previously reported (see Table 3, for 2000-2009 with comparison to Saunois et al., 2016). This difference results from a reduction in double counting due to i) decreased wetland area in WAD2M, especially for high-latitude regions where inland waters, i.e., lakes, small ponds and lakes, were removed, and ii) to some extent, an improved removal of rice agriculture area using the MIRCA-2000 database.

For the last decade, 2008-2017, the average ensemble emissions were 149 Tg $CH_4$ yr$^{-1}$ with a range of 102-182.

### 3.2.2 Other inland water systems (lakes, ponds, reservoirs, streams, rivers)

This category includes methane emissions from freshwater systems (lakes, ponds, reservoirs, streams and rivers). To date, very few process-based models exist for these fluxes, relying on data driven approaches and extrapolations. Meta-data analyses are hampered for methane due to a mix of methodological approaches, which capture different components of emissions, and different scales in space and time, depending on method and time of deployment and data processing (Stanley et al., 2016). Altogether, this inconsistency in

the data collection makes detailed modelling of fluxes highly uncertain. For many lakes, particularly smaller
shallower lakes and ponds, it is established that ebullition and plant fluxes (in lakes with substantial emergent
macrophyte communities) can make up a substantial contribution to fluxes, potentially accounting for 50%
to more than 90% of the flux from these water bodies. While contributions from ebullition appear lower from
rivers, there are currently insufficient measurements from these systems to determine its role (Crawford et
al., 2014; Stanley et al., 2016). Ebullition fluxes are very challenging to measure, due to the high degree of
spatiotemporal variability with very high fluxes occurring in parts of an ecosystem over the time frames of
seconds followed by long periods without ebullition.

**Streams and rivers.** Freshwater methane fluxes from streams and rivers were first estimated to be 1.5 Tg
$CH_4$ yr$^{-1}$ (Bastviken et al., 2011). However, this study had measurements from only 21 sites globally. More
recently, Stanley et al. (2016) compiled a data set of 385 sites and estimated a diffusive emission of 27 Tg
$CH_4$ yr$^{-1}$ (5th–95th percentiles: 0.01–160 Tg $CH_4$ yr$^{-1}$). Detailed regional studies in the tropics and temperate
watersheds (Borges et al., 2015; Campeau and del Giorgio, 2014) support a flux in the range of 27 Tg $CH_4$
yr$^{-1}$ as opposed to the initial ~1.5 Tg $CH_4$ yr$^{-1}$. However, the low number of measurements, the lack of clarity
on ebullitive fluxes, and the large degree of variance in measurements have precluded an accurate spatial
representation of stream and river methane fluxes. Canals and ditches have recently been highlighted as high
areal emitters (e.g. Stanley et al., 2016), and their contribution to large-scale emission is typically included
in estimates for overall running waters so far. No new global estimates have been published since Stanley et
al. (2016) and Saunois et al. (2016). As a result, we use here the same estimate for stream and rivers as in
Saunois et al. (2016): 27 Tg $CH_4$ yr$^{-1}$.

**Lakes and ponds.** Methane emissions from lakes were first estimated to be 1-20 Tg $CH_4$ yr$^{-1}$ based on
measurements in two systems (Great Fresh Creek, Maryland and Lake Erie; Ehhalt (1974)). A subsequent
global emission estimate was 11-55 Tg $CH_4$ yr$^{-1}$ based on measurements from three Arctic lakes and a few
temperate and tropical systems (Smith and Lewis, 1992), and 8-48 Tg $CH_4$ yr$^{-1}$ using extended data from
different latitudes (73 lakes, Bastviken et al. (2004)). Based on data from 421 lakes and ponds, Bastviken et
al. (2011) updated their values to 71.6 Tg $CH_4$ yr$^{-1}$, including emissions from non-saline lakes and ponds.
High-latitude lakes have received a large amount of attention in the last decade. They include both post-
glacial and thermokarst lakes (small water bodies formed when peat over melting permafrost collapse), the
latter having larger emissions per m$^2$ but smaller regional emissions than the former because of their smaller
areal extent (Wik et al., 2016b). Water body depth, sediment type, and eco-climatic region are the key factors
explaining variation in methane fluxes from lakes (Wik et al., 2016b). Small artificial water-bodies (ponds)
have a high surface area to volume ratio, and shallow depth, and are likely to be a notable source of methane,
at least at the regional scale (Grinham et al., 2018; Ollivier et al., 2019). These studies found that emissions
varied by pond type (for example: livestock rearing farm dams vs. cropping farm dams vs. urban ponds vs.

weirs). A rough estimate of the global impact of this emission source is globally significant, between 3 and
8 Tg $CH_4$ $yr^{-1}$ (calculated using the mean emission rates from Grinham et al. (2018) and Ollivier et al. (2019))
and an estimate of global farm impoundment surface area of 77,000 $km^2$ (Downing et al., 2006). This rough
estimate does emphasise the potential significance of these sources, although double counting with current
uncertain estimates from natural inland water systems is possible (Thornton et al., 2016a).

A regional estimate for latitudes above 50° North (Wik et al., 2016b) estimated lake and pond methane
emissions to be 16.5 Tg $CH_4$ $yr^{-1}$ (compared to 13.4 Tg $CH_4$ $yr^{-1}$ in Bastviken et al. (2011), above 54 °N).
Tan et al. (2016) used atmospheric inversion approaches and estimated that the current pan-Arctic (north of
60 °N) lakes emit 2.4-14.2 Tg $CH_4$ $yr^{-1}$, while a process-based lake biogeochemistry model (bLake4Me)
estimated the emissions at 11.9 [7.1-17.3] Tg $CH_4$ $yr^{-1}$ (Tan and Zhuang, 2015). These numbers for northern
or Arctic lakes need to be considered with regard to the latitudinal area encompassed which differ among
studies (Thornton et al., 2016a). Saunois et al. (2016) estimates for emissions from natural lakes and ponds
were based on Bastviken et al. (2011), using the emissions from the northern high latitudes above 50°N from
Wik et al. (2016b), leading to a rounded mean value of 75 Tg $CH_4$ $yr^{-1}$. Based on the bLake4Me gridded map
from Tan and Zhuang (2015), we calculate lake and pond emissions of 5.2 Tg $CH_4$ $yr^{-1}$ above 66°N, close to
the 6.8 Tg $CH_4$ $yr^{-1}$ found by Bastviken et al.(2011). Averaging these two values for the emissions above
66°N and combining with Bastviken et al. (2011) estimates south of 66°N (64.8 Tg $CH_4$ $yr^{-1}$) leads to a
rounded mean global estimate of 71 Tg $CH_4$ $yr^{-1}$ close to Bastviken et al. (2011) (71.6 Tg $CH_4$ $yr^{-1}$ at the
global scale).

**Reservoirs.** On top of emissions pathways described for inland waters, reservoirs have specific ones
including degassing of $CH_4$ from turbines (hydropower reservoirs only) and elevated diffusive emissions in
rivers downstream of the reservoir - these latter emissions are enhanced if the water outlet comes from anoxic
$CH_4$-rich hypolimnion waters in the reservoir (Bastviken et al., 2004; Guérin et al., 2006, 2016). In Saunois
et al. (2016), methane emissions from reservoirs were estimated to be 20 Tg $CH_4$ $yr^{-1}$ using Bastviken et al.
(2011), which was based on data from 32 systems. A more recent and extensive review estimated total
reservoir emissions to be 18 Tg $CH_4$ yr-1 (95% confidence interval 12-30 Tg $CH_4$ yr-1; $n$ = 75 (Deemer et
al., 2016)), and is used to revise our estimate in this study.

**Combination (lakes, ponds, reservoirs, streams and rivers).** Combining emissions from lakes and ponds
from Bastviken et al. (2011) (71.6 Tg $CH_4$ $yr^{-1}$) with the recent estimate of Deemer et al. (2016) for reservoirs
and the streams and river estimates from Stanley et al. (2016) leads to total inland freshwater emissions of
117 Tg $CH_4$ $yr^{-1}$. Recently, using a new up scaling approach based on size weighting productivity and
chlorophyll-A, DelSontro et al. (2018) provided a combined lake and reservoir estimates of 104 (5th–95th
percentiles: 67-165), 149 (5th–95th percentiles: 95-236) and 185 (5th–95th percentiles: 119-295) Tg $CH_4$ $yr^{-1}$, using the lake size distributions from Downing et al. (2006), Messager et al. (2016) and Verpooter et al.

(2014), respectively. These estimates are higher (by 10%, 57% and almost 100%, respectively) than previously reported in Saunois et al. (2016) (ie, 95 Tg CH$_4$ yr$^{-1}$ for lakes, ponds and reservoirs).

Previously, Kirschke et al. (2013) reported a range of 8-73 Tg CH4 yr-1 for this ensemble of emissions and Saunois et al. (2016) a mean value of 122 Tg CH$_4$ yr$^{-1}$ (75 Tg CH$_4$ yr$^{-1}$ for lakes and ponds, adding 20 Tg CH$_4$ yr$^{-1}$ for reservoirs (Bastviken et al., 2011) and 27 Tg CH$_4$ yr$^{-1}$ for streams and rivers (Stanley et al., 2016)). This mean value reported by Saunois et al. (2016) was based on a single set of estimates, to which a 50% uncertainty was associated as a range (60-180 Tg CH$_4$ yr$^{-1}$). Here the new estimates of DelSontro et al.

(2018) lead to a mean estimate of all inland freshwaters at 159 Tg CH$_4$ yr$^{-1}$ associated with the range 117-212 Tg CH$_4$ yr$^{-1}$ that reflects the minimum and maximum values of the available studies (see Methodology, Sect. 2). However, it should be noted that this range does not consider the uncertainty of individual studies. Importantly, these current estimates do not include the smallest size class of lakes or ephemeral streams resulting in a possible misallocation of freshwater fluxes to wetland ecosystems in spite of the attempts to

discount open water emissions from the wetland estimate (see above). The present data indicate that lakes or natural ponds, flooded land/reservoirs and streams/rivers account for 70%, 13% and 17% of the average inland water fluxes, respectively (given the large uncertainty, the percentages should be seen as approximate relative magnitudes only). The anthropogenic part of the inland water fluxes is best constrained for larger reservoirs, but remains less clear for other human-made flooded land. It should be noted that issues regarding

spatiotemporal variability are not considered in consistent ways at present (Natchimuthu et al., 2015; Wik et al., 2016a). Given the inconsistencies in the areal flux data and in area estimates, the aim to make frequent updates of the methane emissions is presently not possible for inland water emissions. Even more than for other emission categories, differences in inland water flux values used to estimate emissions, as well as how the data were processed, are more likely to represent differences between data, rather than reflecting real

temporal trends in the environment.

The improvement in quantifying inland water fluxes is highly dependent on the availability of more accurate assessments of their surface area. For streams and rivers, the 355,000 km$^2$ used in Bastviken et al. (2011) were re-evaluated to 540,000 km$^2$ by Stanley et al. (2016) due to new surface area estimate from Raymond et al. (2013). Regarding lakes and reservoirs, the three current inventories (Downing et al., 2006; Messager

et al., 2016; Verpoorter et al., 2014) show typical differences of a factor of 2 to 5 by size-class. Also, it was noted that small ponds, which were not included in either Downing et al. (2006) or Verpoorter et al. (2014), have a diffusive flux higher than any other size class of lakes (Holgerson and Raymond, 2016). Further analysis, and possibly more refined process-based models, are still necessary and urgent to evaluate these global up scaled estimates against regional specific approaches such as in Wik et al. (2016a) for the northern

high latitude lakes.

In this budget, we report a mean value of 159 Tg CH$_4$ yr$^{-1}$ from freshwater systems (lakes, ponds, reservoirs, streams and rivers), with a range of 117-212 Tg CH$_4$ yr$^{-1}$. This range shows the minimum and maximum estimates but excludes the uncertainty from each single estimate, which is expected to be large.

### 3.2.3 Onshore and offshore geological sources

Significant amounts of methane, produced within the Earth's crust, naturally migrate to the atmosphere through tectonic faults and fractured rocks. Major emissions are related to hydrocarbon production in sedimentary basins (microbial and thermogenic methane), through continuous or episodic exhalations from onshore and shallow marine hydrocarbon seeps and through diffuse soil microseepage (Etiope, 2015). Specifically, five source categories have been considered. Four are onshore sources: gas-oil seeps, mud

volcanoes, diffuse microseepage and geothermal manifestations including volcanoes. One source is offshore: submarine seepage, which may include the same types of gas manifestations occurring on land. Etiope et al. (2019) have produced the first gridded maps of geological methane emissions and their isotopic signature for these five categories, with a global total of 37.4 Tg CH$_4$ yr$^{-1}$ (reproduced in Fig. 4). According to them, the grid maps do not represent, however, the actual global geological-CH$_4$ emission because the datasets used

for the spatial gridding (developed for modelling purposes) were not complete or did not contain the information necessary for improving all previous estimates. Combining the best estimates for the five categories of geological sources (from grid maps or from previous statistical and process-based models), the breakdown by category reveals that onshore microseepage dominate (24 Tg CH$_4$ yr$^{-1}$), the other categories having similar smaller contributions: as average values, 4.7 Tg CH$_4$ yr$^{-1}$ for geothermal manifestations, about

7 Tg CH$_4$ yr$^{-1}$ for submarine seepage and 9.6 Tg CH$_4$ yr$^{-1}$ for onshore seeps and mud volcanoes. These values lead to a global bottom-up geological emission mean of 45 [27-63] Tg CH$_4$ yr$^{-1}$ (Etiope and Schwietzke, 2019).

    While all bottom-up and some top-down estimates, following different and independent techniques from different authors, consistently suggest a global geo-CH$_4$ emission in the order of 40-50 Tg yr-1, the

radiocarbon ($^{14}$C-CH$_4$) data in ice cores reported by Hmiel et al. (2020) appear to lower the estimate, with a minimum of about 1.6 Tg CH$_4$ yr$^{-1}$ and a maximum estimated value of 5.4 Tg CH$_4$ yr$^{-1}$ (95 percent confidence) for the pre-industrial period. The discrepancy between Hmiel et al. (2020) and all other estimates continue to feed the debate. Eastern Siberian Arctic Shelf (ESAS) emissions have been estimated at ~3 Tg CH$_4$ yr$^{-1}$ based on current atmospheric surface observations (Thornton et al., 2020), corresponding to the same order of

magnitude of the estimate from Hmiel et al. (2020) for global geological emissions. However, ESAS emissions are likely from both thermogenic and biogenic origins (e.g. Berchet et al., 2019). More investigation and confrontation between top-down and bottom-up results are needed to reduce this discrepancy.

Waiting for further investigation on this topic, we decided to keep the best estimates from Etiope and Shwietzke (2019) for the mean values, and associate it to the lowest estimates reported in Etiope et al. (2019). Thus, we report a total global geological emission of 45 [18-63] Tg $CH_4$ yr$^{-1}$, with a breakdown between offshore emissions of 7 [5-10] Tg $CH_4$ yr$^{-1}$ and onshore emissions of 38 [13-53] Tg $CH_4$ yr$^{-1}$. The updated bottom-up estimate is slightly lower than the previous budget mostly due to a reduction of estimated emissions of onshore and offshore seeps (see Sect. 3.2.6 for more offshore contribution explanations).

### 3.2.4 Termites

Termites are an infraorder of insects (isoptera), which occur predominantly in the tropical and subtropical latitudes (Abe et al., 2000). $CH_4$ is released during the anaerobic decomposition of plant biomass in their gut (Sanderson, 1996). The uncertainty related to this $CH_4$ source is very high as $CH_4$ emissions from termites in different ecosystem types can vary and are driven by a range of factors, while the number of field measurements, both of termite biomass and emissions, are relatively scarce (Kirschke et al., 2013).

In Kirschke et al. (2013) (see their supplementary material), a re-analysis of $CH_4$ emissions from termites at the global scale was proposed. Their $CH_4$ emissions per unit of area were estimated as the product of termite biomass, termite $CH_4$ emissions per unit of termite mass, and a scalar factor expressing the effect of land use/cover change, the latter two terms were estimated from published literature re-analysis. For tropical climates, termite biomass was estimated by a simple regression model representing its dependence on gross primary productivity (GPP), whereas for forest and grassland ecosystems of the warm temperate climates and for shrub lands of the Mediterranean sub-climate termite biomass was estimated from data reported by Sanderson (1996). The $CH_4$ emission factor per unit of termite biomass ($g_{termite}$) was estimated as 2.8 mg $CH_4$ ($g_{termite}$)$^{-1}$ h$^{-1}$ for tropical ecosystems and Mediterranean shrublands (Kirschke et al., 2013), and 1.7 mg $CH_4$ ($g_{termite}$)$^{-1}$ h$^{-1}$ for temperate forests and grasslands (Fraser et al., 1986). Emissions were scaled-up and annual $CH_4$ fluxes were computed for the three periods 1982-1989, 1990-1999 and 2000-2007 representative of the 1980s, 1990s and 2000s, respectively.

The re-analysis of termite emissions proposed in Saunois et al. (2016) maintained the same approach, but the data was calculated using climate zoning (following the Koppen-Geiger classification) applied to updated climate datasets by Santini and di Paola (2015), and was adapted to consider different combinations of termite biomass per unit area and $CH_4$ emission factor per unit of termite biomass.

Here, this analysis is extended to cover the periods 2000-2007 and 2010-2016. This latest estimate follows the approach outlined above for Saunois et al. (2016). However, in order to extend the analysis to 2016, an alternative MODIS based measure of GPP from Zhang at al. (2017), rather than from Jung et al. (2009), and Jung et al. (2011) was used to estimate termite biomass. To have coherent datasets of GPP and land use, the latter variable, previously derived from Ramankutty and Foley (1999), was substituted for MODIS maps

(Channan et al., 2014; Friedl et al., 2010). These new estimates covered 2000-2007 and 2010-2016 using 2002 and 2012 MODIS data as an average reference year for each period, respectively.

Termite $CH_4$ emissions show only little inter-annual and inter-decadal variability (0.1 Tg $CH_4$ yr$^{-1}$), whereas there is strong regional variability, with tropical South America and Africa being the main sources (23 and 28% of the total emissions, respectively) due to the extent of their natural forest and savannah ecosystems (Fig. 4). Changing the GPP and land use dataset sources had only a minimal impact on the 2000-2007 global termite flux, increasing it from 8.7 Tg $CH_4$ yr$^{-1}$ as found in the first two re-analyses (Kirschke et al., 2013; Saunois et al., 2016) to 9.9 Tg $CH_4$ yr$^{-1}$ (present data), well within the estimated uncertainty (8.7±3.1 Tg $CH_4$ yr$^{-1}$). However, it had a noticeable effect on the spatial distribution of the flux (Fig. S2). The most obvious of these changes is a halving of the Southeast Asian flux, aligned with shifts in the underlying GPP product. Previous studies (Mercado et al., 2009; Zhang et al., 2017) had linked these GPP shifts to a methodological issue with light-use efficiency that drove an underestimation of evergreen broadleaf and evergreen needleleaf forest GPP, biomes which are prevalent in the tropics. This value is close to the average estimate derived from previous up-scaling studies, which report values spanning from 2 to 22 Tg $CH_4$ yr$^{-1}$ (Ciais et al., 2013). In this study, we report a decadal value of 9 Tg $CH_4$ yr$^{-1}$ (range [3-15] Tg $CH_4$ yr$^{-1}$, Table 3).

### 3.2.5 Wild animals

Wild ruminants emit methane through the microbial fermentation process occurring in their rumen, similarly to domesticated livestock species (USEPA, 2010b). Using a total animal population of 100-500 million, Crutzen et al. (1986) estimated the global emissions of $CH_4$ from wild ruminants to be in the range of 2-6 Tg $CH_4$ yr$^{-1}$. More recently, Pérez-Barbería (2017) lowered this estimate to 1.1-2.7 Tg $CH_4$ yr$^{-1}$ using a total animal population estimate of 214 million (range of 210-219), arguing that the maximum number of animals (500 million) used in Crutzen et al. (1986) was poorly justified. Moreover Pérez-Barbería (2017) also stated that the value of 15 Tg $CH_4$ yr$^{-1}$ found in the last IPCC reports is much higher than their estimate because this value comes from an extrapolation of Crutzen's work for the last glacial maximum when the population of wild animals was much larger, as originally proposed by Chappellaz et al. (1993).

Based on these findings, the range adopted in this updated methane budget is 2 [1-3] Tg $CH_4$ yr$^{-1}$ (Table 3).

### 3.2.6 Oceanic sources

Oceanic sources comprise coastal ocean and open ocean methane release. Possible sources of oceanic $CH_4$ include: (1) production from marine (bare and vegetated) sediments or thawing sub-sea permafrost; (2) in situ production in the water column, especially in the coastal ocean because of submarine groundwater discharge (USEPA, 2010b); (3) leaks from geological marine seepage (see also Sect. 3.2.3); and (4) emission from the destabilisation of marine hydrates. Once at the seabed, methane can be transported through the water

column by diffusion in a dissolved form (especially in the upwelling zones), or by ebullition (gas bubbles, e.g. from geological marine seeps), for instance, in shallow waters of continental shelves. In coastal vegetated habitats methane can also be transported to the atmosphere through the *aerenchyma* of emergent aquatic plants (Ramachandran et al., 2004).

**Biogenic emissions from open and coastal ocean.** The most common biogenic ocean emission value found in the literature is 10 Tg $CH_4$ $yr^{-1}$ (Rhee et al., 2009b). It appears that most studies rely on the work of Ehhalt (1974), where the value was estimated on the basis of the measurements done by Swinnerton and co-workers (Lamontagne et al., 1973; Swinnerton and Linnenbom, 1967) for the open ocean, combined with purely speculated emissions from the continental shelf. Based on basin-wide observations using updated methodologies, three studies found estimates ranging from 0.2 to 3 Tg $CH_4$ $yr^{-1}$ (Bates et al., 1996; Conrad and Seiler, 1988; Rhee et al., 2009b), associated with super-saturations of surface waters that are an order of magnitude smaller than previously estimated, both for the open ocean (saturation anomaly ~0.04, see Rhee et al.(2009a), equation 4) and for continental shelf (saturation anomaly ~0.2). In their synthesis, indirectly referring to the original observations from Lambert and Schmidt (1993), Wuebbles and Hayhoe (2002), they use a value of 5 Tg $CH_4$ $yr^{-1}$. Proposed explanations for discrepancies regarding sea-to air methane emissions in the open ocean rely on experimental biases in the former studies of Swinnerton and Linnenbom (Rhee et al., 2009b). This may explain why the Bange et al. (1994) compilation cites a global source of 11-18 Tg $CH_4$ $yr^{-1}$ with a dominant contribution of coastal regions. Here, we report a range of 0-5 Tg $CH_4$ $yr^{-1}$, with a mean value of 2 Tg $CH_4$ $yr^{-1}$ for biogenic emissions from open and coastal ocean (excluding estuaries).

Biogenic emissions from brackish waters (estuaries, coastal wetlands) were not reported in the previous budget (Saunois et al., 2016). Methane emissions from estuaries were originally estimated by Bange et al. (1994), Upstill-Goddard et al. (2000) and Middelburg et al. (2002) to be comprised between 1 and -3 Tg $CH_4$ $yr^{-1}$. This range was later revised upwards by Borges and Abril (2011) to about 7 Tg $CH_4$ $yr^{-1}$ based on a methodology distinguishing between different estuarine types and accounting for the contribution of tidal flats, marshes and mangroves, for a total of 39 systems and a global "inner" estuarine surface area of 1.1 $10^6$ $km^2$ (Laruelle et al., 2013). The same methodology as in Laruelle et al. (2013) has been applied here to the same systems using an expanded database of local and regional measurements (72 systems) and suggests however that global estuarine $CH_4$ emissions were overestimated and may actually not surpass 3-3.5 Tg $CH_4$ $yr^{-1}$. Despite this overall reduction, the specific contribution of sediment and water emissions from mangrove ecosystems is however higher and contributes <0.1 to 1.7 Tg $CH_4$ $yr^{-1}$ globally (Rosentreter et al., 2018). This estuarine estimate does not include the uncertain contribution from large river plumes protruding onto the shelves. Their surface area reaches about 3.7 $10^6$ $km^2$ (Kang et al., 2013) but because of significantly lower $CH_4$ concentration (e.g. Osudar et al., 2015; Zhang et al., 2008) than in inner estuaries, the outgassing associated with these plumes likely does not exceed 1-2 Tg $CH_4$ $yr^{-1}$. Seagrass meadows are also not included

although they might release <0.1 to 2.7 Tg $CH_4$ yr$^{-1}$ (Garcias-Bonet and Duarte, 2017). These methane emissions from vegetated coastal ecosystems can partially offset (Rosentreter et al., 2018) their "blue carbon" sink (e.g. Mcleod et al., 2011; Nellemann et al., 2009). Note that the latter two contributions might partly overlap with oceanic (open and coastal) sources estimates. The total (inner and outer) estuarine emission flux, which is based on only about 80 systems is thus in the range 4-5 Tg $CH_4$ yr$^{-1}$ (including marshes and mangrove). High uncertainties in coastal ocean emission estimates can be reduced by better defining the various coastal ecosystem types and their boundaries to avoid double-counting (e.g. estuaries, brackish wetlands, freshwater wetlands), updating the surface area of each of these coastal systems, and better quantifying methane emission rates in each ecosystem type.

As a result, here we report a range of 4-10 Tg $CH_4$ yr$^{-1}$ for emissions from coastal and open ocean (including estuaries), with a mean value of 6 Tg $CH_4$ yr$^{-1}$.

**Geological emissions.** The production of methane at the seabed is known to be significant. For instance, marine seepages emit up to 65 Tg $CH_4$ yr$^{-1}$ globally at seabed level (USEPA, 2010b). What is uncertain is the flux of oceanic methane reaching the atmosphere. For example, bubble plumes of $CH_4$ from the seabed have been observed in the water column, but not detected in the Arctic atmosphere (Fisher et al., 2011; Westbrook et al., 2009). There are several barriers preventing methane to be expelled to the atmosphere (James et al., 2016). From below the seafloor to the seas surface, gas hydrates and permafrost serve as a barrier to fluid and gas migration towards the seafloor, microbial activity around the seafloor can strongly oxidise methane releases or production, further oxidation occurs in the water column, the oceanic pycnocline acts as a physical barrier towards the surface waters, including efficient dissolution of bubbles, and finally, surface oceans are aerobic and contribute to the oxidation of dissolved methane. However, surface waters can be more supersaturated than the underlying deeper waters, leading to a methane paradox (Sasakawa et al., 2008). Possible explanations involve i) upwelling in areas with surface mixed layers covered by sea-ice (Damm et al., 2015), ii) the release of methane by the degradation of dissolved organic matter phosphonates in aerobic conditions (Repeta et al., 2016), iii) methane production by marine algae (Lenhart et al., 2016), or iv) methane production within the anoxic centre of sinking particles (Sasakawa et al., 2008), but more work is still needed to be conclusive about this apparent paradox.

For geological emissions, the most used value has long been 20 Tg $CH_4$ yr$^{-1}$, relying on expert knowledge and literature synthesis proposed in a workshop reported in Kvenvolden et al. (2001), the author of this study recognising that this was a first estimation and needs revision. Since then, oceanographic campaigns have been organized, especially to sample bubbling areas of active seafloor gas seep bubbling. For instance, Shakhova et al. (2010, 2014) infers 8-17 Tg $CH_4$ yr$^{-1}$ in emissions just for the Eastern Siberian Arctic Shelf (ESAS), based on the extrapolation of numerous but local measurements, and possibly related to thawing subseabed permafrost (Shakhova et al., 2015). Because of the highly heterogeneous distribution of dissolved

CH4 in coastal regions, where bubbles can most easily reach the atmosphere, extrapolation of in situ local measurements to the global scale can be hazardous and lead to biased global estimates. Indeed, using very precise and accurate continuous land shore-based atmospheric methane observations in the Arctic region, Berchet et al. (2016) found a range of emissions for ESAS of ~2.5 Tg CH4 yr$^{-1}$ (range [0-5]), 4-8 times lower than Shakhova's estimates. Such a reduction in ESAS emission estimate has also been inferred from oceanic observations by Thornton et al. (2016b) with a maximum sea-air CH4 flux of 2.9 Tg CH4 yr$^{-1}$ for this region. Etiope et al. (2019) suggested a minimum global total submarine seepage emission of 3.9 Tg CH4 yr$^{-1}$ simply summing published regional emission estimates for 15 areas for identified emission areas (above 7 Tg CH4 yr$^{-1}$ when extrapolated to include non-measured areas). These recent results, based on different approaches, suggest that the current estimate of 20 Tg CH4 yr$^{-1}$ is too large and needs revision.

Therefore, as discussed in Section 3.2.2, we report here a reduced range of 5-10 Tg CH4 yr$^{-1}$ for marine geological emissions compared to the previous budget, with a mean value of 7 Tg CH4 yr$^{-1}$.

**Hydrate emissions.** Among the different origins of oceanic methane, hydrates have attracted a lot of attention. Methane hydrates (or clathrates) are ice-like crystals formed under specific temperature and pressure conditions (Milkov, 2005). Methane hydrates can be either of biogenic origin (formed in situ at depth in the sediment by microbial activity) or of thermogenic origin (non-biogenic gas migrated from deeper sediments and trapped due to pressure/temperature conditions or due to some capping geological structure such as marine permafrost). The total stock of marine methane hydrates is large but uncertain, with global estimates ranging from hundreds to thousands of Pg CH4 (Klauda and Sandler, 2005; Wallmann et al., 2012). Concerning more specifically atmospheric emissions from marine hydrates, Etiope (2015) points out that current estimates of methane air-sea flux from hydrates (2-10 Tg CH4 yr$^{-1}$ in e.g. Ciais et al. (2013) or Kirschke et al. (2013)) originate from the hypothetical values of Cicerone and Oremland (1988). No experimental data or estimation procedures have been explicitly described along the chain of references since then (Denman et al., 2007; IPCC, 2001; Kirschke et al., 2013; Lelieveld et al., 1998). It was estimated that ~473 Tg CH4 have been released in the water column over 100 years (Kretschmer et al., 2015). Those few Tg per year become negligible once consumption in the water column has been accounted for. While events such as submarine slumps may trigger local releases of considerable amounts of methane from hydrates that may reach the atmosphere (Etiope, 2015; Paull et al., 2002), on a global scale, present-day atmospheric methane emissions from hydrates do not appear to be a significant source to the atmosphere, and at least formally, we should consider 0 (<0.1) Tg CH4 yr$^{-1}$ emissions.

**Combination (biogenic and geological) of open and coastal oceanic emissions.** Summing biogenic, geological and hydrate emissions from open and coastal ocean (excluding estuaries) leads to a total of 9 Tg CH4 yr-1 (range 5-17). A recent work (Weber et al., 2019) suggests a new robust estimate of the climatological oceanic flux: the diffusive flux was estimated as 2-6 Tg CH4 yr$^{-1}$ and the ebullitive flux as 2-

11 Tg CH$_4$ yr$^{-1}$, giving a total (open and coastal) oceanic flux estimate of 6-15 Tg CH$_4$ yr$^{-1}$ (90% confidence interval) when the probability distributions for the two pathways are combined. Distribution of open and coastal oceanic fluxes from Weber et al. (2019) is shown in Fig. 4. This more robust estimate took benefit from synthesis of in situ measurements of atmospheric and surface water methane concentrations and of bubbling areas, and of the development of process-based models for oceanic methane emissions. Another recent estimate based on the biogeochemistry model PlankTOM10 (Le Quéré et al., 2016) calculates an open and coastal ocean methane flux (excluding estuaries) of 8 [-13/ +19] Tg CH$_4$ yr$^{-1}$ (Buitenhuis et al., 2019), with a coastal contribution of 44%. Our estimate (9 [5-17] Tg CH$_4$ yr$^{-1}$) agrees well with the estimates of 6-15 Tg CH$_4$ yr$^{-1}$ by Weber et al. (2019) and 8 Tg CH$_4$ yr$^{-1}$ (Buitenhuis et al., 2019).

Methane emissions from brackish water were not estimated in Saunois et al. (2016) and additional 4 Tg CH$_4$ yr$^{-1}$ are reported in this budget. As a result, including estuaries in the oceanic budget, we report a range of 9-22 Tg CH$_4$ yr$^{-1}$, with a mean value of 13 Tg CH$_4$ yr$^{-1}$, leading to similar total oceanic emissions despite a reduced estimate in geological off shore emissions compared to Saunois et al. (2016).

### 3.2.7 Terrestrial permafrost and hydrates

Permafrost is defined as frozen soil, sediment, or rock having temperatures at or below 0°C for at least two consecutive years (Harris et al., 1988). The total extent of permafrost in the Northern Hemisphere is about 14 million km$^2$ or 15% of the exposed land surface (Obu et al., 2019). As the climate warms, large areas of permafrost are also warming, and if soil temperatures pass 0°C, thawing of the permafrost occurs. Permafrost thaw is most pronounced in southern and spatially isolated permafrost zones, but also occurs in northern continuous permafrost (Obu et al., 2019). Thaw occurs either as a gradual, often widespread, deepening of the active layer or as more rapid localised thaw associated to loss of massive ground ice (thermokarst) (Schuur et al., 2015). A total of 1035 ± 150 Pg of carbon can be found in the upper 3 meters of permafrost regions, or ~1300 (1100 to 1500) Pg C for all permafrost (Hugelius et al., 2014).

The thawing permafrost can generate direct and indirect methane emissions. Direct methane emissions rely on the release of methane contained in the thawing permafrost. This flux to the atmosphere is small and estimated to be a maximum of 1 Tg CH$_4$ yr$^{-1}$ at present (USEPA, 2010b). Indirect methane emissions are probably more important. They rely on: 1) methanogenesis induced when the organic matter contained in thawing permafrost is released; 2) the associated changes in land surface hydrology possibly enhancing methane production (McCalley et al., 2014); and 3) the formation of more thermokarst lakes from erosion and soil collapse. Such methane production is probably already significant today and is likely to become more important in the future associated with climate change and strong positive feedback from thawing permafrost (Schuur et al., 2015). However, indirect methane emissions from permafrost thawing are difficult to estimate at present, with very few data to refer to, and in any case largely overlap with wetland and

freshwater emissions occurring above or around thawing areas. For instance, based on lake and soil measurements(Walter Anthony et al., 2016) found that methane emissions ($\sim$4 Tg $CH_4$ $yr^{-1}$) from thermokarst lakes that have expanded over the past 60 years were directly proportional to the mass of soil carbon inputs to the lakes from the erosion of thawing permafrost.

Here, we choose to report only the direct emission range of 0-1 Tg $CH_4$ $yr^{-1}$, keeping in mind that current wetland, thermokarst lakes and other freshwater methane emissions already likely include a significant indirect contribution originating from thawing permafrost.

### 3.2.8 Vegetation

Three distinct pathways for the production and emission of methane by living vegetation are considered here (see Covey and Megonigal (2019) for an extensive review). Firstly, plants produce methane through an abiotic photochemical process induced by stress (Keppler et al., 2006). This pathway was initially criticized (e.g. Dueck et al., 2007; Nisbet et al., 2009), and although numerous studies have since confirmed aerobic emissions from plants and better resolved its physical drivers (Fraser et al., 2015), global estimates still vary by two orders of magnitude (Liu et al., 2015). This plant source has not been confirmed in-field however, and although the potential implication for the global methane budget remains unclear, emissions from this source are certainly much smaller than originally estimated in Keppler et al. (2006) (Bloom et al., 2010; Fraser et al., 2015). Second, and of clearer significance, plants act as "straws", drawing up and releasing microbially produced methane from anoxic soils (Cicerone and Shetter, 1981; Rice et al., 2010). For instance, in the forested wetlands of Amazonia, tree stems are the dominant ecosystem flux pathway for soil-produced methane, therefore, including stem emissions in ecosystem budgets can reconcile regional bottom-up and top-down estimates (Pangala et al., 2017). Third, the stems of both living trees (Covey et al., 2012) and dead wood (Covey et al., 2016) provide an environment suitable for microbial methanogenesis. Static chambers demonstrate locally significant through-bark flux from both soil- (Pangala et al., 2013, 2015), and tree stem-based methanogens (Pitz and Megonigal, 2017; Wang et al., 2016). A recent synthesis indicates stem $CH_4$ emissions significantly increase the source strength of forested wetlands, and modestly decrease the sink strength of upland forests (Covey and Megonigal, 2019). The scientific activity covering $CH_4$ emissions in forested ecosystems reveals a far more complex story than previously thought, with an interplay of, productive/consumptive, aerobic/anaerobic, biotic/abiotic, processes occurring between upland/wetland soils, trees, and atmosphere. Understanding the complex processes that regulate $CH_4$ source–sink dynamics in forests and estimating their contribution to the global methane budget requires cross-disciplinary research, more observations, and new models that can overcome the classical binary classifications of wetland versus upland forest and of emitting versus uptaking soils (Barba et al., 2019; Covey and Megonigal, 2019). Although we recognize these emissions are potentially large (particularly tree transport from inundated soil),

global estimates for each of these pathways remain highly uncertain and/or are currently ascribed here to other flux categories sources (e/g. inland waters, wetlands, upland soils).

### 3.3 Methane sinks and lifetime

Methane is the most abundant reactive trace gas in the troposphere and its reactivity is important to both tropospheric and stratospheric chemistry. The main atmospheric sink of methane (~90% of the total sink mechanism) is oxidation by the hydroxyl radical (OH), mostly in the troposphere (Ehhalt, 1974). Other losses are by photochemistry in the stratosphere (reactions with chlorine atoms (Cl) and excited atomic oxygen $(O(^1D))$, oxidation in soils (Curry, 2007; Dutaur and Verchot, 2007), and by photochemistry in the marine

boundary layer (reaction with Cl; Allan et al. (2007), Thornton et al. (2010)). Uncertainties in the total sink of methane as estimated by atmospheric chemistry models are in the order of 20-40% (Saunois et al., 2016). It is much less (10-20%) when using atmospheric proxy methods (e.g. methyl chloroform, see below) as in atmospheric inversions (Saunois et al., 2016). In the present release of the global methane budget, we estimate bottom-up methane chemical sinks and lifetime mainly based on global model results from the Chemistry

Climate Model Initiative (CCMI) (Morgenstern et al., 2017).

#### 3.3.1 Tropospheric OH oxidation

OH radicals are produced following the photolysis of ozone $(O_3)$ in the presence of water vapour. OH is destroyed by reactions with CO, $CH_4$, and non-methane volatile organic compounds.

Following the Atmospheric Chemistry and Climate Model Intercomparison Project (ACCMIP), which

studied the long-term changes in atmospheric composition between 1850 and 2100 (Lamarque et al., 2013), a new series of experiments was conducted by several chemistry-climate models and chemistry-transport models participating in the Chemistry-Climate Model Initiative (CCMI) (Morgenstern et al., 2017). Mass-weighted OH tropospheric concentrations do not directly represent methane loss, as the spatial and vertical distributions of OH affect this loss, through, in particular, the temperature dependency and the distribution

of methane (e.g. Zhao et al., 2019). However, estimating OH concentrations and, spatial and vertical distributions is a key step in estimating methane loss through OH. Over the period 2000-2010, the multi-model mean (11 models) global mass-weighted OH tropospheric concentration was $11.7\pm1.0 \times 10^5$ molecules $cm^{-3}$ (range 9.9-14.4 x $10^5$ molecules $cm^{-3}$, Zhao et al. (2019)) consistent with the previous estimates from ACCMIP ($11.7\pm1.0 \times 10^5$ molecules $cm^{-3}$, with a range of 10.3-13.4 x $10^5$ molecules $cm^{-3}$, Voulgarakis et al.

(2013) for year 2000) and the estimates of Prather et al. (2012) at $11.2\pm1.3 \times 10^5$ molecules $cm^{-3}$. Nicely et al. (2017) attribute the differences in OH simulated by different chemistry transport models to, in decreasing order of importance, different chemical mechanisms, various treatment of the photolysis rate of ozone, and modeled ozone and carbon monoxide. Besides the uncertainty on global OH concentrations, there is an

uncertainty in the spatial and temporal distribution of OH. Models often simulate higher OH in the northern hemisphere leading to a NH/SH OH ratio greater than 1 (Naik et al., 2013; Zhao et al., 2019). However, there is evidence for parity in inter-hemispheric OH concentrations (Patra et al., 2014), which needs to be confirmed by other observational and model-derived estimates.

OH concentrations and their changes can be sensitive to climate variability (Dlugokencky et al., 1996; Holmes et al., 2013; Turner et al., 2018), biomass burning (Voulgarakis et al., 2015), and anthropogenic activities. For instance, the increase of the oxidizing capacity of the troposphere in South and East Asia associated with increasing $NO_X$ emissions (Mijling et al., 2013) and decreasing CO emissions (Yin et al., 2015), possibly enhances $CH_4$ oxidation and therefore limits the atmospheric impact of increasing emissions (Dalsøren et al., 2009). Despite such large regional changes, the global mean OH concentration was suggested to have changed only slightly over the past 150 years (Naik et al., 2013). This is due to the compensating effects of the concurrent increases of positive influences on OH (water vapour, tropospheric ozone, nitrogen oxides ($NO_x$) emissions, and UV radiation due to decreasing stratospheric ozone), and of OH sinks (methane burden, carbon monoxide and non-methane volatile organic compound emissions and burden). CCMI models show OH inter-annual variability ranging from 0.4% to 1.8% (Zhao et al., 2019) over 2000-2010, lower than the value deduced from methyl chloroform measurements (proxy, top-down approach). However, these simulations consider meteorology variability but not emission interannual variability (e.g., from biomass burning) and thus are expected to simulate lower OH inter annual variability than in reality. Using an empirical model constrained by global observations of ozone, water vapor, methane, and temperature as well as the simulated effects of changing $NO_x$ emissions and tropical expansion, Nicely et al. (2017) found an inter-annual variability in OH of about 1.3-1.6% between 1980 and 2015, in agreement with methyl chloroform proxy (Montzka et al., 2011).

We report here a climatological range for the tropospheric loss of methane by OH oxidation of 553 [476-677] Tg $CH_4$ $yr^{-1}$ derived from the seven models that contributed to CCMI for the total tropospheric loss of methane by OH oxidation over the period 2000-2009 (tropopause height at 200 hPa), which is slightly higher than the one from the ACCMIP models (528 [454-617] Tg $CH_4$ $yr^{-1}$ reported in Kirschke et al. (2013) and Saunois et al. (2016).

### 3.3.2 Stratospheric loss

In the stratosphere, $CH_4$ is lost through reactions with excited atomic oxygen O($^1$D), atomic chlorine (Cl), atomic fluorine (F), and OH (Brasseur and Solomon, 2005; le Texier et al., 1988). Uncertainties in the chemical loss of stratospheric methane are large, due to uncertain inter-annual variability in stratospheric transport as well as its chemical interactions and feedbacks with stratospheric ozone (Portmann et al., 2012). Particularly, the fraction of stratospheric loss due to the different oxidants is still uncertain, with possibly 20-

35% due to halons, about 25% due to O($^1$D) mostly in the high stratosphere and the rest due to stratospheric OH (McCarthy et al., 2003).

In this study, seven chemistry climate models from the CCMI project (Table S4) are used to provide estimates of methane chemical loss, including reactions with OH, O($^1$D), and Cl; CH$_4$ photolysis is also included but occurs only above the stratosphere. Considering a 200 hPa tropopause height, the CCMI models suggest an estimate of 31 [12-37] Tg CH$_4$ yr$^{-1}$ for the methane stratospheric sink for the period 2000-2010 (Table S4). The 20 Tg difference compared to the mean value reported by Kirschke et al. (2013) and Saunois et al. (2016) for the same period (51 [16-84] Tg CH$_4$ yr$^{-1}$), is probably due to the plausible double-counting of O($^1$D) and Cl oxidations in our previous calculation, as the chemistry-climate models usually report the total chemical loss of methane (not OH oxidation only).

We report here a climatological range of 12-37 Tg CH$_4$ yr$^{-1}$ associated to a mean value of 31 Tg CH$_4$ yr$^{-1}$.

### 3.3.3 Tropospheric reaction with Cl

Halogen atoms can also contribute to the oxidation of methane in the troposphere. Allan et al. (2005) measured mixing ratios of methane and δ$^{13}$C-CH$_4$ at two stations in the southern hemisphere from 1991 to 2003, and found that the apparent kinetic isotope effect (KIE) of the atmospheric methane sink was significantly larger than that explained by OH alone. A seasonally varying sink due to atomic chlorine (Cl) in the marine boundary layer of between 13 and 37 Tg CH$_4$ yr$^{-1}$ was proposed as the explanatory mechanism (Allan et al., 2007; Platt et al., 2004). This sink was estimated to occur mainly over coastal and marine regions, where NaCl from evaporated droplets of seawater react with NO$_2$ to eventually form Cl$_2$, which then UV-dissociates to Cl. However significant production of nitryl chloride (ClNO$_2$) at continental sites has been recently reported (Riedel et al., 2014) and suggests the broader presence of Cl, which in turn would expand the significance of the Cl sink in the troposphere. Recently, using a chemistry transport model, Hossaini et al. (2016) suggest a chlorine sink in the lower range of Allan et al. (2007), ~12-13 Tg CH$_4$ yr$^{-1}$ (about 2.5 % of the tropospheric sink). They also estimate that ClNO$_2$ yields a 1 Tg yr$^{-1}$ sink of methane. Another modelling study (Wang et al., 2019b) produced a more comprehensive analysis of global tropospheric chlorine chemistry and found a chlorine sink of 5 Tg yr$^{-1}$, representing only 1% of the total methane tropospheric sink. Both the KIE approach and chemistry transport model simulations carry uncertainties (extrapolations based on only a few sites and use of indirect measurements, for the former; missing sources, coarse resolution, underestimation of some anthropogenic sources for the latter). However, Gromov et al. (2018) found that chlorine can contribute only 0.23% the tropospheric sink of methane (about 1 Tg CH$_4$ yr$^{-1}$) in order to balance the global $^{13}$C(CO) budget.

Awaiting further work to better assess the magnitude of the chlorine sink in the methane budget, we suggest a lower estimate but a larger range than in Saunois et al. (2016) and used the following climatological value for the 2000s: 11 [1-35] Tg $CH_4$ $yr^{-1}$.

### 3.3.4 Soil uptake

Unsaturated oxic soils are sinks of atmospheric methane due to the presence of methanotrophic bacteria, which consume methane as a source of energy. Dutaur and Verchot (2007) conducted a comprehensive meta-analysis of field measurements of $CH_4$ uptake spanning a variety of ecosystems. Extrapolating to the global scale, they reported a range of $36 \pm 23$ Tg $CH_4$ $yr^{-1}$, but also showed that stratifying the results by climatic zone, ecosystem and soil type led to a narrower range (and lower mean estimate) of $22 \pm 12$ Tg $CH_4$ $yr^{-1}$. Modelling studies, employing meteorological data as external forcing, have also produced a considerable range of estimates. Using a soil depth-averaged formulation based on Fick's law with parameterizations for diffusion and biological oxidation of $CH_4$, Ridgwell et al. (1999) estimated the global sink strength at 38 Tg $CH_4$ $yr^{-1}$, with a range 20-51 Tg $CH_4$ $yr^{-1}$ reflecting the model structural uncertainty in the base oxidation parameter. Curry (2007) improved on the latter by employing an exact solution of the one-dimensional diffusion-reaction equation in the near-surface soil layer (i.e., exponential decrease in $CH_4$ concentration below the surface), a land surface hydrology model, and calibration of the oxidation rate to field measurements. This resulted in a global estimate of 28 Tg $CH_4$ $yr^{-1}$ (9-47 Tg $CH_4$ $yr^{-1}$), the result reported by Zhuang et al. (2013), Kirschke et al. (2013) and Saunois et al. (2016). Ito and Inatomi (2012) used an ensemble methodology to explore the variation in estimates produced by these parameterizations and others, which spanned the range 25-35 Tg $CH_4$ $yr^{-1}$. Murguia-Flores et al. (2018) further refined the Curry (2007) model's structural and parametric representations of key drivers of soil methanotrophy, demonstrating good agreement with the observed latitudinal distribution of soil uptake (Dutaur and Verchot, 2007). Their model simulated a methane soil sink of 32 Tg $CH_4$ $yr^{-1}$ for the period 2000-2017 (Fig. 4), compared to 38 and 29 Tg $CH_4$ $yr^{-1}$ using the Ridgwell et al. (1999) and Curry (2007) parameterizations, respectively, under the same meteorological forcing. As part of a more comprehensive model accounting for a range of methane sources and sinks, Tian et al. (2010, 2015, 2016) computed vertically-averaged $CH_4$ soil uptake including the additional mechanisms of aqueous diffusion and plant-mediated (*aerenchyma*) transport, arriving at the estimate $30 \pm 19$ Tg $CH_4$ $yr^{-1}$ (Tian et al., 2016). The still more comprehensive biogeochemical model of Riley et al. (2011) included vertically resolved representations of the same processes considered by Tian et al. (2016), in addition to grid cell fractional inundation and, importantly, the joint limitation of uptake by both $CH_4$ and $O_2$ availability in the soil column. Riley et al. (2011) estimated a global $CH_4$ soil sink of 31 Tg $CH_4$ $yr^{-1}$ with a structural uncertainty of 15-38 Tg $CH_4$ $yr^{-1}$ (a higher upper limit resulted from an elevated gas diffusivity to mimic convective transport; as this is not usually considered, we adopt the lower upper bound

associated with no limitation of uptake at low soil moisture). A model of this degree of complexity is required to explicitly simulate situations where the soil water content increases enough to inhibit the diffusion of oxygen, and the soil becomes a methane source (Lohila et al., 2016). This transition can be rapid, thus creating areas (for example, seasonal wetlands) that can be either a source or a sink of methane depending on the season.

The previous Curry (2007) estimate can be revised upward based on subsequent work and the increase in $CH_4$ concentration since that time, which gives a central estimate of 30.1 Tg $CH_4$ $yr^{-1}$. Considering structural uncertainty in the various models' assumptions and parameters, we report here the median and range of Tian et al. (2016): 30 [11-49] Tg $CH_4$ $yr^{-1}$ for the periods 2000-2009 and 2008-2017.

### 3.3.5 $CH_4$ lifetime

The atmospheric lifetime of a given gas in steady state may be defined as the global atmospheric burden (Tg) divided by the total sink (Tg/yr) (IPCC, 2001). Global models provide an estimate of the loss of the gas due to individual sinks, which can then be used to derive lifetime due to a specific sink. For example, methane's tropospheric lifetime is determined as global atmospheric methane burden divided by the loss from OH oxidation in the troposphere, sometimes called "chemical lifetime". Methane total lifetime corresponds to the global burden divided by the total loss including tropospheric loss from OH oxidation, stratospheric chemistry and soil uptake. The CCMI models (described in Morgenstein et al. (2017)) estimate the tropospheric methane lifetime at about 9 years (average over years 2000-2009), with a range of 7.2-10.1 years (see Table S4). While this range agrees with previous values found in ACCMIP (9.3 [7.1-10.6] years, Voulgarakis et al. (2013)), the mean value reported here is lower than previously reported, probably due to a smaller and different ensemble of climate models. Adding 30 Tg to account for the soil uptake to the total chemical loss of the CCMI models, we derive a total methane lifetime of 7.8 years (average over 2000-2009 with a range of 6.5-8.8 years). These updated model estimates of total methane lifetime agree with the previous estimates from ACCMIP (8.2 [6.4-9.2] years for year 2000, Voulgarakis et al. (2013)). Reducing the large spread in methane lifetime (between models, and between models and observation-based estimates) would 1) bring an improved constraint on global total methane emissions, and 2) ensure an accurate forecast of future climate.

### 4 Atmospheric observations and top-down inversions

### 4.1 Atmospheric observations

Systematic atmospheric $CH_4$ observations began in 1978 (Blake et al., 1982) with infrequent measurements from discrete air samples collected in the Pacific at a range of latitudes from 67°N to 53°S. Because most of

these air samples were from well-mixed oceanic air masses and the measurement technique was precise and accurate, they were sufficient to establish an increasing trend and the first indication of the latitudinal gradient of methane. Spatial and temporal coverage was greatly improved soon after (Blake and Rowland, 1986) with the addition of the Earth System Research Laboratory from US National Oceanic and Atmospheric Administration (NOAA/ESRL) flask network (Steele et al. (1987), Fig. 1), and of the Advanced Global Atmospheric Gases Experiment (AGAGE) (Cunnold et al., 2002; Prinn et al., 2000), the Commonwealth Scientific and Industrial Research Organisation (CSIRO, Francey et al. (1999)), the University of California Irvine (UCI, Simpson et al. (2012)) and in situ and flask measurements from regional networks, such as ICOS (Integrated Carbon Observation System) network in Europe (https://www.icos-ri.eu/). The combined datasets provide the longest time series of globally averaged $CH_4$ abundance. Since the early-2000s, $CH_4$ column-averaged mole fractions have been retrieved through passive remote sensing from space (Buchwitz et al., 2005a, 2005b; Butz et al., 2011; Crevoisier et al., 2009; Frankenberg et al., 2005; Hu et al., 2018). Ground-based Fourier transform infrared (FTIR) measurements at fixed locations also provide time-resolved methane column observations during daylight hours, and a validation dataset against which to evaluate the satellite measurements such as TCCON network (e.g. Pollard et al., 2017; Wunch et al., 2011), or Network for Detection of Atmospheric Composition Change (NDACC) (e.g. Bader et al., 2017).

In this budget, in-situ observations from the different networks were used in the top-down atmospheric inversions to estimate methane sources and sinks over the period 2000-2017. Satellite observations from TANSO/FTS instrument on board the satellite GOSAT were used to estimate methane sources and sinks over the period 2009-2017. Other atmospheric data (FTIR, airborne measurements, AirCore, isotopic measurements…) have been used for validation by some groups, but not specifically in this study. However, further information is provided in the Supplementary Material and a more comprehensive validation of the inversions is planned to use some of these data.

### 4.1.1 In situ $CH_4$ observations and atmospheric growth rate at the surface

We use globally averaged $CH_4$ mole fractions at the Earth's surface from the four observational networks (NOAA/ESRL, AGAGE, CSIRO and UCI). The data are archived at the World Data Centre for Greenhouse Gases (WDCGG) of the WMO Global Atmospheric Watch (WMO-GAW) program, including measurements from other sites that are not operated as part of the four networks. The $CH_4$ in-situ monitoring network has grown significantly over the last decade due to the emergence of laser diode spectrometers which are robust and accurate enough to allow deployments with minimal maintenance enabling the development of denser networks in developed countries (Stanley et al., 2018; Yver Kwok et al., 2015), and new stations in remote environment (Bian et al., 2015; Nisbet et al., 2019).

The networks differ in their sampling strategies, including the frequency of observations, spatial distribution, and methods of calculating globally averaged $CH_4$ mole fractions. Details are given in the supplementary material of Kirschke et al. (2013). The global average values of $CH_4$ concentrations presented in Fig. 1 are computed using long-time series measurements through gas chromatography with flame ionization detection (GC/FID), although chromatographic schemes vary among the labs. Because GC/FID is a relative measurement method, the instrument response must be calibrated against standards. The current WMO reference scale, maintained by NOAA/ESRL, WMO-X2004A (Dlugokencky et al., 2005) was updated in July 2015. NOAA and CSIRO global means are on this scale. AGAGE uses an independent standard scale maintained by Tohoku University (Aoki et al., 1992), but direct comparisons of standards and indirect comparisons of atmospheric measurements show that differences are below 5 ppb (Tans and Zwellberg, 2014; Vardag et al., 2014). UCI uses another independent scale that was established in 1978 and is traceable to NIST (Flores et al., 2015; Simpson et al., 2012), but has not been included in standard exchanges with other networks so differences with the other networks cannot be quantitatively defined. Additional experimental details are presented in the supplementary material from Kirschke et al. (2013) and references therein.

In Fig. 1, (a) globally averaged $CH_4$ and (b) its growth rate (derivative of the deseasonalized trend curve) through to 2017 are plotted for the four measurement programs using a procedure of signal decomposition described in Thoning et al. (1989). We define the annual $G_{ATM}$ as the increase in the atmospheric concentrations from Jan. 1 in one year to Jan. 1 in the next year. Agreement among the four networks is good for the global growth rate, especially since ~1990. The large differences observed mainly before 1990 reflect probably the different spatial coverage of each network. The long-term behaviour of globally averaged atmospheric $CH_4$ shows a decreasing but positive growth rate (defined as the derivative of the deseasonalized mixing ratio) from the early-1980s through 1998, a near-stabilization of $CH_4$ concentrations from 1999 to 2006, and a renewed period with positive persistent growth rates since 2007, slightly larger after 2014. When a constant atmospheric lifetime is assumed, the decreasing growth rate from 1983 through 2006 may imply that atmospheric $CH_4$ was approaching steady state, with no trend in emissions. The NOAA global mean $CH_4$ concentration was fitted with a function that describes the approach to a first-order steady state ($_{SS}$ index):

$[CH_4](t) = [CH_4]_{ss}-([CH_4]_{ss}-[CH_4]_0)e^{-t/\tau}$; solving for the lifetime, $\tau$, gives 9.3 years, which is very close to current literature values (e.g. Prather et al. (2012), $9.1 \pm 0.9$ years). Such an approach includes uncertainties, especially due to the strong assumption of no trend in emissions and sinks, which does not agree with some study explaining the stabilization period by decreasing emissions associated to increasing sink (e.g. Bousquet et al., 2006). However, this value seems consistent albeit higher than the chemistry climate estimates (8.2 years, see Sect. 3.3.5)

From 1999 to 2006, the annual increase of atmospheric $CH_4$ was remarkably small at $0.6\pm0.1$ ppb $yr^{-1}$. After 2006, the atmospheric growth rate has recovered to a level similar to that of the mid-1990s (~5 ppb $yr^{-1}$), or

even to that of the 1980s for 2014 and 2015 (>10 ppb yr$^{-1}$). On decadal timescales, the annual increase is on average 2.1±0.3 ppb yr$^{-1}$ for 2000-2009, 6.6±0.3 ppb yr$^{-1}$ for 2008-2017 and 6.1±1.0 ppb yr$^{-1}$ for the year 2017.

### 4.1.2 Satellite data of column average CH$_4$

In this budget, we use satellite data from the JAXA satellite Greenhouse Gases Observing SATellite (GOSAT) launched in January 2009 (Butz et al., 2011; Morino et al., 2011) containing the TANSO-FTS instrument, which observes in the shortwave infrared (SWIR). Different retrievals of methane based on TANSO-FTS/GOSAT products are made available to the community: from NIES (Yoshida et al., 2013), from SRON (Schepers et al., 2012) and from University of Leicester (Parker et al., 2011). The three retrievals are used by the top-down systems (Table 4 and S6). Although GOSAT retrievals still show significant unexplained biases and limited sampling in cloud covered regions and in the high latitude winter, it represents an important improvement compared to the first satellite measuring methane from space, SCIAMACHY (Scanning Imaging Absorption spectrometer for Atmospheric CartograpHY) both for random and systematic observation errors (see Table S2 of Buchwitz et al. (2016)).

Atmospheric inversions based on SCIAMACHY and GOSAT CH$_4$ retrievals were reported in Saunois et al. (2016). Here, only inversions using GOSAT retrievals are used.

### 4.2 Top-down inversions used in the budget

An atmospheric inversion is the optimal combination of atmospheric observations, of a model of atmospheric transport and chemistry, of a prior estimate of methane sources and sinks, and of their uncertainties, in order to provide improved estimates of the sources and sinks, and their uncertainty. The theoretical principle of methane inversions is detailed in the Supplementary Material (ST2) and an overview of the different methods applied to methane is presented in Houweling et al. (2017).

We consider here an ensemble of inversions gathering various chemistry transport models, differing in vertical and horizontal resolutions, meteorological forcing, advection and convection schemes, and boundary layer mixing. Including these different systems is a conservative approach that allows to cover different potential uncertainties of the inversion, among them: model transport, set-up issues, and prior dependency. General characteristics of the inversion systems are provided in Table 4. Further details can be found in the referenced papers and in the Supplementary Material. Each group was asked to provide gridded flux estimates for the period 2000-2017, using either surface or satellite data, but no additional constraints were imposed so that each group could use their preferred inversion setup. A set of prior emission distributions was built from the most recent inventories or model-based estimates (see Supplementary Material), but its use was not mandatory (Table S6). This approach corresponds to a flux assessment, but not to a model inter-comparison

as the protocol was not too stringent. Estimating posterior uncertainty is time and computer resource consuming, especially for the 4D-var approaches and Monte Carlo methods. Posterior uncertainties have been provided by only two groups and are found to be lower than the ensemble spread. Indeed, chemistry transport models differ in inter-hemispheric transport, stratospheric methane profiles and OH distribution, which limitations are not fully considered in the individual posterior uncertainty. As a result, we do not use

the posterior uncertainties provided by these two groups but report the minimum-maximum range among the different top-down approaches.

Nine atmospheric inversion systems using global Eulerian transport models were used in this study compared to eight in Saunois et al. (2016). Each inversion system provided one or several simulations, including sensitivity tests varying the assimilated observations (surface or satellite) or the inversion setup. This

represents a total of 22 inversion runs with different time coverage: generally, 2000-2017 for surface-based observations, and 2010-2017 for GOSAT-based inversions (Table 4 and Table S6). In poorly observed regions, top-down surface inversions may rely on the prior estimates and bring little or no additional information to constrain (often) spatially overlapping emissions (e.g. in India, China). Also, we recall that many top-down systems solve for the total fluxes at the surface only or for some categories that may differ

from the GCP categories. When multiple sensitivity tests were performed the mean of this ensemble was used not to overweight one particular inverse system. It should also be noticed that some satellite-based inversions are in fact combined satellite and surface inversions as they use satellite retrievals and surface measurements simultaneously (Alexe et al., 2015; Bergamaschi et al., 2013; Houweling et al., 2014). Nevertheless, these inversions are still referred to as satellite-based inversions.

Each group provided gridded monthly maps of emissions for both their prior and posterior total and for sources per category (see the categories Sect. 2.3). Results are reported in Sect. 5. Atmospheric sinks from the top-down approaches have been provided for this budget, and are compared with the values reported in Kirschke et al. (2013). Not all inverse systems report their chemical sink; as a result, the global mass imbalance for the top-down budget is derived as the difference between total sources and total sinks for each

model when both fluxes were reported.

### 5 Methane budget: top-down and bottom-up comparison

### 5.1 Global methane budget

### 5.1.1 Global total methane emissions

**Top-down estimates.** At the global scale, the total emissions inferred by the ensemble of 22 inversions is

1655 576 Tg $CH_4$ $yr^{-1}$ [550-594] for the 2008-2017 decade (Table 3), with the highest ensemble mean emission of

596 Tg CH$_4$ yr$^{-1}$ [572-614] for 2017. Global emissions for 2000-2009 (547 Tg CH$_4$ yr$^{-1}$) are consistent with Saunois et al. (2016) and the range for global emissions, 524-560 Tg CH$_4$ yr$^{-1}$ is in line with range in Saunois et al. (2016) (535-569), although the ensemble of inverse systems contributing to this budget is different than for Saunois et al. (2016). Indeed, only six inverse systems of the nine examined here (Table S7) contributed previously to the Saunois et al. (2016) budget. The range reported gives the minimum and maximum values among studies and do not reflect the individual full uncertainties. Also, most of the top-down models use the same OH distribution from the TRANSCOM experiment (Patra et al., 2011), leading to a global budget quite constrained, probably explaining the rather low range (10 %) compared to bottom-up estimates (see below).

**Bottom-up estimates.** The estimates made via the bottom-up approaches considered here are quite different from the top-down results, with global emissions almost 30% larger, at 737 Tg CH$_4$ yr$^{-1}$ [594-881] for 2008-2017 (Table 3). Moreover, the range estimated using bottom up approaches does not overlap with that of the top-down estimates. The bottom-up estimates are given by the sum of individual anthropogenic and natural processes, without any constraint on the total. For the period 2000-2009, the discrepancy between bottom-up and top-down was 30% of the top-down estimates in Saunois et al. (2016) (167 Tg CH$_4$ yr$^{-1}$); this has been reduced only slightly (now 156 Tg CH$_4$ yr$^{-1}$ for the same 2000-2009 period). This reduction is due to 1) a better agreement in the anthropogenic emissions (top-down and bottom-up difference reducing from 19 Tg CH$_4$ yr$^{-1}$ to 2 Tg CH$_4$ yr$^{-1}$), 2) a reduction in the estimates of some natural sources other than wetlands based on recent literature (by 7 Tg CH$_4$ yr$^{-1}$ from geological sources, by 8 Tg CH$_4$ yr$^{-1}$ from wild animals, and by 3 Tg CH$_4$ yr$^{-1}$ from allocation of wildfires to biomass & biofuel burning, see Table 3), and 3) a reduction of 35 Tg CH$_4$ yr$^{-1}$ in the bottom-up estimates of wetland emissions by models when excluding lakes and paddies as wetlands (see Sect. 5.1.2 below). These reductions (-70 Tg CH$_4$ yr$^{-1}$) in the bottom-up budget are partially offset by revised freshwater emissions with higher values (+ 37 Tg CH$_4$ yr$^{-1}$) resulting from the integration of a recent study on lake, pond and reservoir emissions (DelSontro et al. (2018), see Sect. 3.2.2) and the integration of estuary emissions in this budget (+4 Tg CH$_4$ yr$^{-1}$). Overall, the uncertainty range of some natural emissions has decreased in this study compared to Kirschke et al. (2013) and Saunois et al. (2016), for example for oceans, termites, wild animals, and geological sources. However, the uncertainty in the global budget remains high because of the large range reported for emissions from freshwater systems. Still, as noted in Kirschke et al. (2013), such large global emissions from the bottom-up approaches are not consistent with top-down estimates that rely on OH burden constrained by methylcholorform atmospheric observations, and are very likely overestimated. This overestimation likely results from errors related to up-scaling local measurements and double counting of some natural sources (e.g. wetlands, other inland water systems, see Sect. 5.1.2).

### 5.1.2 Global methane emissions per source category

The global methane emissions from natural and anthropogenic sources (see Sect. 2.3) for 2008-2017 are presented in Fig. 5, Fig. 6, and Table 3. Top-down estimates attribute about 60% of total emissions to anthropogenic activities (range of 55-70 %), and 40% to natural emissions. As natural emissions estimated from bottom-up approaches are much larger, the anthropogenic versus natural emission ratio is nearly 1, not consistent with ice core data. A current predominant role of anthropogenic sources of methane emissions is consistent with and strongly supported by available ice core and atmospheric methane records. These data indicate that atmospheric methane varied around 700 ppb during the last millennium before increasing by a factor of 2.6 to ~1800 ppb since pre-industrial times. Accounting for the decrease in mean-lifetime over the industrial period, Prather et al. (2012) estimated from these data a total source of $554\pm56$ Tg $CH_4$ in 2010 of which about 64% ($352\pm45$ Tg $CH_4$) was of anthropogenic origin, consistent with the range in our stop-down estimates.

**Wetlands.** For 2008-2017, the top-down and bottom-up derived estimates of 181 Tg $CH_4$ $yr^{-1}$ (range 159-200) and 149 Tg $CH_4$ $yr^{-1}$ (range 102-182), respectively, are statistically consistent. Bottom-up mean wetland emissions for the 2000-2009 period are smaller in this study than those of Saunois et al. (2016). Conversely, the current 2000-2009 mean top-down wetland estimates are larger than those of Saunois et al. (2016) (Table 3). The reduction in wetland emissions from bottom-up models is related to an updated wetland extent data set (WAD2M, see Sect. 3.2.1). Top-down wetlands emissions estimates are higher on average but the range is reduced by 50% compared to Saunois et al. (2016) for 2000-2009. In the bottom-up estimates, the amplitude of the range of emissions of 102-179 is similar to that in Saunois et al. (2016) (151-222 for 2000-2009), and narrowed by a third compared to the previous estimates from Melton et al. (2013) (141-264) and from Kirschke et al. (2013) (177-284). Here and in Saunois et al. (2016), the land surface models were forced with the same wetland extent and climate forcing (see Sect 3.2.1) contrary to Melton et al. (2013) and Kirschke et al. (2013). This suggests that differences in wetland extent explain about a third (30-40%) of the former range of the emission estimates of global natural wetlands. The remaining range is due to differences in model structures and parameters. Bottom-up and top-down estimates for wetland emissions differ more in this study (~30 Tg $yr^{-1}$ for the mean) than in Saunois et al. (2016) (~17 Tg $yr^{-1}$), due to reduced estimates from the bottom-up models and increased estimates from the top-down models. Natural emissions from freshwater systems are not included in the prior fluxes entering the top-down approaches. However, emissions from these non-wetland systems may be accounted for in the posterior estimates of the top-down models, as these two sources are close and probably overlap at the rather coarse resolution of the top-down models. In the top-down budget, natural wetlands represent 30% on average of the total methane emissions

but only 22% in the bottom-up budget (because of higher total emissions inferred). Neither bottom-up nor top-down approaches included in this study point to significant changes in wetland emissions between the two decades 2000-2009 and 2008-2017 at the global scale.

**Other natural emissions.** The discrepancy between top-down and bottom-up budgets is the largest for the natural emission total, which is 371 Tg $CH_4$ $yr^{-1}$ [245-488] for bottom-up and only 218 Tg $CH_4$ $yr^{-1}$ [183-248] for top-down over the 2008-2017 decade. This discrepancy comes from the estimates in "other natural" emissions (freshwater systems, geological sources, termites, oceans, and permafrost). Indeed, for the 2008-2017 decade, top-down inversions infer non-wetland emissions of 37 Tg $CH_4$ $yr^{-1}$ [21-50], whereas the sum of the individual bottom-up emissions is 222 Tg $CH_4$ $yr^{-1}$ [143-306]. Atmospheric inversions infer about the same amount over the decade 2000-2009 as over 2008-2017, which is almost half of the value reported in Saunois et al. (2016) (68 [21-130] Tg $CH_4$ $yr^{-1}$). This reduction is either due to 1) a more consistent way of considering other natural emissions in the various inverse systems or 2) a difference in the ensemble of top-down inversions reported here. It is worth noting that lacking gridded products to use in their prior, most of the top-down models include only ocean and termite emissions in their prior scenarios. Some of them now include geological sources, but none include freshwater or permafrost emissions in their prior fluxes, and thus in their posterior estimates. Regarding the bottom-up budget, the two main contributors to the larger bottom-up total are freshwaters (~75%) and geological emissions (~15%), both of which have large uncertainties and lack of spatially explicit representation with gridded products available to date, for freshwaters for example. Because of the discrepancy, the category "other natural" represents 7% of total emissions in the top-down budget, but up to 25% in the bottom-up budget.

Geological emissions are associated with relatively large uncertainties, and marine seepage emissions are still widely debated (Thornton et al., 2020). However, summing up all bottom-up fossil-$CH_4$ related sources (including the anthropogenic emissions) leads to a total of 173 Tg $CH_4$ $yr^{-1}$ [131-219] in 2008-2017, which is about 30% of the top-down global methane emissions, and 23% of the bottom-up total global estimate. These results agree with the value inferred from [14]C atmospheric isotopic analyses of 30% contribution of fossil-$CH_4$ to global emissions (Etiope et al., 2008; Lassey et al., 2007b). This total fossil fuel emissions from bottom-up approaches agrees well with the [13]C-based estimate of Schwietzke et al. (2016) at 192 ± 32 Tg $CH_4$ $yr^{-1}$. Uncertainties on bottom-up estimates of natural emissions lead to probably overestimated total methane emissions resulting in a lower contribution compared to Lassey et al. (2007b). All non-geological and non-freshwater land source categories (wild animals, termites, permafrost) have been evaluated at a lower level than in Kirschke et al. (2013) and Saunois et al. (2016), and contribute only 13 Tg $CH_4$ $yr^{-1}$ [4-19] to global emissions. From a top-down point of view, the sum of all the natural sources is more robust than the partitioning between wetlands and other natural sources. Better constraining the partitioning of methane emissions between wetlands and freshwater systems, including emissions from thawing permafrost, may be

the key to reconcile top-down and bottom-up budget on natural sources. Also, including all known spatio-temporal distributions of natural emissions in top-down prior fluxes would be a step forward to consistently compare natural versus anthropogenic total emissions between top-down and bottom-up approaches.

**Anthropogenic emissions.** Total anthropogenic emissions for the period 2008-2017 were assessed to be statistically consistent between top-down (359 Tg $CH_4$ yr$^{-1}$, range 336-376) and bottom-up approaches (366 Tg $CH_4$ yr$^{-1}$, range 349-393). The partitioning of anthropogenic emissions between agriculture and waste, fossil fuels extraction and use, and biomass and biofuel burning, also shows good consistency between top-down and bottom-up approaches, though top-down approaches suggest less fossil fuel and more agriculture and waste emissions than bottom-up estimates (Table 3 and Fig. 5 and 6). For 2008-2017, agriculture and waste contributed an estimated 217 Tg $CH_4$ yr$^{-1}$ [207-240] for the top-down budget and 206 Tg $CH_4$ yr$^{-1}$ [191-223] for the bottom-up budget. Fossil fuel emissions contributed 111 Tg $CH_4$ yr$^{-1}$ [81-131] for the top-down budget and 128 Tg $CH_4$ yr$^{-1}$ [113-154] for the bottom-up budget. Biomass and biofuel burning contributed 30 Tg $CH_4$ yr$^{-1}$ [22-36] for the top-down budget and 29 Tg $CH_4$ yr$^{-1}$ [25-39] for the bottom-up budget. Biofuel methane emissions rely on very few estimates currently (Wuebbles and Hayhoe, 2002). Although biofuel is a small source globally (~12 Tg $CH_4$ yr$^{-1}$), more estimates are needed to allow a proper uncertainty assessment. Overall for top-down inversions the global fraction of total emissions for the different source categories is 38% for agriculture and waste, 19% for fossil fuels, and 5% for biomass and biofuel burning. With the exception of biofuel emissions, the uncertainty associated with global anthropogenic emissions appears to be smaller than that of natural sources but with an asymmetric uncertainty distribution (mean significantly different than median). The relative agreement between top-down and bottom-up approaches may indicate the limited capability of the inversion to separate the emissions and should therefore be treated with caution. Indeed, in poorly observed regions, top-down inversions rely on the prior estimates and bring little or no additional information to constrain (often) spatially overlapping emissions (e.g. in India, China). Also, as many top-down systems solve for the total fluxes at the surface or for some categories that may differ from the GCP categories, their posterior partitioning relies on the prior ratio between categories that are prescribed using bottom-up inventories

### 5.1.3 Global budget of total methane sinks

**Top-down estimates.** The $CH_4$ chemical removal from the atmosphere is estimated to 518 Tg $CH_4$ yr$^{-1}$ over the period 2008-2017, with an uncertainty of about ±5% (range 474-532 Tg $CH_4$ yr$^{-1}$). All the inverse models account for $CH_4$ oxidation by OH and $O(^1D)$, and some include stratospheric chlorine oxidation (Table S6). In addition, most of the top-down models use OH distribution from the TRANSCOM experiment (Patra et al., 2011), probably explaining the rather low range of estimates compared to bottom-up estimates (see below). Differences between transport models affect the chemical removal of $CH_4$, leading to different

chemical loss rates, even with the same OH distribution. However, uncertainties in the OH distribution and magnitude (Zhao et al., 2019) are not considered in our study, while it could contribute to a significant change in the chemical sink, and then in the derived posterior emissions through the inverse process (Zhao et al., 2020). The chemical sink represents more than 90% of the total sink, the rest being attributable to soil uptake (38 [27-45] Tg $CH_4$ $yr^{-1}$). Half of the top-down models use the climatological soil uptake magnitude (37-38 Tg $CH_4$ $yr^{-1}$) and distribution from Ridgwell et al. (1999), while half of the models use an estimate from the biogeochemical model VISIT (Ito and Inatomi, 2012), which calculates varying uptake between 31 and 38 Tg $CH_4$ $yr^{-1}$ over the 2000-2017 period. These sink estimates used as prior in the inversions are generally higher than the mean estimate of the soil sink calculated by bottom-up models (30 Tg $CH_4$ $yr^{-1}$, Sec. 3.3.4).

**Bottom-up estimates.** The total chemical loss for the 2000s reported here is 595 Tg $CH_4$ $yr^{-1}$ with an uncertainty of 22% (~130 Tg $CH_4$ $yr^{-1}$). Differences in chemical schemes (especially in the stratosphere) and in the volatile organic compound treatment probably explain most of the discrepancies among models (Zhao et al., 2019).

### 5.2 Latitudinal methane budget

### 5.2.1 Latitudinal budget of total methane emissions

The latitudinal breakdown of emissions inferred from atmospheric inversions reveals a dominance of tropical emissions at 368 Tg $CH_4$ $yr^{-1}$ [337-399], representing 64% of the global total (Table 5). 32% of the emissions are from the mid-latitudes (186 Tg $CH_4$ $yr^{-1}$ [166-204]) and 4% from high latitudes (above 60°N). The ranges around the mean latitudinal emissions are larger than for the global methane sources. While the top-down uncertainty is about ±5% at the global scale, it increases to ±10% for the tropics and the northern mid-latitudes to more than ±25% in the northern high-latitudes (for 2008-2017, Table 5). Both top-down and bottom-up approaches consistently show that methane decadal emissions have increased by about 20 Tg $CH_4$ $yr^{-1}$ in the tropics, and by 7-18 Tg $CH_4$ $yr^{-1}$ in the northern mid-latitudes between 2000-2009 and 2008-2017, but not in the northern high latitudes.

Over 2010-2017, at the global scale, satellite-based inversions infer almost identical emissions to ground-based inversions (difference of 3 [0-7] Tg $CH_4$ yr-1), when comparing consistently surface versus satellite-based inversions for each system. This difference is much lower than the range derived between the different systems (range of 20 Tg $CH_4$ $yr^{-1}$ using surface- or satellite-based inversions). This result reflects that differences in atmospheric transport among the systems probably have more impact than the types of observations assimilated on the estimated global emissions. In Saunois et al. (2016), satellite-based inversions reported 12 Tg higher global methane emissions compared to surface-based inversions. Differences in the

ensemble, the use of only GOSAT data and the treatment of satellite data within each system compared to Saunois et al. (2016) explain the contrasting results.

As expected, the regional distributions of inferred emissions differ depending on the nature of the observations used (satellite or surface). The largest differences (satellite-based minus surface-based inversions) are observed over the tropical region, between -13 and +26 Tg $CH_4$ $yr^{-1}$ (90°S to 30°N), and the northern mid-latitudes (between -20 and +15 Tg $CH_4$ $yr^{-1}$). Satellite data provide stronger constraints on fluxes in tropical regions than surface data, due to a much larger spatial coverage. It is therefore not surprising that differences between these two types of observations are found in the tropical band, and consequently in the northern mid-latitudes to balance total emissions, thus affecting the north-south gradient of emissions. However, the regional patterns of these differences are not consistent through the different inverse systems. Indeed, some systems found higher emissions in the tropics when using GOSAT instead of surface observations, while others found the opposite. This difference between the systems may depend on whether or not a bias correction is applied to the satellite data based on surface observations, and also on the modelled horizontal and vertical model transports, in the troposphere and in the stratosphere.

### 5.2.2 Latitudinal methane emissions per source category

The analysis of the latitudinal methane budget per source category (Fig. 7) can be performed both for bottom-up and top-down approaches but with limitations. On the bottom-up side, some natural emissions are not (yet) available at regional scale (mainly inland waters). Therefore, for freshwater emissions, we applied the latitudinal distribution of Bastviken et al. (2011) to the global reported value. Further details are provided in the Supplement to explain how the different bottom-up sources were handled. On the top-down side, as already noted, the partitioning of emissions per source category has to be considered with caution. Indeed, using only atmospheric methane observations to constrain methane emissions makes this partitioning largely dependent on prior emissions. However, differences in spatial patterns and seasonality of emissions can be utilized to constrain emissions from different categories by atmospheric methane observations (for those inversions solving for different sources categories, see Sect. 2.3).

Agriculture and waste are the largest sources of methane emissions in the tropics (130 [121-137] Tg $CH_4$ $yr^{-1}$ for the bottom-up budget and 139 [127-157] Tg $CH_4$ $yr^{-1}$ for the top-down budget, about 38% of total methane emissions in this region). However, wetland emissions are nearly as large with 116 [71-146] Tg $CH_4$ $yr^{-1}$ for the bottom-up budget and 135 [116-155] Tg $CH_4$ $yr^{-1}$ for the top-down budget. One top-down model suggests lower emissions from agriculture and waste compared to the ensemble but suggests higher emissions from fossil fuel: this recalls the necessary caution when discussing sectorial partitioning when using top-down inversions. Anthropogenic emissions dominate in the northern mid-latitudes, with the highest contribution from agriculture and waste emissions (42% of total emissions), closely followed by fossil fuel

emissions (31% of total emissions). Boreal regions are largely dominated by wetland emissions (60% of total emissions).

The uncertainty for wetland emissions is larger in the bottom-up models than in the top-down models, while uncertainty in anthropogenic emissions is larger in the top-down models than in the inventories. The large uncertainty in tropical wetland emissions (65%) results from large regional differences between the bottom-up land-surface models. Although they are using the same wetland extent, their responses in terms of flux density show different sensitivities to temperature, water vapour pressure, precipitation, and radiation.

More regional discussions were developed in Saunois et al. (2016) and have been updated in Stavert et al. (2020).

## 6 Future developments, missing elements, and remaining uncertainties

In this budget, uncertainties on sources and sinks estimated by bottom-up or top-down approaches have been highlighted as well as discrepancies between the two budgets. Limitations of the different approaches have also been highlighted. Four shortcomings of the methane budget were already identified in Kirschke et al. (2013) and Saunois et al. (2016). Although progress has been made, they are still relevant, and actions are needed. However, these actions fall into different timescales and parties. In the following, we revisit the four shortcomings, or axis of research, of the current methane budget; how each weakness has been corrected since Saunois et al. (2016), followed by a list of recommendations, from higher to lower priority, associated with the involved parties.

1. *Towards a decrease of the high uncertainty in the amount of methane emitted by wetland and inland water systems, and a weakened double counting issue.*

The remaining large uncertainties strongly suggest the need to develop more studies integrating the different systems (wetlands, ponds, lakes, reservoirs, streams, rivers, estuaries, and marine systems), to avoid double counting issues, to associate proper emission to each category, but also to account for lateral fluxes. Since Saunois et al. (2016), several workshops (e.g. Turner et al., 2019) and publications (e.g. Knox et al., 2019; Thornton et al., 2016a) contributed to implement previous recommendations and strategies to reduce uncertainties of methane emissions due to wetlands and other freshwater systems. One achievement is the reduced estimate (by ~20%, i.e. 35 Tg $CH_4$ $yr^{-1}$) of the global wetland emissions, due to a refined wetland extent analysis and modifications of land surface model calibration.

Methodology changes that could be integrated into the next methane budget releases include:
-   Calibrating land surface models independently from top-down estimates;
-   Evaluating land surface models against in-situ observations such as FLUXNET-$CH_4$ (Knox et al., 2019); and

-     Using different wetland extent products to infer wetland emissions (e.g. WAD2M, GIEMS-2 (Prigent et al., 2020)).

Next steps, in the short term, for modelling, can be addressed by the land biogeochemistry community:

-     Finalizing a global high-resolution (typically tens of meters) classification of saturated soils and inundated surfaces based on satellite data (visible and microwave), surface inventories, and expert

knowledge. This improved area distribution will prevent double counting between wetlands and other freshwater systems, when used by land surface models;

-     Finalizing ongoing efforts to develop process–based modelling approaches to estimate freshwater methane emissions, including lateral fluxes, and avoiding upscaling issues, as recently done by e.g. Maavara et al. (2019) for $N_2O$; and

-     Using the collected flux measurements within the FLUXNET-$CH_4$ activity (Knox et al., 2019) to provide global flux maps based on machine learning approaches (Peltola et al., 2019).

Over the long run, developing measurement systems will help to improve estimates of wetland and inland water sources, and further reduce uncertainties:

-     More systematic measurements from sites reflecting the diverse lake morphologies will allow us to

better understand the short-term biological control on ebullition variability, which remains poorly known (Wik et al., 2014, 2016a); and

-     Extending monitoring of methane fluxes year round from the different natural sources (wetlands, freshwaters) complemented with environmental meta-data (e.g., soil temperature and moisture, vegetation types, water temperature, acidity, nutrient concentrations, NPP, soil carbon density) will

allow us to enrich the FLUXNET- $CH_4$ observation dataset, and to better constrain methane fluxes and their isotopic signatures in land-surface models (Glagolev et al., 2011; Turetsky et al., 2014).

2. *Towards a better assessment of uncertainties for global methane sinks in top-down and bottom-up budgets.*

The inverse systems used here have the same caveats as described in Saunois et al. (2016) (same OH field, same kind of proxy method to optimize it) leading to quite constrained atmospheric sink and therefore total global methane sources. Although we have used a state-of-the-art ensemble of Chemistry Transport Models (CTM) and Climate Chemistry Models (CCM) simulations from the CCMI (Chemistry-Climate Model Initiative, Morgenstern et al. (2017)), the uncertainty of derived $CH_4$ chemical loss from the chemistry

climate models remains at the same (large) level compared to the previous intercomparison project ACCMIP (Lamarque et al., 2013). Nicely et al. (2017) found that the main cause of the large differences in the CTM representation of $CH_4$ lifetime is variations in the chemical mechanisms implemented in the models. Using the ensemble of CTMs and CCMs from the CCMI experiment, Zhao et al. (2019) quantified the range of $CH_4$

loss induced by the ensemble of OH fields to be equivalent up to about half of the discrepancies between CH$_4$ observations and simulations as forced by the current anthropogenic inventories. These results emphasize the need to first assess, and then improve, atmospheric transport and chemistry models, especially vertically, and to integrate robust representation of OH fields in atmospheric models.

Methodology changes that could be integrated into the next methane budget include:

- Integrating sensitivity tests on the prior fluxes (use of updated fluxes for natural sources, soil uptake); and
- Integrating sensitivity tests on chemical sinks (different OH fields, including interannual variability).

Next steps, in the short term, could include developments by the atmospheric modelling community in:

- Assessing the impact of using updated and varying soil uptake estimates, especially considering a warmer climate. Indeed, for top-down models resolving for the net flux of CH$_4$ at the surface integrating a larger estimate of soil uptake would allow larger emissions, and then reduce the uncertainty with the bottom-up estimates of total CH$_4$ sources;
- Further studying the reactivity of the air parcels in the chemistry climate models and defining new diagnostics to assess modelled CH$_4$ lifetimes;
- Developing robust representation of 3D OH fields to be used in the inverse models: based on chemistry climate models and using correction from measurements, on multi-species assimilating systems (e.g. Gaubert et al., 2017; Miyazaki et al., 2015), or on simple parametrization applied at grid scales; and
- Integrating the aforementioned different potential OH chemical fields, including also inter-annual variability, to assess the impact on the methane budget following Zhao et al. (2020).

Over the long run, other parameters should be (better) integrated into top-down approaches, among them:

- The magnitude of the CH$_4$ loss through oxidation by tropospheric chlorine, a process debated in the recent literature. More modeling (Thanwerdas et al., 2019) and instrumental studies should be devoted to reducing the uncertainty of this potential additional sink before integrating it in top-down models.

3. *Towards a better partitioning of methane sources and sinks by region and process using top-down models*

In this work, we report inversions assimilating satellite data from GOSAT, which bring more constraints than provided by surface stations alone, especially over tropical continents. However, we found that satellite- and surface-based inversions, and the different inversions systems do not consistently infer the same regional flux distribution.

Methodology changes that could be integrated into the next methane budget releases include:

- Integrating GOSAT and GOSAT-2 (launched in October 2018, with expected improved precision and accuracy (JAXA, 2019)) for the satellite inversion; and

- Investigating the reasons for the regional differences derived by the inverse systems based on the model evaluation and more detailed questionnaire to the modelers on the treatment of satellite data (bias correction) and stratospheric profiles.

Next steps, in the short term, could integrate developments to be made by the top-down community:

- Evaluating the benefits of using new satellite missions with high spatial resolution and "imaging capabilities" (Crisp et al., 2018) at the global scale, such as the TROPOMI instrument on Sentinel 5P, launched in October 2017 (Hu et al., 2018);

- Integrating the newly available updated gridded products for the different natural sources of $CH_4$ in their prior fluxes to reach a full spatial description of sources and sinks, and to be able to better compare the top-down budget with the bottom-up budget;

- Releasing more regular updates and intercomparison of emission inventories in order to improve prior scenarios of inverse studies and reduce the need for extending them beyond their available coverage;

- Developing 4D variational inversion system using isotopic and/or co-emitted species in the top-down budget. Indeed methane isotopes can provide additional constraints to partition the different $CH_4$ sources and sinks, if isotopic signatures can be better known spatially and temporally (Ganesan et al., 2018). Radiocarbon can help for fossil / non-fossil emissions (Lassey et al., 2007b, 2007a; Petrenko et al., 2017), $^{13}CH_4$ and $CH_3D$ for biogenic / pyrogenic / thermogenic emissions, and OH loss (Röckmann et al., 2011), and emerging clumped isotope measurements for biogenic/thermogenic emissions (Stolper et al., 2014) and OH loss (Haghnegahdar et al., 2017). Also, carbon monoxide (e.g. Fortems-Cheiney et al., 2011) can provide useful constraints for biomass burning emissions, and ethane for fugitive emissions (e.g. Simpson et al., 2012; Turner et al., 2019).

- Improving the availability of in-situ data for the scientific community, especially ones covering poorly documented regions such as China (Fang et al., 2015), India (Lin et al., 2015; Tiwari and Kumar, 2012) and Siberia (Sasakawa et al., 2010; Winderlich et al., 2010), which are not delivered so far to international databases.

Over the long run, integrating more measurements and regional studies will help to improve the top-down systems, and further reduce the uncertainties:

- Integrating global data from future satellite instruments with intrinsic low-bias, such as active LIDAR techniques with MERLIN (Ehret et al., 2017), that are promising to overcome issues of

systematic errors (Bousquet et al., 2018) and should provide measurements over the Arctic, contrary to the existing and planned passive missions;

- Extending the $CH_4$ surface networks to poorly observed regions (e.g. Tropics, China, India, high latitudes) and to the vertical dimension: aircraft regular campaigns (e.g. Paris et al., 2010; Sweeney et al., 2015); Aircore campaigns (e.g. Andersen et al., 2018; Membrive et al., 2017) ; TCCON observations (e.g. Wunch et al., 2011, 2019). These observations are still critical to complement satellite data that do not observe well in cloudy regions and at high latitudes, and also to evaluate and eventually correct satellite biases (Buchwitz et al., 2016);

- Extending and developing continuous isotopic measurements of methane using laser-based instruments to help partitioning methane sources and to be integrated in 4D variational isotopic inversions;

- Developing regional components of the $CH_4$ budget to improve global totals by feeding them with regional top-down and bottom-up approaches; for example, regional inversions using regional measurements and high resolution models (such as the INGOS project (Bergamaschi et al., 2018b) or the VERIFY project (https://verify.lsce.ipsl.fr) with the European ICOS network (https://www.icos-ri.eu/home). The RECCAP-2 project should also provide a scientific framework to further refine GHG budgets, including methane, at regional scales (https://www.reccap2-gotemba2019.org/).

4. *Towards reducing uncertainties in the modelling of atmospheric transport in the models used in the top-down budget*

The TRANSCOM experiment synthesized in Patra et al. (2011) showed a large sensitivity of the representation of atmospheric transport on methane concentrations in the atmosphere. In particular, the modelled $CH_4$ budget appeared to depend strongly on the troposphere-stratosphere exchange rate and thus on the model vertical grid structure and circulation in the lower stratosphere. Also, regional changes in the methane budget depends on the characteristics of the atmospheric transport models used in the inversion (Bruhwiler et al., 2017; Locatelli et al., 2015). This axis of research is demanding important development from the atmospheric modelling community. Waiting for future improvements (finer horizontal and vertical resolutions, more accurate physical parameterization, increase in computing resources…), assessing atmospheric transport error and the impact on the top-down budget remains crucial.

Methodology changes that could be integrated into the next methane budget releases include:

- Evaluating the inversions provided against independent measurements such as aircraft regular campaigns (e.g. Paris et al., 2010; Sweeney et al., 2015); Aircore campaigns (e.g. Andersen et al.,

2018; Membrive et al., 2017) ; TCCON observations (e.g. Wunch et al., 2011, 2019), and use this evaluation to weight the different models used in the methane budget

Next steps, in the short term, could include some development to be addressed by the top-down community to reduce atmospheric transport errors:

- Developing further methodologies to extract stratospheric partial column abundances from
observations such as TCCON data (Saad et al., 2014; Wang et al., 2014), Aircore or, even ACE-FTS or MIPAS satellite data, and use them to replace erroneous simulated stratospheric profiles.

In the long run, developments within atmospheric transport models such as the implementation of hybrid vertical coordinates (Patra et al., 2018) or of hexagonal-icosaedric grid with finer resolution (Dubos et al., 2015; Niwa et al., 2017a), and improvements in the simulated boundary layer dynamics are promising to
reduce atmospheric transport errors.

## 7 Conclusions

We have built a global methane budget by using and synthesizing a large ensemble of published methods and results using a consistent and transparent approach, including atmospheric observations and inversions (top-down models), process-based models for land surface emissions and atmospheric chemistry, and
inventories of anthropogenic emissions (bottom-up models and inventories). For the 2008-2017 decade, global $CH_4$ emissions are 576 Tg $CH_4$ yr$^{-1}$ (range of 550-594 Tg $CH_4$ yr$^{-1}$ ), as estimated by top-down inversions. About 60% of global emissions are anthropogenic (range of 50-70%). Bottom-up models and inventories suggest much larger global emissions (737 Tg $CH_4$ yr$^{-1}$ [594-881]) mostly because of larger and more uncertain natural emissions from inland water systems, natural wetlands and geological leaks, and some
likely unresolved double counting of these sources. It is also likely that some of the individual bottom-up emission estimates are too high, leading to larger global emissions from the bottom-up perspective than the atmospheric constraints suggest.

The latitudinal breakdown inferred from top-down approaches reveals a dominant role of tropical emissions (~64%) compared to mid (~32%) and high (~4%) northern latitudes (above 60°N) emissions.

Our results, including an extended set of atmospheric inversions, are compared with the previous budget syntheses of Kirschke et al. (2013) and Saunois et al. (2016), and show overall good consistency when comparing the same decade (2000-2009) at the global and latitudinal scales, although estimation methods and reported studies have evolved between the three budgets. While, a comparison of top-down emissions estimates determined with and without satellite data agrees well globally they differ significantly at the
latitudinal scale. Most worryingly, these differences were not even consistent in sign with some models showing notable increases in a given latitudinal flux and others decreases. This suggests that while the

inclusion of satellite data may, in the future, significantly increase our ability to attribute fluxes regionally this is not currently the case due to their existing inherent biases along with the inconsistent application of methods to account for these biases, but also differences in model transport, especially in the stratosphere
(see recommendations in Sect. 6).

Among the different uncertainties raised in Kirschke et al. (2013), Saunois et al. (2016) estimated that 30-40% of the large range associated with modelled wetland emissions in Kirschke et al. (2013) was due to the estimation of wetland extent. Here, wetland emissions are 35 Tg $CH_4$ $yr^{-1}$ smaller than previous estimates due to a refinement of wetland extent. The magnitude and uncertainty of all other natural sources have been
revised and updated, leading to smaller emission estimates for oceans, geological sources, and wild animals, but higher emission estimates associated to larger range for non-wetland freshwater systems. This result places a number one priority on reducing uncertainties in emissions from inland water systems by better quantifying the emission factors of each contributing sub-systems (streams, rivers, lakes, ponds) and reducing both uncertain up-scaling and likely double counting with wetland emissions. As a second priority,
the uncertainty on the chemical loss of methane need to be better assessed in both the top-down and the bottom-up budgets. Our work also suggests the need for more interactions among groups developing emission inventories in order to clarify the definition of the sectoral breakdown in inventories. Such an approach would allow easier comparisons at the sub-category scale.

Building on the improvement of the points detailed in Sect. 6, our aim is to update this budget synthesis as a
living review paper regularly (~every two/three years). Each update will produce a more recent decadal $CH_4$ budget, highlight changes in emissions and trends, and incorporate newly available data and model improvements.

In addition to the decadal $CH_4$ budget presented in this paper, and following former studies (e.g. Bousquet et al., 2006), trends and year-to-year changes in the methane cycle continues to be thoroughly discussed in the
recent literature (e.g. Nisbet et al., 2019; Turner et al., 2019). After almost a decade of stagnation in the late 1990s and early 2000s (Dlugokencky et al., 2011; Nisbet et al., 2016), a sustained atmospheric growth rate of more than +5 ppb $yr^{-1}$ has been observed since 2007, with a further acceleration after 2014 (Nisbet et al., 2019), and several years with a 2-digit atmospheric growth as in the 1980s. To date, no consensus has yet been reached in explaining the $CH_4$ trend since 2007. A likely explanatory scenario, already introduced in
Saunois et al. (2017) and further investigated by some other studies since then, includes, by decreasing order of certainty, a positive contribution from microbial and fossil sources (e.g. Nisbet et al., 2019; Schwietzke et al., 2016), a negative contribution from biomass burning emissions before 2014 (Giglio et al., 2013; Worden et al., 2017), a downward revision of Chinese emissions, a negligible role of Arctic emission changes (e.g. Nisbet et al., 2019; Saunois et al., 2017), a tropical dominance of the increasing emissions (e.g. Saunois et
al., 2017). Changes in atmospheric OH concentrations, the largest methane sink, may have contributed to the

recent increase in methane concentrations (e.g. Dalsøren et al., 2016; Holmes et al., 2013; McNorton et al., 2016, 2018; Morgenstern et al., 2018; Rigby et al., 2017; Turner et al., 2017), but considerable uncertainty in OH interannual variability and trends need to be further investigated. The challenging and sustained increase of atmospheric $CH_4$ during the past decade still needs additional research to be fully understood

(Nisbet et al., 2019; Turner et al., 2019). The GCP will continue to take its part in analysing and synthesizing recent changes in the global to regional methane cycle based on the ensemble of top-down and bottom-up studies gathered for the budget analysis presented here.

## 8 Data availability

The data presented here are made available in the belief that their dissemination will lead to greater

understanding and new scientific insights on the methane budget and changes to it, and helping to reduce its uncertainties. The free availability of the data does not constitute permission for publication of the data. For research projects, if the data used are essential to the work to be published, or if the conclusion or results largely depend on the data, co-authorship should be considered. Full contact details and information on how to cite the data are given in the accompanying database.

The accompanying database includes one Excel file organized in the following spreadsheets and two netcdf files defining the regions used to extend the anthropogenic inventories.

The file Global_Methane_Budget_2000-2017_v2.0.xlsx includes (1) a summary, (2) the methane observed mixing ratio and growth rate from the four global networks (NOAA, AGAGE, CSIRO and UCI), (3) the evolution of global anthropogenic methane emissions (including biomass burning emissions) used to produce

Fig. 2, (4) the global and latitudinal budgets over 2000–2009 based on bottom-up approaches, (5) the global and latitudinal budgets over 2000–2009 based on top-down approaches, (6) the global and latitudinal budgets over 2008–2017 based on bottom-up approaches, (7) the global and latitudinal budgets over 2008– 2017 based on top-down approaches, (8) the global and latitudinal budgets for year 2017 based on bottom-up approaches, (9) the global and latitudinal budgets for year 2017 based on top-down approaches, and (10) the

list of contributors to contact for further information on specific data.

This database is available from ICOS (https://doi.org/10.18160/GCP-CH4-2019, Saunois et al., 2019) and the Global Carbon Project (http://www.globalcarbonproject.org).

**Author contributions**.

MS, AS and BP gathered the bottom-up and top data sets and performed the post processing and analysis. MS, AS, BP, PB, PeC, and RJ coordinated the global budget. MS, AS, BP, PB, PeC, RJ, SH PP and PCi contribute to the update of the full text and all coauthors appended comments. MS, ED and GP produced the

figures. VA, NG, AI, FJ, TK, LL, KMcD, PM, JMe, JMu, CP, SP, WR, HS, HT, WZ, ZZ, Qing Z, Qiuan Z and Qianlai Z performed surface land model simulations to compute wetland emissions. DB, MC, PC, SC, KC, GE, GH, KMJ, GL, SN, CP, PRa, Pre, BT, NV, TW provided data sets usefull for natural emission estimates and/or contribute to text on bottom-up natural emissions. LHI, GJM, FT, GvW, KMC, provided anthropogenic data sets and contribute to the text for this section. PP, BP, NC, MI, SM, JMcN, YN, AS, AT, YY, and BZ performed atmospheric inversions to compute top-down methane emission estimates. DRB, GB, CCr, CF, PK, RL, TM, IM, SO'D, RJP, RP, MR, IJS, PS, YT, RFW, DWo, DWu, YYo are PI of atmospheric observations used in top-down inversions and/or contributed the text describing atmospheric methane observations. YZ, MvW, AV, VN, MIH contribute to the chemical sink section by providing data set, processing data and/or contribute to the text. FMF and CCu provided data for the soil sink and contributed to the text of this section.

**Competing interests.** The authors declare that they have no conflict of interest.

**Acknowledgements**

This paper is the result of a collaborative international effort under the umbrella of the Global Carbon Project, a project of Future Earth and a research partner of the World Climate Research Programme. We acknowledge all the people and institutions who provided the data used in the global methane budget as well as the institutions funding parts of this effort (see Table A1). We acknowledge the modelling groups for making their simulations available for this analysis, the joint WCRP SPARC/IGAC Chemistry-Climate Model Initiative (CCMI) for organising and coordinating the model data analysis activity, and the British Atmospheric Data Centre (BADC) for collecting and archiving the CCMI model output.We acknowledge Adrian Gustafson for his contribution to prepare the simulations of LPJ-GUESS. Paul A. Miller, Adrian Gustafson and Wenxin Zhang acknowledge this work as a contribution to the Strategic Research Area MERGE. FAOSTAT data collection, analysis and dissemination is funded through FAO regular budget funds. The contribution of relevant experts in member countries is gratefully acknowledged. We acknowledge Juha Hatakka (FMI) for making methane measurements at the Pallas station and sharing the data with the community.

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

-------

**Table 1: B-U models and inventories for anthropogenic and biomass burning inventories used in this study. \*Due to its limited sectorial breakdown this dataset was not used in Table 3. ^Extended to 2017 for this study as described in Section 3.1.1.**

| B-U models and inventories | Contribution | Time period (resolution) | Gridded | References |
|---|---|---|---|---|
| CEDS (country based) | Fossil fuels, Agriculture and waste, Biofuel | 1970-2015^ (yearly) | no | Hoesly et al. (2018) |
| CEDS (gridded)* | Fossil fuels, Agriculture and waste, Biofuel | 1970-2014 (monthly) | 0.5x0.5° | Hoesly et al. (2018) |
| EDGARv4.2.3 | Fossil fuels, Agriculture and waste, Biofuel | 1990-2012^ (yearly) | 0.1x0.1° | Janssens-Maenhout et al. (2019) |
| IIASA GAINS ECLIPSEv6 | Fossil fuels, Agriculture and waste, Biofuel | 1990-2015^ (1990-2015 yearly, >2015 5-yr interval interpolated to yearly) | 0.5x0.5° | Höglund-Isaksson (2012) |
| USEPA | Fossil fuels, Agriculture and waste, Biofuel, Biomass Burning | 1990-2030 (10-yr interval, interpolated to yearly) | no | USEPA (2012) |
| FAO-CH4 | Agriculture, Biomass Burning | 1961-2016^ 1990-2016 (Yearly) | no | Frederici et al. (2015) ; Tubiello et al.(2013);Tubiello (2019) |
| FINNv1.5 | Biomass burning | 2002-2018 (daily) | 1km resolution | Wiedinmyer et al. (2011) |
| GFASv1.3 | Biomass burning | 2003-2016 (daily) | 0.1x0.1° | Kaiser et al. (2012) |
| GFEDv4.1s | Biomass burning | 1997-2017 (monthly) | 0.25x0.25° | Giglio et al. (2013) |
| QFEDv2.5 | Biomass burning | 2000-2017 (daily) | 0.1x0.1° | Darmenov and da Silva (2015) |

**Table 2: Biogeochemical models that computed wetland emissions used in this study. Runs were performed for the whole period 2000-2017. Models run with prognostic (using their own calculation of wetland areas) and/or diagnostic (using WAD2M) wetland surface areas (see Sect 3.2.1).**

| Model | Institution | Prognostic | Diagnostic | References |
|---|---|---|---|---|
| CLASS-CTEM | Environment and Climate Change Canada | y | y | Arora et al. (2018); Melton and Arora (2016) |
| DLEM | Auburn University | n | y | Tian et al. (2010, 2015) |
| ELM | Lawrence Berkeley National Laboratory | y | y | Riley et al. (2011) |
| JSBACH | MPI | n | y | Kleinen et al. (2019) |
| JULES | UKMO | y | y | Hayman et al. (2014) |
| LPJ GUESS | Lund University | n | y | McGuire et al. (2012) |
| LPJ MPI | MPI | n | y | Kleinen et al. (2012) |
| LPJ-WSL | NASA GSFC | y | y | Zhang et al. (2016) |
| LPX-Bern | University of Bern | y | y | Spahni et al. (2011) |
| ORCHIDEE | LSCE | y | y | Ringeval et al. (2011) |
| TEM-MDM | Purdue University | n | y | Zhuang et al. (2004) |
| TRIPLEX_GHG | UQAM | n | y | Zhu et al. (2014, 2015) |
| VISIT | NIES | y | y | Ito and Inatomi (2012) |

**Table 3: Global methane emissions by source type in Tg CH$_4$ yr$^{-1}$ from Saunois et al. (2016) (left column pair) and for this work using bottom-up and top-down approaches). Because top-down models cannot fully separate individual processes, only five categories of emissions are provided (see text). Uncertainties are reported as [min-max] range of reported studies. Differences of 1 Tg CH$_4$ yr$^{-1}$ in the totals can occur due to rounding errors.**

|  | Saunois et al. (2016) | | This work | | | | | |
|---|---|---|---|---|---|---|---|---|
| Period of time | 2000-2009 | | 2000-2009 | | 2008-2017 | | 2017 | |
| Approaches | bottom-up | top-down | bottom-up | top-down | bottom-up | top-down | bottom-up | top-down |
| **NATURAL SOURCES** | | | | | | | | |
| **Wetlands** | **183** [151-222] | **166** [125-204] | **147** [102-179] | **180** [153-196] | **149** [102-182] | **181** [159-200] | **145** [100-183] | **194** [155-217] |
| **Other natural sources** | **199** [104-297] | **68** [21-130] | **222** [143-306] | **35** [21-47] | **222** [143-306] | **37** [21-50] | **222** [143-306] | **39** [21-50] |
| **Other land sources** | **185** [99-272] | | **209** [134-284] | | | | | |
| Freshwaters [a] | 122 [60-180] | | 159 [117-212] | | | | | |
| Geological (onshore) | 40 [30-56] | | 38 [13-53] | | | | | |
| Wild animals | 10 [5-15] | | 2 [1-3] | | | | | |
| Termites | 9 [3-15] | | 9 [3-15] | | | | | |
| Wildfires | 3 [1-5] | | (**) | | | | | |
| Permafrost soils (direct) | 1 [0-1] | | 1 [0-1] | | | | | |
| Vegetation | (*) | | (*) | | | | | |
| **Oceanic sources** | **14** [5-25] | | **13** [9-22] | | | | | |
| Geological (offshore) | 12 [5-20] | | 7 [5-12] | | | | | |
| Biogenic open and coastal [b] | 2 [0-5] | | 6 [4-10] | | | | | |
| **TOTAL NATURAL SOURCES** | **382** [255-519] | **234** [194-292] | **369** [245-485] | **215** [176-243] | **371** [245-488] | **218** [183-248] | **367** [243-489] | **232** [194-267] |
| **ANTHROPOGENIC SOURCES** | | | | | | | | |
| **Agriculture and waste** | **190** [174-201] | **183** [112-241] | **192** [178-206] | **202** [198-219] | **206** [191-223] | **217** [207-240] | **213** [198-232] | **227** [205-246] |
| Enteric ferm. & manure | 103 [95-109][c] | | 104 [93-109] | | 111 [106-116] | | 115 [110-121] | |
| Landfills & waste | 57 [51-61][c] | | 60 [55-63] | | 65 [60-69] | | 68 [64-71] | |
| Rice cultivation | 29 [23-35][c] | | 28 [23-34] | | 30 [25-38] | | 30 [24-40] | |
| **Fossil fuels** | **112** [107-126] | **101** [77-126] | **110** [94-129] | **101** [71-151] | **128** [113-154] | **111** [81-131] | **135** [121-164] | **108** [91-121] |
| Coal mining | 36 [24-43][e] | | 32 [24-42] | | 42 [29-61] | | 44 [31-63] | |
| Oil & Gas | 76 [64-85][cf] | | 73 [60-85] | | 80 [68-92] | | 84 [72-97] | |
| Industry | - | | 2 [0-6] | | 3 [0-7] | | 3 [0-8] | |
| Transport | - | | 4 [1-11] | | 4 [1-12] | | 4 [1-13] | |
| **Biomass & biof. burn.** | **30** [26-34] | **35** [16-53] | **31** [26-46] | **29** [23-35] | **30** [26-40] | **30** [22-36] | **29** [24-38] | **28** [25-32] |
| Biomass burning | 18 [15-20] | | 19 [15-32] | | 17 [14-26] | | 16 [11-24] | |
| Biofuel burning | 12 [9-14] | | 12 [9-14] | | 12 [10-14] | | 13 [10-14] | |
| **TOTAL ANTHROPOGENIC SOURCES [g]** | **338** [329-342] | **319** [255-357] | **334** [321-358] | **332** [312-347] | **366** [349-393] | **359** [336-376] | **380** [359-407] | **364** [340-381] |
| **SINKS** | | | | | | | | |
| **Total chemical loss** | **604** [483-738] | **514**[e] | **595** [489-749] | **505** [h] [459-516] | **595** [489-749] | **518** [h] [474-532] | **595** [489-749] | **531** [h] [502-540] |
| Tropospheric OH | 528 [454-617] | | 553 [476-677] | | | | | |
| Stratospheric loss | 51 [16-84] | | 31 [12-37] | | | | | |
| Tropospheric Cl | 25 [13-37] | | 11 [1-35] | | | | | |

| | | | | | | | | |
|---|---|---|---|---|---|---|---|---|
| Soil uptake | **28** [9-47] | **32** [27-38] | **30** [11-49] | **34** [27-41] | **30** [11-49] | **38** [27-45] | **30** [11-49] | **40** [37-47] |
| TOTAL SINKS | **632** [592-785] | **546**[d] | **625** [500-798] | **540** [486-556] | **625** [500-798] | **556** [501-574] | **625** [500-798] | **571** [540-585] |
| **SOURCES – SINKS IMBALANCE** | | | | | | | | |
| TOTAL SOURCES | **719** [583-861] | **552** [535-566] | **703** [566-842] | **547** [524-560] | **737** [594-881] | **576** [550-594] | **747** [602-896] | **596** [572-614] |
| TOTAL SINKS | **632** [592-785] | **546**[d] | **625** [500-798] | **540** [486-556] | **625** [500-798] | **556** [501-574] | **625** [500-798] | **571** [540-585] |
| IMBALANCE | | **6**[d] | **78** | **3** [-10-38][h] | **112** | **13** [0- 49][h] | **120** | **12** [0- 41][h] |
| ATMOSPHERIC GROWTH [i] | | **6.0** [4.9-6.6] | | **5.8** [4.9-6.6] | | **18.2** [17.3-19.0] | | **16.8** [14.0-19.5] |

(*) uncertain but likely small for upland forest and aerobic emissions, potentially large for forested wetland, but likely included elsewhere

(**) We stop reporting this value to avoid potential double counting with satellite-based products of biomass burning (see Sect. 3.1.5)

a: Freshwater includes lakes, ponds, reservoirs, streams and rivers

b: includes flux from hydrates considered at 0 for this study, includes estuaries

c: For IIASA inventory the breakdown of agriculture and waste (rice, Enteric fermentation & manure, Landfills & waste) and fossil fuel (coal, oil, gas & industry) sources used the same ratios as the mean of EDGAR and USEPA inventories in Saunois et al. (2016).

d: total sink was deduced from global mass balance and not directly computed in Saunois et al. (2016).

e: computed as the difference of global sink and soil uptake in Saunois et al. (2016).

f: Industry and transport emissions were included in the Oil & Gas category in Saunois et al. (2016)

g: Total anthropogenic emissions are based on estimtes of full anthropogenic inventory and not on the sum of "Agriculture and Waste", "Fossil fuels" and "Biofuel and biomass burning" categories (see Sect. 3.1.2)

h: Some inversions did not provide the chemical sink. These values are derived from a subset of the inversion ensemble.

i: Atmospheric growth are given in the same unit Tg CH$_4$ yr$^{-1}$, based on the conversion factor of 2.75 Tg CH$_4$ ppb$^{-1}$ given by Prather et al. (2012) and the atmospheric growth rates provided in the text in ppb yr$^{-1}$.

**Table 4: Top-down studies used in our new analysis, with their contribution to the decadal and yearly estimates noted. For decadal means, top down studies have to provide at least 8 years of data over the decade to contribute to the estimate.**

| Model | Institution | Observation used | Time period | Number of inversions | 2000-2009 | 2008-2017 | 2017 | References |
|---|---|---|---|---|---|---|---|---|
| Carbon Tracker-Europe CH$_4$ | FMI | Surface stations | 2000-2017 | 1 | y | y | y | Tsuruta et al. (2017) |
| Carbon Tracker-Europe CH$_4$ | FMI | GOSAT NIES L2 v2.72 | 2010-2017 | 1 | n | y | y | Tsuruta et al. (2017) |
| GELCA | NIES | Surface stations | 2000-2015 | 1 | y | y | n | Ishizawa et al. (2016) |
| LMDz-PYVAR | LSCE/CEA | Surface stations | 2010-2016 | 2 | n | y | n | Yin et al. (2015) |
| LMDz-PYVAR | LSCE/CEA | GOSAT Leicester v7.2 | 2010-2016 | 4 | n | y | n | Yin et al. (2015) |
| LMDz-PYVAR | LSCE/CEA | GOSAT Leicester v7.2 | 2010-2017 | 2 | n | y | y | Zheng et al. (2018b, 2018a) |
| MIROC4-ACTM | JAMSTEC | Surface stations | 2000-2016 | 1 | y | y | n | Patra et al. (2016, 2018) |
| NICAM-TM | NIES | Surface stations | 2000-2017 | 1 | y | y | y | Niwa et al. (2017a, 2017b) |
| NIES-TM-FLEXPART (NTF) | NIES | Surface stations | 2000-2017 | 1 | y | y | y | Maksyutov et al. (2020); Wang et al. (2019a) |
| NIES-TM-FLEXPART (NTF) | NIES | GOSAT NIES L2 v2.72 | 2010-2017 | 1 | n | y | y | Maksyutov et al. (2020); Wang et al. (2019a) |
| TM5-CAMS | TNO/VU | Surface stations | 2000-2017 | 1 | y | y | y | Bergamaschi et al. (2010, 2013); Pandey et al. (2016); Segers and Houwelling (2018) |
| TM5-CAMS | TNO/VU | GOSAT ESA/CCI v2.3.8 | 2010-2017 | 1 | n | y | y | Bergamaschi et al. (2010, 2013); Pandey et al. |

| | | (combined with surface observations) | | | | | | (2016); Segers and Houwelling (2018) |
|---|---|---|---|---|---|---|---|---|
| TM5-4DVAR | EC-JRC | Surface stations | 2000-2017 | 2 | y | y | y | Bergamaschi et al. (2013, 2018) |
| TM5-4DVAR | EC-JRC | GOSAT OCPR v7.2 (combined with surface observations) | 2010-2017 | 2 | n | y | y | Bergamaschi et al. (2013, 2018) |
| TOMCAT | Uni. of Leeds | Surface stations | 2003-2015 | 1 | n | y | n | McNorton et al. (2018) |

**Table 5: Global and latitudinal total methane emissions in Tg CH$_4$ yr$^{-1}$, as decadal means (2000-2009 and 2008-2017) and for the year 2017, for this work using bottom-up and top-down approaches. Global emissions for 2000-2009 are also compared with Saunois et al. (2016) and Kirschke et al. (2013) for top-down and bottom-up approcahes. Latitudinal total emissions for 2000-2009 are compared with Saunois et al. (2016) for top-down studies only. Uncertainties are reported as [min-max] range. Differences of 1 Tg CH$_4$ yr$^{-1}$ in the totals can occur due to rounding errors.**

| Period | 2000-2009 | | 2008-2017 | | 2017 | |
|---|---|---|---|---|---|---|
| Approach | Bottom-up | Top-down | Bottom-up | Top-down | Bottom-up | Top-down |
| **Global** | | | | | | |
| This work | **703** [566-842] | **547** [524-560] | **737** [594-881] | **576** [550-594] | **747** [602-896] | **596** [572-614] |
| *Saunois et al. (2016)* | *719 [583-861]* | *552 [535-566]* | - | - | - | - |
| *Kirschke et al. (2013)* | *678 [542-852]* | *553 [526-569]* | - | - | - | - |
| **90°S-30°N** | | | | | | |
| This work | **408** [322-532] | **346** [320-379] | **430** [338-547] | **368** [337-399] | **434** [343-568] | **383** [351-405] |
| *Saunois et al. (2016)* | - | 356 [334-381] | - | - | - | - |
| **30°N-60°N** | | | | | | |
| This work | **252** [202-342] | **178** [159-199] | **267** [218-349] | **186** [166-204] | **272** [223-351] | **188** [171-209] |
| *Saunois et al. (2016)* | - | 176 [159-195] | - | - | - | - |
| **60°N-90°N** | | | | | | |
| This work | **42** [28-70] | **23** [17- 32] | **43** [26-72] | **22** [17- 29] | **40** [24- 70] | **24** [21- 28] |
| *Saunois et al. (2016)* | - | 20 [15-25] | - | - | - | - |

**Table 6: Latitudinal methane emissions in Tg CH$_4$ yr$^{-1}$ for the last decade 2008-2017, based on top-down and bottom-up approaches. Uncertainties are reported as [min-max] range of reported studies. Differences of 1 Tg CH$_4$ yr$^{-1}$ in the totals can occur due to rounding errors. For bottom-up approaches, other natural sources (as a result total methane emissions and natural emissions) are not reported here due to a lack of spatial distribution for some sources (freshwater). Bottom-up anthropogenic estimates are based only on the gridded products from EDGARv4.3.2 and GAINS.**

| Latitudinal band | 90°S- 30°N | | 30°N-60°N | | 60°-90°N | |
|---|---|---|---|---|---|---|
| Approach | Bottom-up | Top-Down | Bottom-up | Top-Down | Bottom-up | Top-Down |
| **Natural Sources** | 228 [155340] | 160 [130-189] | 115 [70- 192] | 42 [29-54] | 31 [18- 55] | 16 [11-20] |
| **Natural Wetland** | 116 [71-146] | 135 [116-155] | 25 [10-43] | 33 [24-48] | 9 [2-18] | 13 [7-16] |
| **Other natural** | 112 [84-194] | 25 [14-36] | 90 [60-149] | 9 [4-14] | 22 [16-37] | 3 [2-4] |
| **Anthropogenic sources** | 202 [183-217] | 208 [186-229] | 152 [148-157] | 144 [117-170] | 12 [8-8] | 6 [2-10] |
| **Agriculture & Waste** | 130 [121-137] | 139 [127-157] | 80 [77-84] | 78 [67-87] | 1 [1-1] | 1 [1-2] |
| **Fossil Fuels** | 53 [43-71] | 47 [37-52] | 67 [61-71] | 60 [34-85] | 10 [6-15] | 4 [2-7] |
| **Biomass & biofuel burning** | 20 [18-22] | 22 [18-28] | 7 [6-9] | 6 [5-8] | 1 [0-1] | 1 [1-1] |
| **Sum of sources** | 430 [338-557] | 368 [337-399] | 267 [218-349] | 186 [166-204] | 43 [26- 72] | 22 [17- 29] |

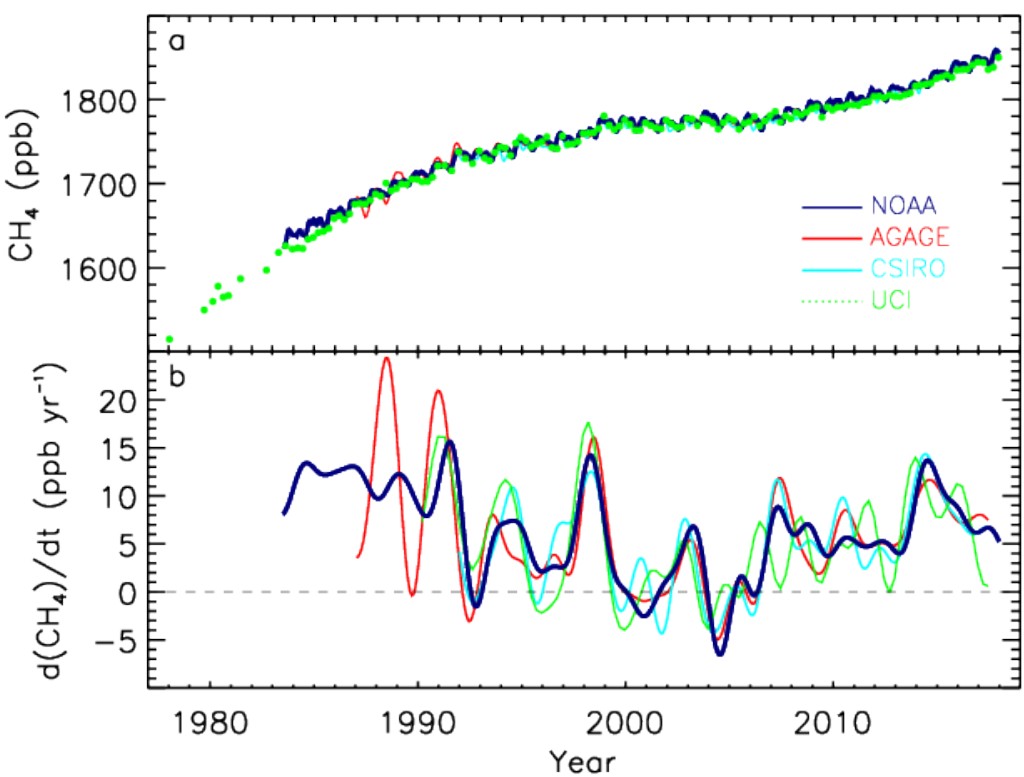

**Figure 1: Globally averaged atmospheric CH$_4$ (ppb) (a) and its annual growth rate G$_{ATM}$ (ppb yr$^{-1}$) (b) from four measurement programs, National Oceanic and Atmospheric Administration (NOAA), Advanced Global Atmospheric Gases Experiment (AGAGE), Commonwealth Scientific and Industrial Research Organisation (CSIRO), and University of California, Irvine (UCI). Detailed descriptions of methods are given in the supplementary material of Kirschke et al.** (2013).

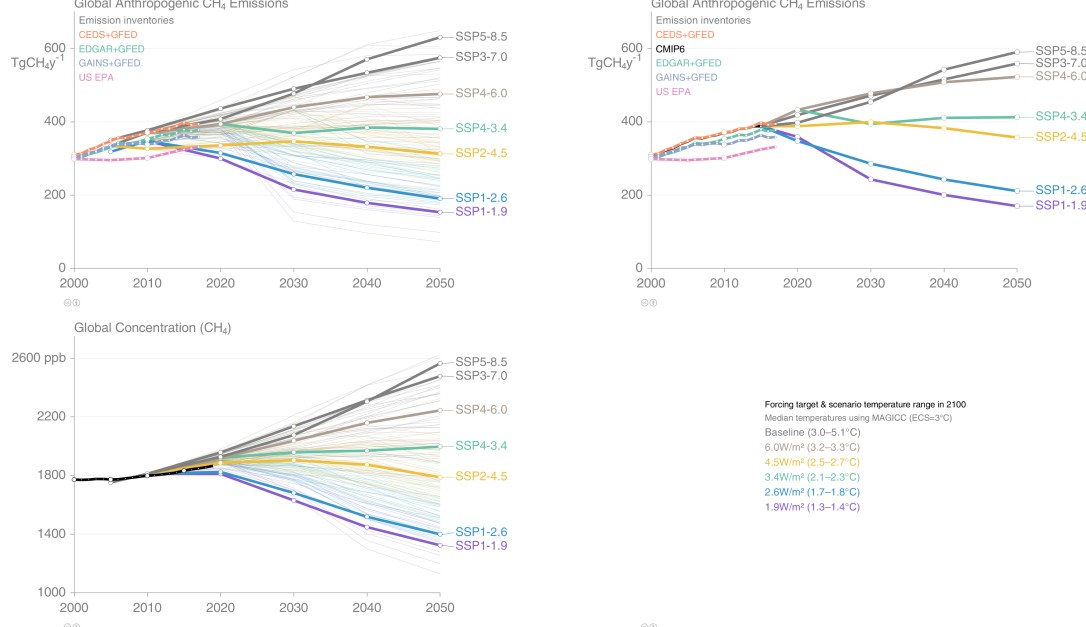

**Figure 2:** Top: Global anthropogenic methane emissions (including biomass burning) from historical inventories and future projections (in Tg CH$_4$ yr$^{-1}$). Top left panel shows inventories and the unharmonized Shared Socioeconomic PAthways (Riahi et al., 2017), with highlighted scenarios representing scenarios assessed in CMIP6 (O'Neill, et al., 2016). Top right panel shows the selected scenarios harmonized with historical emissions (CEDS) for CMIP6 activities (Gidden et al., 2019). USEPA and GAINS estimates have been linearly interpolated from the 5-year original products to yearly values. After 2005, USEPA original estimates are projections. Bottom left: Global methane concentrations for NOAA surface site observations (black) and projections based on SSPs (Riahi et al., 2017) with concentrations estimated using MAGICC (Meinshausen et al., 2011).

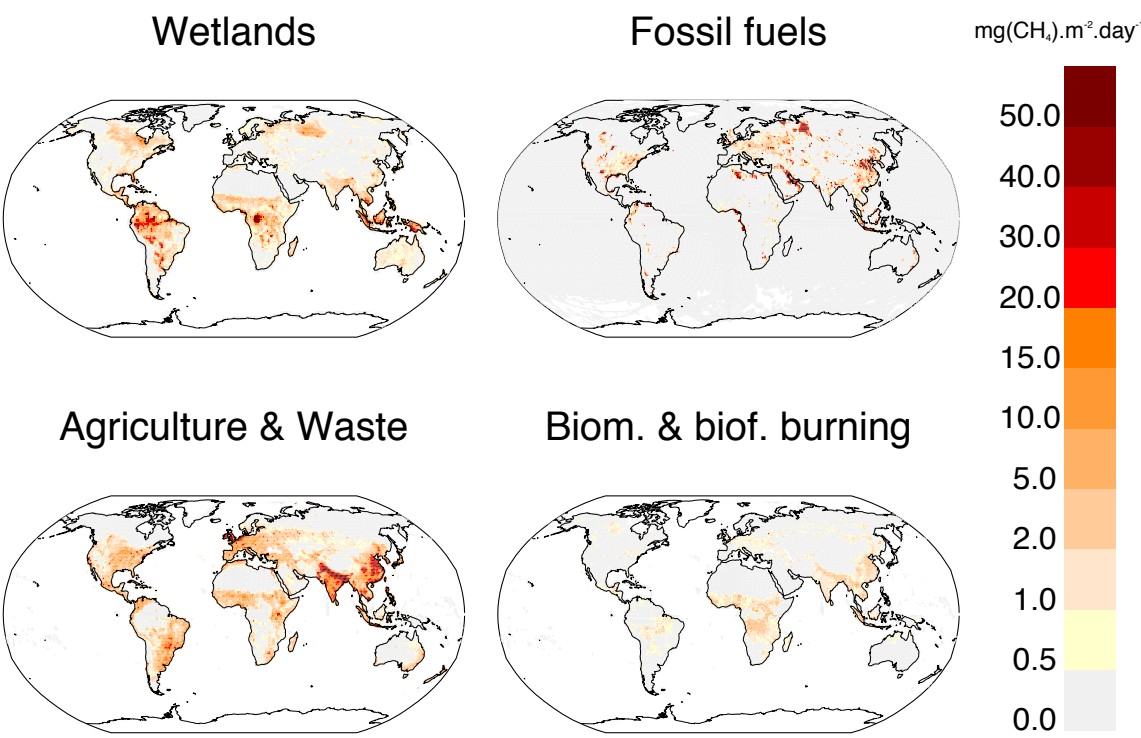

**Figure 3: Methane emissions from four source categories: natural wetlands (excluding lakes, ponds, and rivers), biomass and biofuel burning, Agriculture and Waste, and Fossil fuels for the 2008-2017 decade in mg CH$_4$ m$^{-2}$ day$^{-1}$. The wetland emission map represents the mean daily emission average over the 13 biogeochemical models listed in Table 2 and over the 2008-2017 decade. Fossil fuel and Agriculture and Waste emission maps are derived from the mean estimates of gridded CEDS, EGDARv4.3.2 and GAINS models. The biomass and biofuel burning map results from the mean of the biomass burning inventories listed in Table 1 added to the mean of the biofuel estimate from CEDS, EDGARv4.3.2 and GAINS models.**

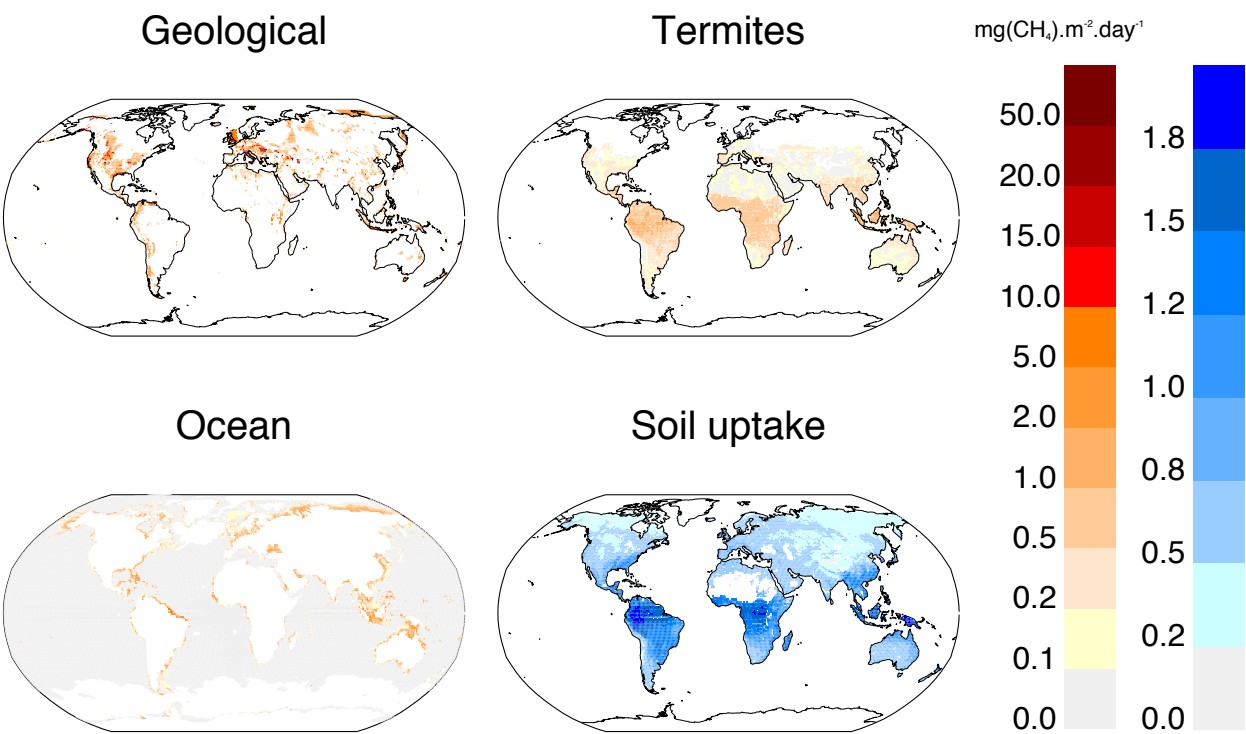

**Figure 4: Methane emissions (mg CH$_4$ m$^{-2}$ day$^{-1}$) from three natural sources (left color scale): geological (Etiope et al., 2019), termites (this study) and oceans (Weber et al., 2019), and methane uptake in soils (mg CH$_4$ m$^{-2}$ day$^{-1}$) presented in positive units (right color scale), and based on Murguia-Flores et al. (2018).**

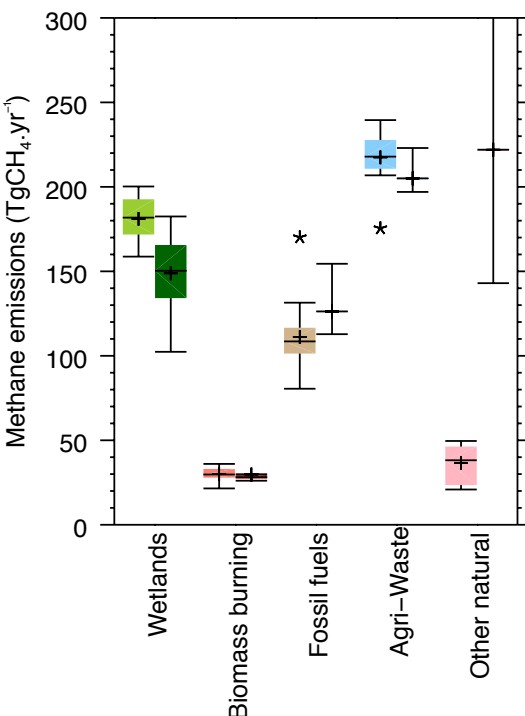

**Figure 5: Methane global emissions from the five broad categories (see Sect. 2.3) for the 2008-2017 decade for top-down inversions models (left light coloured boxplots) in Tg CH₄ yr⁻¹ and for bottom-up models and inventories (right dark coloured boxplots). Median value, first and third quartiles are presented in the boxes. The whiskers represent the minimum and maximum values when suspected outliers are removed (see Sect. 2.2). Suspected outliers are marked with stars when existing. Bottom-up quartiles are not available for bottom-up estimates, except for wetland emissions. Mean values are represented with "+" symbols, these are the values reported in Table 3.**

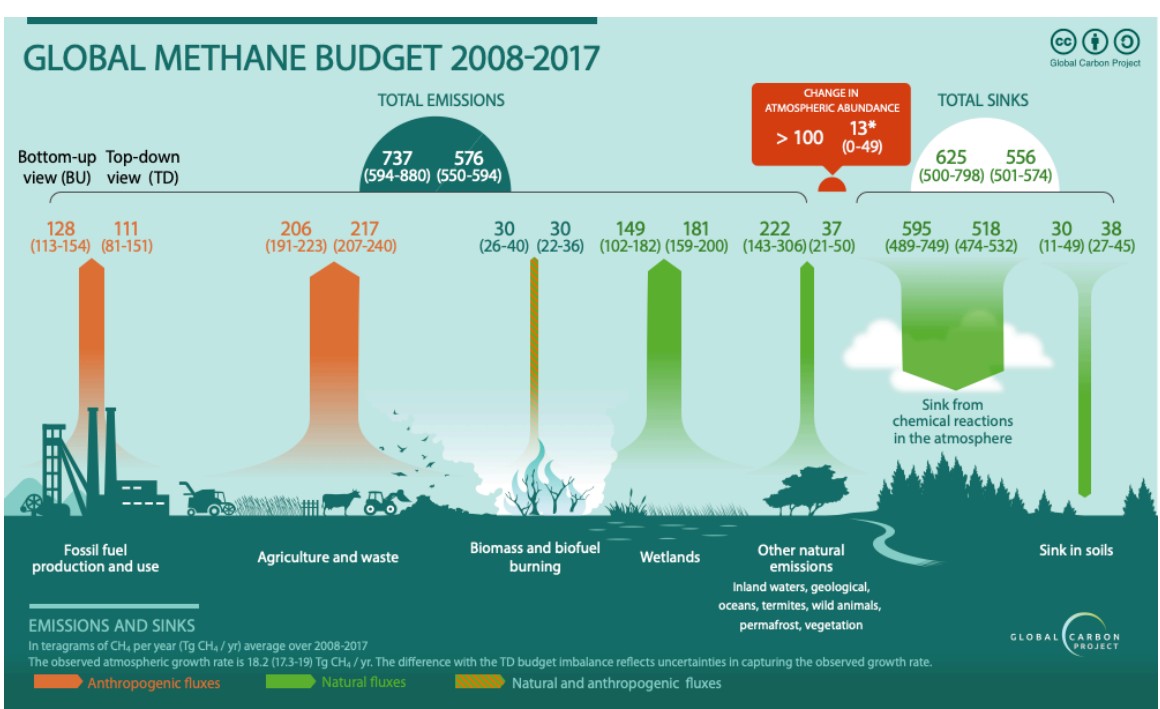

**Figure 6: Global Methane Budget for the 2008-2017 decades. Both bottom-up (left) and top-down (right) estimates are provided for each emission and sink category in Tg CH$_4$ yr$^{-1}$, as well as for total emissions and total sinks. Biomass and biofuel burning emissions are depicted here as both natural and anthropogenic emissions while they are fully included in anthropogenic emissions in the budget tables and text (Sect. 3.1.5).**

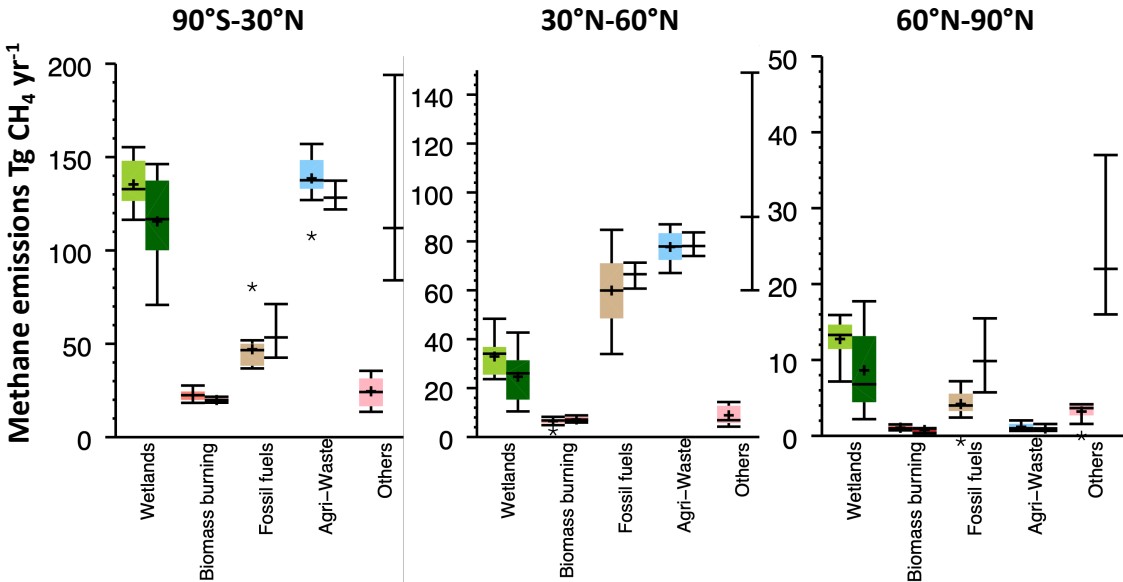

**Figure 7: Methane latitudinal emissions from the five broad categories (see Sect. 2.3) for the 2008-2017 decade for top-down inversions models (left light coloured boxplots) in Tg CH₄ yr⁻¹ and for bottom-up models and inventories (right dark coloured boxplots). Median value, first and third quartiles are presented in the boxes. The whiskers represent the minimum and maximum values when suspected outliers are removed (see Sect. 2.2). Suspected outliers are marked with stars as shown. Bottom-up quartiles are not available for bottom-up estimates, except wetland emissions. Mean values are represented with "+" symbols, these are the values reported in Table 6.**

**Table A1.** Funding supporting the production of the various components of the global methane budget in addition to the authors' supporting institutions (see also acknowledgements).

| Funder and grant number (where relevant) | Authors/Simulations/Observations |
| --- | --- |
| ARC Linkage project LP150100519 | Judith Rosentreter |
| Australian National Environmental Science Program-Earth Systems and Climate Hub | Joseph Canadell |
| Environment Research and Technology Development Fund (2-17002) of the Ministry of the Environment, Japan | Naveen Chandra and Prabir K. Patra |
| Environment Research and Technology Development Fund (2-1701) of the Ministry of the Environment, Japan | Yosuke Niwa |
| Environment Research and Technology Development Fund (2-1710) of the Ministry of the Environment, Japan | Akihiko Ito |
| ESA GHG-CCI | Robert Parker |
| European Research Council (ERC; grant no. 725546, METLAKE) | David Bastviken |
| European Union's Horizon 2020 research and innovation programme under grant agreement (VERIFY project) No. 776810 | Pierre Regnier and Glen P. Peters |
| F.R.S.- FNRS for post-doctoral funding at the ULB | Goulven Laruelle |
| German Federal Ministry of Education and Research (BMBF) under the PalMod programme | Thomas Kleinen |
| Gordon and Betty Moore Foundation through Grant GBMF5439 "Advancing Understanding of the Global Methane Cycle" to Stanford University | Rob Jackson; co-P.I.s Philippe Bousquet, Marielle Saunois, Yuanhong Zhao, Josep Canadell, Gustaf Hugelius, and Ben Poulter |
| Linköping University | David Bastviken |
| Ministry of the Environment, Japan | Shamil Maksyutov |
| NASA | Donald R. Blake and Isobel J. Simpson (UCI) |
| NASA grant NNX17AI74G | Peter Raymond |
| NASA grant NNX17AK11G | Thomas Weber |
| NASA grant NNX17AK20G | Qianlai Zhuang and Licheng Liu |
| NASA through their Terrestrial Ecology program. | Ben Poulter |
| National Science and Engineering Research Council of Canada (NSERC) discovery grant | Changhui Peng |
| Newton Fund through the Met Office Climate Science for Service Partnership Brazil (CSSP Brazil) | Nicolas Gedney |
| NIES GOSAT Project | Shamil Maksyutov |
| RUDN "5-100" | Simona Castaldi |
| Swedish Research Council (VR) and Formas project no. 2016-01201 | Paul A. Miller, Adrian Gustafson and Wenxin Zhang |
| Swedish Research Council VR | Patrick Crill |
| Swedish Research Councils VR and FORMAS | David Bastviken |
| Swiss National Science Foundation (#200020_172476) | Fortunat Joos and Jurek Mueller |
| UK National Centre for Earth Observation (nceo020005) | Robert Parker |
| US Department of Energy, BER, RGCM, RUBISCO project under contract #DE-AC02-05CH11231 | William J. Riley and Qing Zhu |
| Computing Resources | |
| ECMWF computing resources under the special project "Improve European and global $CH_4$ and $N_2O$ flux inversions (2018-2020)". | Peter Bergamaschi |

| | |
|---|---|
| LSCE computing resources | Marielle Saunois, Philippe Bousquet, Bo Zheng and Yi Yin |
| NASA: grants NAG5-12669, NNX07AE89G, NNX11AF17G and NNX16AC98G to MIT | MIT theory and inverse modeling |
| Swedish National Infrastructure for Computing (SNIC) at the Lund University Centre for Scientific and Technical Computing (Lunarc), project no. 2017/1-423 – Aurora resource | LPJGUESS Simulations |
| Support for atmospheric observations | |
| Australian Antarctic Division | CSIRO flask network |
| Australian Institute of Marine Science | CSIRO flask network |
| Bureau of Meteorology (Australia) | Cape Grim AGAGE, CSIRO flask network |
| Commonwealth Scientific and Industrial Research Organisation (CSIRO, Australia) | Cape Grim AGAGE, CSIRO flask network |
| Department of the Environment and Energy (DoEE, Australia) | Cape Grim AGAGE |
| Meteorological Service of Canada | CSIRO flask network |
| NASA: grants NAG5-12669, NNX07AE89G, NNX11AF17G and NNX16AC98G to MIT; grants NAG5-4023, NNX07AE87G, NNX07AF09G, NNX11AF15G and NNX11AF16G to SIO | Operation of Mace Head, Trinidad Head, Barbados, American Samoa, and Cape Grim AGAGE stations and SIO calibration |
| National Oceanic and Atmospheric Administration (NOAA, USA) contract RA133R15CN0008 to the University of Bristol | Barbados |
| NOAA USA | CSIRO flask network |
| Refrigerant Reclaim Australia | Cape Grim AGAGE |
| UK Department for Business, Energy & Industrial Strategy (BEIS) contract TRN1537/06/2018 to the University of Bristol | Mace Head |
| ALICE High Performance Computing Facility at the University of Leicester | GOSAT retrievals |
| Japanese Ministry of Environment | GOSAT data, Robert Parker |
| Japanese Aerospace Exploration Agency, National Institute for Environmental Studies | GOSAT data, Robert Parker |