# Peer review of "The Global Methane Budget 2000-2017"

_Earth System Science Data, 2019_

## Short Comment (SC1) · 23 Sep 2019

Tonatiuh Guillermo Nuñez Ramirez

tnunez@bgc-jena.mpg.de

The emission estimates for the decade of 2000-2009 in Saunois 2016 have larger ranges than in Saunois 2019. I think, if the GCP condenses all information that exists, there should be at least a table explaining which studies were left out and why as supplementary information.

A major problem with the bottom-up budget is that it is so much larger than the top-down estimate. The border between wetlands and other fresh water systems is very fuzzy and more discussion is required. Historically wetlands have been classified as bogs, fens, swamps, floodplains, and shallow lakes (Bartlett 1993). For example lake Chapala, Mexico's largest lake, has a maximum depth of 2 m, is it a wetland or a lake? Is a floodplain to be considered a wetland or a freshwater system? For example, in the Amazon inundation can vary for several meters. Furthermore, in the Eastern Amazon,

emissions tend to be larger at as river flow starts to decrease in August and September (Devol 1988, Beck 2012, Ringeval 2014, Basso 2016). This seasonal maximum is not capture by any of the WETCHIMP models, which instead show a maximum between January and April (Ringeval 2014). Ringeval (2014) were able to reproduce a seasonal cycle of CH4 emissions from the Amazon mainstream that was more similar to observations by using output from a hydrology model to identify floodplains. Furthermore, from your description it seems you classify as wetlands as saturated soils and fresh water systems can be lakes, rivers, reservoirs. Early studies, e.g. Devol (1988, 1990), Bartlett (1988, 1990), Tathy 1992, Keller 1994 and Melack (2004) made measurements both over saturated soils, emergent plants and open water. These early studies were used to calibrate many models, for example, in Spahni (2011), the LPJ-Bern model was calibrate to match the seasonal cycle from an inverse modeling estimate. Furthermore, DelSontro et al. (2018) has very high emissions but is stratification and transport within a lake were not taken in consideration. For example, lakes in East Africa are highly stratified and anoxic below the mixed layer but the amount of emissions estimated by DelSontro (2018) is difficult to bring in agreement with satellite $CH_4$ cartographies (e.g. Frankenberg, 2011). they do not emit high quantities of methane continuously due to the same stratification of the water column. Is the future of CH4 emission modeling the merging of dynamic vegetation models with hydrology models?

With respect to the soil sink, your estimates are based on published model estimates. However, in these models, the sink strength depends on atmospheric mixing ratio (often a global constant value). For example, in the Curry (2007) model, the flux *j* is

$$j = C_0 * g_0 * r_w * r_c * \sqrt{D * k}$$

where $C_0$ [ppm] is the $CH_4$ mixing ratio and $g_0$ is a conversion factor from ppm to mass units. Taking this into account, the sink becomes much larger may become much larger and changes in time in proportion to the atmospheric abundance. Furthremore, both Ridgwell (1999) and Curry (2007) had use the ideal gas law to set the g0 parameter to 610 and 578 assuming a pressure of 100 kPa and temperatures of 0°C and 15°C

respectively. By determine the g0 per gridcell based on monthly temperatures and pressure, the g0 ranges between 320 and 750.

Additionally, there are important contribution from Hackstein (1994, 1996, 2006) concerning potentially large emissions from wild terrestrial vertebrates and three arthropod taxa apart from termites.

In the future, it would be useful to also have estimates of the year-to-year variability for wetlands and OH in order to understand what drove the observed year-to-year variability of the growth rate.

References

Bartlett, K. B., Crill, P. M., Sebacher, D. I., Harriss, R. C., Wilson, C., Melack, J. M. (1988). Methane Flux From the Central Amazonian Floodplain. Journal of Geophysical Research-Atmospheres, 93(D2), 1571–1582. http://doi.org/10.1029/JD093iD02p01571

Bartlett, K. B., Crill, P. M., Bonazi, J. A., Richey, J. E., Harriss, R. C. (1990). Methane Flux From the Amazon River Floodplain - Emissions During Rising Water. Journal of Geophysical Research-Atmospheres, 95, 16773–16788. http://doi.org/10.1029/JD095iD10p16773

Bartlett, K. B., Hariss, R. C. (1993). Review and Assessment of Methane Emissions From Wetlands (Vol. 26, pp. 261–320). Presented at the Chemosphere.

Basso, L. S., Gatti, L. V., Gloor, M., Miller, J. B., Domingues, L. G., Correia, C. S. C., Borges, V. F. (2016). Seasonality and interannual variability of CH4 fluxes from the eastern Amazon Basin inferred from atmospheric mole fraction profiles. Journal of Geophysical Research-Atmospheres, 121(1), 168–184. http://doi.org/10.1002/2015JD023874

Beck, V. (2012). Determination of the methane budget of the Amazon region utilizing airborne methane observations in combination with atmospheric transport and vegetation modeling. (C. Gerbig  K.-U. Totsche, Eds.). Max Planck Institute for Biogeochemistry, Jena.

Curry, C. L. (2007). Modeling the soil consumption of atmospheric methane at the global scale. Global Biogeochemical Cycles, 21(4), –. http://doi.org/10.1029/2006GB002818

Devol, A. H., Richey, J. E., CLARK, W. A., King, S. L.,  Martinelli, L. A. (1988). Methane Emissions to the Troposphere From the Amazon Floodplain. Journal of Geophysical Research-Atmospheres, 93(D2), 1583–1592.

Devol, A. H., Richey, J. E., Forsberg, B. R.,  Martinelli, L. A. (1990). Seasonal dynamics in methane emissions from the Amazon River floodplain to the troposphere. Journal of Geophysical Research (ISSN 0148-0227), 95, 16417–16426. http://doi.org/10.1029/JD095iD10p16417

DelSontro, T., Beaulieu, J. J.,  Downing, J. A. (2018). Greenhouse gas emissions from lakes and impoundments: Upscaling in the face of global change. Limnology and Oceanography Letters, 3(3), 64–75. http://doi.org/10.1007/s101150200013

Frankenberg, C., Aben, I., Bergamaschi, P., Dlugokencky, E. J., van Hees, R., Houweling, S., et al. (2011). Global column-averaged methane mixing ratios from 2003 to 2009 as derived from SCIAMACHY: Trends and variability. Journal of Geophysical Research, 116(D4), D04302–. http://doi.org/10.1029/2010JD014849

Hackstein, J. H. P.,  Stumm, C. K. (1994). Methane Production in Terrestrial Arthropods (Vol. 91, pp. 5441–5445). Presented at the Proceedings of the National Academy of Sciences of the United States of America. http://doi.org/10.1073/pnas.91.12.5441

Hackstein, J. H. P.,  van Alen, T. A. (1996). Fecal methanogens and vertebrate evolution. Evolution, 559–572.

Hackstein, J. H. P., Alen, T. A.,  Rosenberg, J. (2006). Methane Production by Terrestrial Arthropods. In H. König  A. Varma (Eds.), Soil Biology: Intestinal Microorganisms

of Soil Invertebrates (Vol. 6, pp. 155–180). Berlin/Heidelberg: Springer-Verlag Berlin. http://doi.org/10.1007/3-540-28185-1₇

Keller, M., Stallard, R. F. (1994). Methane emission by bubbling from Gatun Lake, Panama. Journal of Geophysical Research, 99(D4), 8307. http://doi.org/10.1029/92JD02170

Melack, J. M., Hess, L., Gastil, M., Forsberg, B., Hamilton, S., Lima, I., Novo, E. (2004). Regionalization of methane emissions in the Amazon Basin with microwave remote sensing. Global Change Biology, 10(5), 530–544. http://doi.org/10.1111/j.1529-8817.2003.00763.x

Ringeval, B., Houweling, S., van Bodegom, P. M., Spahni, R., van Beek, R., Joos, F., Röckmann, T. (2014). Methane emissions from floodplains in the Amazon Basin: towards a process-based model for global applications. Biogeosciences, 11, 1519–1558. http://doi.org/10.5194/bg-11-1519-2014

Saunois, M., Bousquet, P., Poulter, B., Peregon, A., Ciais, P., Canadell, J. G., et al. (2016). Global Methane Budget 2000-2012. Earth System Science Data, 8, 697–751. http://doi.org/10.5194/essd-8-697-2016

Spahni, R., Wania, R., Neef, L. (2011). Constraining global methane emissions and uptake by ecosystems. Biogeosciences.

Tathy, J. P., Cros, B., Delmas, R., Marenco, A., Servant, J., Labat, M. (1992). Methane Emission From Flooded Forest in Central Africa. Journal of Geophysical Research-Atmospheres, 97(D6), 6159–6168.

---

## Referee Comment (RC1) · Anonymous Referee #1 · 7 Nov 2019

Global CH4 budget

General comments

Thank authors for massive compilation effort. Important product! One hopes a large community will see and use. Good product for ESSD.

Push data out to 2008-2017 decade? Various valid reasons for this but scattered somewhat randomly through the narrative. Pull all of this information into a short clear statement very near the top of the manuscript to explain time span and time lag?

I and many others will want to use this in classroom. We might assign paper as background reading. Students will loudly object - too long! This reviewer agrees. Too long by a factor of two, at least. Almost long enough to need an executive summary? See suggestion below.

In present organization, permafrost emissions sit within natural emissions. Technically, and increasingly so, permafrost emissions forced by permafrost thaw should instead account in the anthropogenic category? But if you move terrestrial permafrost sources then you also should move sub-marine permafrost emissions, boreal plant sinks or sources, etc. No easy solution but the current location will prove increasingly problematic.

Whether or not the authors intend, this reviewer concludes that we face a larger and more urgent geospatial rather chemical challenge. In short terms, we need to fix the wetland inland freshwater distribution and extent uncertainties as our highest priority, with understanding and quantification of atmospheric OH distributions, reactions and impacts as probably a second priority. If authors intend these priorities, they should state them explicitly? They can always revise in a subsequent version. If this reviewer developed a wrong impression, then authors need to reconsider information they presented and how.

Topical editor suggested that I do a full review. Finding the track-changes version somewhere between daunting and impenetrable, I agreed. After plowing through so much material, however, I almost wish I had tried instead to follow the track changes version. Fundamentally, this is way too much material. Removal of the regional details to separate papers helps a lot, but the present version still represents way too much material. You want strong interest and to serve many users, but at present the too-long, too-disorganized, too-detailed text represents a severe barrier to almost every reader and potential user. Amazing compilation effort, but you need to present it after and on the basis of an equally amazing editorial effort. The latter effort remains sadly missing.

Specific comments - long list and I know I missed some issues.

Line 304: "the relatively small and variable number of studies" Do the authors mean a small number of studies with variable results? A variable number of studies covering small regions or sectors? Confusing.

Line 317: "processed in the same way" Does this phrase mean that monthly and annual emission data, from any source, were subjected to the identical spatial 1x1 gridding? But some source data exists at higher spatial resolution? E.g. would need some form of disaggregation? Later (lines 320, 321) a reader finds that calculations of monthly and annual means came from (came after?) gridding. Confusion between spatial and temporal averaging here?

Line 332: "the regional budget presented in (Stavert, 2019)" should read as 'the regional budget presented by Stavert (2019)'? Proofreaders will not know what to do with these punctuation errors so authors must take care to express them correctly.

Line 336: In the version I read ("the TransCom inter-comparison map (Gurney et al., 2004)", the Gurney et al reference shows as a non-functional hot link, different to all other references?

Lines 358, 35: again some references as apparently hot links, others in standard (expected) format. Why?

Line 359: "most of the inverse system the" Instead: inverse systems?

Line 373: "Methane sources and sinks" Because the (long and detailed) section 3 deals entirely with bottom-up (e.g. line 374) emissions, and because section 4 carries the explicit title of top-down (e.g. at line 1677: "Atmospheric observations and top-down inversions"), shouldn't the title of section 3 read as 'Bottom-up sources and sinks"? (By this point one hopes that most readers/users know that the discussion refers to CH4.) If section 3 does NOT deal solely with bottom-up sources and processes (like most readers, this reviewers loses track of bottom-up vs top-down deep in section 3, e.g. in the section on CH4 lifetimes) then the reference to bottom-up at line at 374, 375 needs to change?

Line 395 to 400: good section on definitions used here vs those used by IPCC. This reviewer agrees with these choices but, more important, thanks authors for explaining the differences.

Line 436: FAOSTAT is annual or every 5 years? Annual, evidently, from Table 1?

Line 457: "Although, country emissions" Delete the comma?

Line 472: Here the reader finds FAO-CH4 at annual resolution so FAOSTAT (question above) must also provide annual data? Different to FAO land use inventories with 5-year update cycles?

Lines 475 to 503: Some deception here? None of the five sources provides actual 2017 data? Instead, authors have, by necessity, applied source-specific extrapolations to each of the five sources to reach 2017. So in fact we really only have valid bottom-up data to 2015 (perhaps, with a stretch, to 2016 for some sources) but for reasons we don't learn, authors advertise (e.g. through the title) coverage through 2017. I applaud the desire to update from Saunier 2016 but the definition of "the most recent available year" (e.g. on line 302) refers only to atmospheric concentrations but falls short on all bottom-up source data? Authors should make explicit mention of this discontinuity? Authors, knowing the UN systems, live with these inherent reporting delays but readers/ users may not understand? Earlier and later the authors refer to differences between top-down total CH4 emissions numbers and bottom-up numbers. This extrapolation represents an additional uncertainty factor? This discontinuity, e.g. 2017 data in some cases but extrapolations from 2015 or earlier in many other cases, impacts Figure 3?

Also, Table 2, biogeochemical models for wetland CH4 emission all "were performed for the whole period 2000-2017" (Table 2) but forced with invariant wetland surface area maps? We know that wetland surface areas have changes substantially but do the data show changes specific to 2008-2017?

Line 553: "agreement Despite the offset of the SSP scenarios compared to the recent inventories" Need a period between agreement and Despite. What does 'offset' mean in this context? Divergence? Divergence from SSP2.6 or 3.4 but tracking 8.5? Not clear what authors intend with this sentence.

Line 560: "transport" here but "transportation" already mention above (line 557). In one case you mean the act or process of transportation and in the other case the actual transport of fossil fuel products (e.g. via pipeline)? For readers who do not know IPCC categories, we need clarification here.

Line 606, 607: "applying "Tier 1" approaches for coal mine emissions is not accurate enough" ?are not accurate enough?

Line 649: "and, could lead to". Delete the comma?

Lines 665 to 669: studies find emission underestimates due to inability to correctly account for CH4 from fracking? Otherwise, why does reader find this particular sentence in this particular location?

Line 672: "abnormal operating conditions" abnormal = fracking? Abnormal = fracking done not according to regulations or best practices? Abnormal = some other type of fossil fuel extraction?

Lines 675 to 683: how does this section contribute to the overall or fossil-fuel-specific budget calculations? The sentence following (lines 684 to 685) provides a short sufficient summary which could replace much of the prior text?

Lines 724, 725: "the volatile solids component" This reader does not know the chemical meaning of the term 'volatile solid'?

Lines 723 to 736: highly redundant. Could reduce to the sentence on lines 727 to 729: "Ambient temperature, moisture, and manure storage or residency time affect the amount of CH4 produced because they influence the growth of the microorganisms responsible for CH4
formation". For budget purposes, we don't need more than that?

Line 751: Start here while deleting the prior paragraph?

Line 775: "the work of (Carlson et al., 2016)" you mean 'the work of Carlson et al. (2016). Please take care with punctuation, as proofreaders will not know what you want.

Line 779: "northward shift of rice cultivation". Unless cultivation practices change with latitude, a latitudinal shift will not necessarily change or reduce emissions?

Lines 837 to 905: way too much information here! Please only include what we need to know to understand this/these budget estimates. Also we seem to have lost the useful end-of-section summary of what this budget effort concludes for this source?

Line 979: by this point the reader has encountered anaerobic conditions many times (waste water ponds, rice paddies, bovine guts, etc.). Do we really need a new definition and explanation of anaerobic metabolism at every point. Reduce, s.v.p.

Line 980: "limit oxygen availability and creates suitable redox" create, not creates?

Line 987: are these processes diffusive or advective? This reader does not understand the phrase: "molecular diffusion limited advection"

Line 992: "transportation and are further regulated by" What are 'further regulated'? The land-surface. models? The CH4 emissions? The model parameters? Very confusing, please re-write.

Line 995: "(Supplementary Material, Melton et al. (2013); Bohn et al., 2015)" Again, inconsistent and incorrect use of parentheses. Relying on bibliometrics software from various contributors evidently does not work. I have pointed out a few such errors, skipped over many, and anticipate many to come. A waste of this reviewer's time to point them all out. Assign one of your co-authors to search for, evaluate and correct every parenthesis. Authors must take this responsibility. Proofreaders will not find nor know how to correct all of them.

Line 1008: "a monthly global wetland area dataset" Does wetland surface area vary only seasonally or does it also evolve with time over several years?

Line 1056 to 1074: a hard-nosed editor could re-write this entire section in two sentences or perhaps even one sentence. How much of this is relevant to this current version of a CH4 budget?

Line 1107: "south of 60°N) with. Tan and Zhuang" something wrong with punctuation here?

Lines 1165 to 1174: here we get expert opinion on what next and what needed for lakes and rivers. Good! But doesn't this section belong elsewhere/later?

Line 1234: why specify exact (and proprietary) GIS software here? Same scale up would have worked in any GIS environment?

Line 1341: "which allows very little methane; even from established". Punctuation error?

Line 1376, 1377: Authors already defined marine clathrates earlier (line 392). Reader does not need redundant definitions?

Line 1398: In earlier sections readers found summaries first, then itemization of individual components following. Here we get the opposite: components first followed by summary. Chose one or the other but do not confuse us with both?

Line 1455: Many references in this section appear as (non-functional) hot links. Many other references do not. Please fix this!

Line 1511: Finding the (strange) unit here of "molec" and having earlier seen reference to mole fractions (strictly, in an SI units sense moles per mole but more casually expressed as ppb), one wishes the authors had provided a conversion table as in the global carbon budget. They have done a very good job of keep flux units consistent as Tg CH4 yr-1, but they could help readers by cross-referencing the various concentration units.

Lines 1498 to 1558: do we need an extensive discussion of OH concentrations, reactions, hemispheric distributions and lifetimes to understand the magnitude and validity of the present CH4 budget? This review thinks not. One could summarize this entire section in two or three lines?

Line 1526: Pinatubo eruption represented a climate variation? Hardly.

Line 1530: "enhances CH4 consumption"? You mean 'enhances the CH4 sink' or 'enhances CH4 oxidation?

Line 1548: "consistent, albeit lower, than the value deduced from"? Very sloppy wording, leaves a reader guessing at what you mean here.

Lines 1577 to 1579: This sentence summarizes the entire stratospheric dynamics and photochemistry section. Why do we need all the rest of this?

Line 1590: Should CH4 sink terms use a different sign than source terms? I suspect carbon budget has already wrestled with this flux direction issue. What did they decide? Does GCP adhere to a consistent sink/source nomenclature convention?

Line 1606: "the KIE approach" KIE = kinetic isotope effect, but you have't defined it yet?

Line 1679:  systematic atmospheric?

Line 1695: A couple TCCON sites, perhaps Australia or NZ or both, have published CH4 data in ESSD?

Lines 1699 to 1711, other data. If not useful or used for this global CH4 budget, why mention it/them?

Lines 1713 to 1716: a reader just saw this identical list a few lines earlier. We don't need to see two mentions of the same information?

Line 1739: And you show $G_{ATM}$ for the previous version of global CH4 budget (e.g. in Table 3) but not for this version. Why not?

Line 1745: By definition, a growth rate can not be both positive and stable. Instead of stable, you mean consistent or persistent?

Line 1751 to 1755: Here a reader finds the recent growth values, nicely documented with appropriate uncertainties. Why do these values not also appear in Table 3?

Lines 1756 and following, Satellite data. Important to show command of this data for for budget purposes but way way too much information here. Why write four or five sentences of description about a source that you - for valid reasons - can not use? Is this a budget discussion or narrative about the history of SWIR sensors? You define SWIR at least twice in this section. Compilation - good - but not followed by tight budget-focussed editing - bad.

Line 1778: "retrieval approaches, Proxy and Full Physics. The proxy method retrieves" ???. A former sub-heading now buried in this text? We do not need all this information?

Line 1792: "only inversions using GOSAT retrievals are used." After nearly 40 lines of text a reader finally finds this short conclusion. Please can someone apply a sharp red pencil to this manuscript - focus on the CH4 budget!

Line 1799: "ensemble of inversions gathering various chemistry transport" Why? To reduce uncertainties or provide independent validation for the CH4 budget terms (e.g.

use top-down to constrain bottom-up) or to ensure that a long list of inversion modelers get their work listed and recognized via this global CH4 budget paper. I apologize if this reviewer's questions seem uniformed, irrelevant or even rude. But pity the poor readers / users. They want to use this product to understand CH4 budgets! Instead, they confront pages of tangential descriptions replete with (obscured by) more and more acronyms. Seriously, am I reading the CH4 budget or a narrative recount of every CH4 activity that has occurred?

Lines 1801, 1802: "assume that this model range is sufficient to cover the range of transport model errors in the estimate of methane fluxes". You 'assume'? This assembly of world experts on CH4 uses an ensemble approach because they can't (or won't) distinguish useful from not useful inversions? Again, does this approach help quality of the budget? If so, quantify that. Does it help users? If so, tell us. Show us! Did the inversions prove useful in the previous CH4 budget? If so, how? Although these authors claim (lines 1807, 1808) intention to do flux assessment rather than a MIP, the section in its present description looks, smells and sounds like a MIP to this reader.

Lines 1844 to 1846: Here, deep in the manuscript where few readers will notice, we finally read a justification of costs and benefits of time span and time lag of the current product. Good discussion! But we should have read it in the introduction? It sets out temporal goals that guide the entire product!

Lines 1850 and following: This is a good useful orderly section! But: a) it repeats a lot of material from earlier; and b) a reader needed to work through - or, more likely, skip through - an unusually large amount of text to get here. Authors could argue that this section depends on all the prior detail. This reader/reviewer asks authors to consider how to make the user's 'life' easier rather than putting priority on including all details. Details can and should go in individual research papers. The budget should not replay, only summarize?

Line 1882: Introductory sentence implies treatment by the five source categories but discussion/summary that follows treats natural vs anthropogenic (with wetlands called out separately) but does not follow the five-category organization that you used earlier, for example. Revise this opening sentence to better convey what will follow?

Line 1914; "study derive significant changes in wetland emissions" derive? I think you mean 'identify' or 'reveal' or 'point to'?

Line 1919: We just read (lines 1875) that uncertainty ranges for, e.g. ocean, termites, geological sources have reduced (improved) in this version, but here the authors point to those same specific sources as plausible reasons for the top-down bottom-up discontinuity? Uncertainty goes down but discrepancy goes up? Mathematically I think that works but this discussion implies a substantial weakness in our understanding. This reader particularly wonders about permafrost in this regard (see opening comments above), not so much about the permafrost numbers but more about whether assigning

permafrost emissions as natural emissions doesn't presently or won't in the future exacerbate this discrepancy?

Line 1930: "Improved area estimates …" By this point reader has seen this recommendation at least twice. It really rather belongs in a summary or future work section, not here? This entire paragraph repeats earlier text and belongs instead in a summary section?

Line 1948: "which is about 30% (23%) of global methane" What does the 23% indicate here? About 30% but actually 23%? 23% of global but 30% of anthropogenic? Some plus minus 23% uncertainty? I find the answer to my question in line 1949, 30% of top-down but only 23% of bottom-up. Readers should encounter definition before data, not the other way around?

Line 1958: "partition of methane emissions between wetlands and freshwater systems should still receive a high priority" this includes the thawing permafrost freshwater confusion?

Lines 1982, 1983: Valid caution but a reader should have encountered it earlier?

Lines 1997 to 1999: Very confusing here, again the source/sink flux direction problem. The text says 'model inputs somewhat higher' but what you mean is some models show a higher value for total sink because they include a larger land sink term? For a known atmospheric CH4 concentration, that larger sink term must lead to a correspondingly larger source term! Here, we read about a larger central top-down estimate. Authors intend this phrase to refer to the larger sink term, or to a larger total CH4 emissions term? The authors, no doubt, know various sink, source and net terms, but here they fail to express themselves in clear language, leaving readers therefore confused. This statement occurs within the section on CH4 sinks, so one wonders …

Line 2006: "chemical lifetime and to narrow it down in" Authors have addressed the issue of lifetime many times and in great detail prior to this statement. A reader knows at this point a lifetime of 9 years, plus or minus perhaps 0.5 years. How would refinement of this lifetime calculation improve the overall budget calculation, much less mitigation planning? E.g. does a lifetime of 8 years or 10 years really make a difference at this point? This reviewer suspects not. This section represents an(other) example of everyone's scientific priority gaining equal weight in the compiled budget. This (small, to this reviewer) remaining uncertainty has larger overall priority than fixing the distribution and extent uncertainties of freshwater (also mentioned numerous times)? Organizationally, this entire section (lines 2005 to 2012) belongs in a summary / future work section.

Line 2024: A reader has now seen this statement at least four times?

Line 2033: Readers already know, at least twice, why SCIAMACHY data not used.

Lines 2024 to 2053: Good knowledgeable expert discussion of impact of satellite vs in situ data on inversions and emissions. But, of what relevance, in specific plus/minus Tg CH4 terms, to the global CH4 budget? A reader sees "clearly show" but at the same time "not systematically consistent". I almost used a profane shorthand here: what do the authors want readers to learn, and how does it matter to the CH4 budget? No clue in this section.  More "further investigation" ….

Line 2082: Section 6, developments, missing elements, remaining uncertainties. Important section. Readers will not disagree with topics raised here. But the section reads more as a recitation of the previous CH4 budget issues while omitting issues raised in this version. Ebullition, for example, mentioned repeatedly in the text, does not emerge here. The entire tension between spatial uncertainties and chemical uncertainties, a theme of this paper, does not emerge here, at least not in those recognizable terms. Pages of text on the need to improve distribution or lifetime of OH, but only one faint mention here, that basically repeats what we already read?  We could get a list: previous but now resolved, previous but not yet resolved, new. Here a reader finds no sense of priority, just a large wish list not much changed from the prior budget. Use the GWP of CH4, find the largest or most tractable uncertainties, convert those to climate urgency: where should we put our efforts? If we know new satellites have recently or will soon come on line, what do we need for validation? Where will that data have most impact in this budget?  More, more, more. More systematic. More integrated. Just another wish list. After all the work to compile this information, the world-wide experts then throw up their hands - they don't know what they need next? They need everything? Not helpful.

After working to assimilate all the details provided to this point, this reader basically glazed over this section. It seemed only vaguely related to what we just learned, a white-paper level of what we should do to better understand CH4 but not well connected to all that we just read about efforts and information need to assemble this budget.

Line 2089: "Knox and al., 2019)" ??

Lines 2209 to 2219: a short clear paragraph of budget outcomes - thank you! Notable, unfortunately, because it differs so much in style, brevity and clarity from most of the preceding text.

Lines 2226 to 2231: weaknesses and cautions about extracting regional information from satellites. Important, clear, useful. But link this to specific recommendations in Section 6, where now we just find a wish list for more of everything?

Line 2237: a "clear priority"! Finally! First one? Does permafrost fit into this inland freshwater priority?

Line 2242: "also place importance in" Should a reader conclude here that authors consider improvements in OH distributions as second in importance to the freshwater extent problem?

Line 2246 to 2253: Short clear paragraph, contains a bit of the unhelpful 'need everything' philosophy, but if we have these topics listed here we could get rid of much of Section 6?

Lines 2254 to 2257: Good statement, good motivation, good summary of forward plans, but repeats almost verbatim what readers saw earlier (lines 1844 to 1846)? Some repeat expected in a good summary but not clear here why we need this detail twice?

Line 2268: "a negative contribution (from biomass burning" You really do not want punctuation errors in such an important paragraph?

Line 2299 and following, Acknowledgements. Interesting section. I think global carbon budget does something like this but through a table instead? Nothing wrong with this approach but text includes several (perhaps many) punctuation and tense errors. Someone of the author team should read this carefully?

Page 126, 127, Table 3. Cells need reformatting, presumably will happen during typesetting. Authors will need to check text wraps. Repeat the question raised above about why no $G_{ATM}$ rate information for current version(s)?

---

## Referee Comment (RC2) · Michael Prather (Referee) · 6 Dec 2019

Rev of ESSD-2019-128

This is a very well written and thorough review of the methane budget. The abstract and introduction read easily and layout the scope of the work. The remaining 2000 lines are a bit harder to get through, but this paper is meant as a reference work and not a "beach read." For a paper on methane, it is great to see Ehhalt's original work as one of the prime references. I have two major suggestions and then several minor/editorial comments on my read-through.

**1 As a reference work, it would be very useful to have a Table of Contents up front. The TOC should be as detailed as possible (e.g., add another level: 3.3.2.1. Lakes and Ponds) and possibly include a list of key words for each section. This would greatly help those looking for specific discussions.**

**2 The Section 3.3.5 on CH4 Lifetime contains some serious errors in understanding. I am not sure how to fix it, but the authors should think about the purpose and what they want to get across. See details below.**

L217: The Prather 10% is only 68% confidence interval, perhaps if the Saunois 15% is 90% confidence, they are consistent.

L225: Probably more important here is not 'credible scenarios,' but credible mitigation strategies.

L231: do you want to mention the Paris Accord's "stocktaking"?

L243: This is a slight mis-direct. It is not the lack of some direct observations, but by my calculation, the number of OH "measurements" needed to integrate the loss of methane (1 km x 100 m parcels, every $\frac{1}{2}$ hour (clouds)) is of order 2x1014 per yr. I would put this as an impossible task without some modeling and other tricks.

L290: Here and later, you cannot reference a work in prep as a real reference. You can describe ongoing work by Staevert and colleagues that will follow up this paper and come up with a shorthand notation for this work.

L318-324: If you are being precise, please define where the edges of the 1-degree grid fall, one the 0.0, or 0.5, or? Also does what you describe mean that coastal boxes (<100% Land and <100% Ocean) have no emissions – you should definitely want to warn people as it will look funny when plotted. Also what do you do about large lakes (resolved at 1 degree)?

L383: This discussion of natural vs. anthropogenic intrigued me. We have a long history of trying to break emissions cleanly in these two camps. With natural often being ignored in the scenarios (RCP, SRES), such that in the IPCC SAR & TAR, I had to make up the natural sources to ensure a consistent budget and initial trajectory for the future scenarios. The other problem is that attribution is still not an exact science and thus declaring something like wildfires or wetland loss as "anthropogenic" is not trivial.

Let me propose that the sources be split into "pre-agricultural" and new "anthropogenic" types of sources (such as ag or industry). This allows us to work our best science on how wetland methane emissions have changed, without trying to ascribe cause. The changes in the "pre-ag" sources probably can be attributed in many cases to direct human intervention.

L529: can you make this simpler to read: "perhaps because AMAP analysed data from a wider range of inventories and projections, plus it was referenced to one year only (2005) rather than averaged over a decade, as done here."

L552: "realized" in what? the scenarios or real life?

L660ff: In terms of shale gas emissions, is the DeCarlo work on the Marcellus Shale consistent with these? Goetz, J. D., et al., Analysis of local-scale background concentrations of methane and other gas-phase species in the Marcellus Shale, Elementa Science of the Anthropocene, 1–20, doi:10.1525/journal.elementa 182, 2017.

L1026: Is this really true? If we have all positive values but some far outliers, then you would reject a positive flux because the std dev exceeded the mean?

L1174: I am unsure what the "up scaling issues" means.

L1205ff: This sentence on the Petrenko work is a bit perjorative and full of inuendo ("If it is correct", "which is questionable"). There is nothing obviously wrong with the work, unless the 14C community is worried about it, and if so state why. The sentence on L1208 about the discrepancy is fine.

L1233: I am not sure what 'termite' as a unit is?

L1500-1507: I am a little worried about this section, but have no simple fix to recommend. (1) the effect of $CH_4$ oxidation on HOx depends on the pathway for $H_2CO$, if it photolyzes on one path you get 2 HOx, otherwise, if OH is high, then OH-destruction it yields 0 net HOx. NOx plays a role in this, but HOx levels are also very important. (2) The short lifetime of OH should go back to Levy or Logan, not a 2004 reference.

(3) "estimate"? One can estimate OH from models without observations. (4) Many, many OH measurements are made in the free troposphere by Brune and others. I am beginning to think it best to stop at 'compounds.' on L1501 and jump to L1507 'Following...'

L1511: I think you have to recognize here that mass weighted OH concentrations are NOT a good measure of the methane loss. Since this is a methane paper, you should focus on the average methane loss rate from these models. I know this is a hobby of mine, but please look at the table in my 1990 paper (Prather & Spivakovsky, Tropospheric OH and the lifetimes of HCFCs, JGR: 95, 18723-18729, 1990), also reproduced by Mark Lawrence in 2000, and one can see that the average OH varies by 20-40% depending on how it is weighted. What is relevant here is when OH is weighted by mass and exp(-1800/T). In fact, the OH values do not tell you the methane lifetime unless you know the temperature weighting.

L1527-1560: The Holmes et al paper (2013, Future methane, hydroxyl, and their uncertainties: key climate and emission parameters for future predictions, Atmos. Chem. Phys., 13, 285–302, doi:10.5194/acp-13-285-2013) really addresses recent OH variability and should be included with this discussion. (Sorry to push my own papers again, but it is a balanced survey of OH variations from cause [all those listed] to methylchloroform-derived variability.)

L1579: It is unclear if your 60 Tg is irreversible mixing (i.e., loss in strat) or the cross tropopause flux. The amount of CH4 entering the stratosphere is 10x or more larger than this number, most of which crosses into the lower stratosphere inn the sub-tropics and is then transported into the troposphere with little chemistry. I would drop this whole sentence as it does not say much.

L1590: We have a much more accurate measure of the stratospheric loss from the Plumb & Ko relationship and the observed N2O-CH4 tracer slopes in the lower stratosphere. I do not know when this was last revisited, however.

L1654ff: This section 3.3.5 on CH4 Lifetime has some major problems. Excuse my didactic diversion here. "Lifetime" is a budgetary number since the standards and notation in the 1995/96 IPCC SAR through AR5. It is the burden divided any loss rate. Note that the total burden in the system must be used to take advantage of adding inverse lifetimes. Hence you will see the "lifetime of CH4 against trop OH loss" is the Tg (including stratosphere) divided by the Tg/yr lost to OH in the troposphere. If you use this definition carefully then inverse lifetimes can added and we can think of the lifetime due to stratospheric loss being about 160 yr as is recommended here. The OH lifetime should be noted and taken from the MIPs, it should be about 11 yr. "Perturbation time, response time, e-fold time" are used to define the decay of a perturbation about any atmospheric state (again, steady-state does not matter for these definitions). Since 1994, for CH4 we have known that this time scale is about 1.4 times the total lifetime because of chemical feedbacks whereby CH4 suppresses OH. This is well known, consistently modeled and increases all the integrated impacts by this factor. With a budget lifetime of 9.1 yr, the time scale for CH4 perturbations is about 12 yr. Any perturbation to a chemically reactive species will excite a large number of chemical modes – each with its own pattern of species and its own decay time. Hence CO is an indirect greenhouse gas because it generates a CH4 perturbation that decays with a 12-yr e-fold. "Steady state" is not required for these numbers, but the steady-state lifetime does have some magic properties. It is the effective average over all the different chemical modes (their amplitudes and time scales) excited by a perturbation.

The rest of the lifetime numbers from the recent MIPS look to be OK, but make sure that these are calculated using the full burden.

L1708: The references to Rigby 2017 and Turner 2017 really should include also the accompanying Prather and Holmes paper (Over-explaining or under-explaining methane's role in climate change, PNAS 114(21) 5324-5326, doi: 10.1073/pnas.1704884114, 2017) that points out the fundamental error in modeling CH4 perturbations for both papers (i.e., they did not include the chemical feedbacks).

L1750. Based on notes above, I might expect the time scale here to be 12 yr rather than 9 yr, but I think it is probably close enough for this simple analysis.

L1795ff: This is a good discussion of results, but I wonder how much these inversions depend on the a priori's for lifetime, etc. I do not see how you can cover this here, but can you comment? OK, see L1988.

L1988: Yes, if everyone uses the same trop OH, then the budget total is pretty well fixed (except for T biases, and maybe ITCZ gradients).

L2154ff & L2200ff: Agreed. We continue to produce models with the same range in CH4 OH-lifetime and no means of differentiating them. I would not emphasize the vertical as our model failings include horizontal as well. We need to develop some key observational tests that reflect the reactivity of the air parcels. To start, we need to run the MIPS with some more focused diagnostics that tell us why the models CH4 lifetimes are so different. Even more interesting is that the historical trends and the scenario projections often go in different directions across the model ensemble. And we should not forget that our goal is accurately simulate CH4 loss, not just get OH correct.

L2273: This is where you might want to look at the Holmes (2013 ACP) effort to model OH variability and match it to CH3CCL3 observations and CH4 lifetimes. It is multi-model and more physically based than these inversions.

I am glad you got this paper out. It is valuable and the issues above can be easily addressed. Congrats,

Michael Prather

---

## Author Comment (AC1) · 2 Mar 2020

**The Global Methane Budget: 2000-2017**

Saunois et al., ESSDD, 2019

**Detailed Response to Anonymous Referee #1**

We acknowledge the referee for the time spent on reading and commenting on the paper. We thank him for his useful corrections and suggestions on the paper, which have helped clarifying and improving the manuscript. Below are the responses to his comments (in italics, blue). Changes in the text follow each response in bold font.

In particular, the text has been shortened by 5 pages (8%), from 61 to 56 pages in the ESSDD format.

**General comments**

*Thank authors for massive compilation effort. Important product! One hopes a large community will see and use. Good product for ESSD.*

We thank the reviewer for the compliment.

*Push data out to 2008-2017 decade? Various valid reasons for this but scattered somewhat randomly through the narrative. Pull all of this information into a short clear statement very near the top of the manuscript to explain time span and time lag?*

We included a subsection in the methodology – Sect.2.2 "Period of the budget and availability of data" to justify the analysis period to 2017. In this Section, we explain:

- Why some inventories were extrapolated to push the budget to 2017
- That surface models were runs over 2000-2017, covering the full period of the budget using dynamical wetland extent
- That inversions were also run until beginning 2018 to cover 2000-2017

*I and many others will want to use this in classroom. We might assign paper as background reading. Students will loudly object - too long! This reviewer agrees. Too long by a factor of two, at least. Almost long enough to need an executive summary? See suggestion below.*

Indeed, the manuscript is a long one. We reduced its length following the reviewer suggestions, and removed excessive details and repetitions. **A table of content** has also been added prior the introduction to help the reader to easily find his section of interest of the layout of this - long – paper.

*In present organization, permafrost emissions sit within natural emissions. Technically, and increasingly so, permafrost emissions forced by permafrost thaw should instead account in the anthropogenic category? But if you move terrestrial permafrost sources then you also should move sub-marine permafrost emissions, boreal plant sinks or sources, etc. No easy solution but the current location will prove increasingly problematic.*

Permafrost emissions sit within natural emissions. This is also the case for reservoirs, managed wetlands, or wetland changes that could be attributed to human activities. Attributing emissions to either "natural" or "anthropogenic category is challenging. Also, some – many – "natural" emissions of methane are/will be perturbed by anthropogenic climate change. In our study, we define *anthropogenic* as emissions caused by direct human activities since pre-industrial/pre-agricultural time and *natural* as all other sources, some of them being perturbed by climate change.

We agree (and the second reviewer as well) that some clear definition need to be given to what is called here "natural "and "anthropogenic ". We addressed this by splitting the former Section 2.3

(Definition of regions and source category), in two sections and creating a Section 2.4 "Definition of source categories". In particular we moved a paragraph from the beginning of Section 3 to Section 2.3 to better define the frontier between "anthropogenic" and "natural" sources.

After a short description of the different processes inducing methane emissions, the text shows the following definition of natural versus anthropogenic:

"**In the following, we present the different methane sources classified from anthropogenic or natural origin. "Natural sources" refer to pre-agricultural emissions even if they are perturbed by anthropogenic climate change, and "anthropogenic sources" are caused by direct human activities since pre-industrial/pre-agricultural time (3000-2000 BP, Nakasawa et al. (1993)) including agriculture, waste management and fossil fuel related activities. Natural emissions are split between "wetland" and "other natural" emissions (e.g., non-wetland inland waters, wild animals, termites, land geological sources, oceanic geological and biogenic sources, and terrestrial permafrost). Anthropogenic emissions contain: "agriculture and waste emissions", "fossil fuel emissions", "biomass and biofuel burning emissions", assuming that all types of fires cause anthropogenic sources, although they are partly of natural origin (Fig. 6, see also Table 3 and 6).**

**Our definition of natural/anthropogenic sources does not correspond exactly to the definition used by UNFCCC following the IPCC guidelines (IPCC, 2006), where, for pragmatic reasons, all emissions from managed land are reported as anthropogenic, which is not the case here. For instance, we consider all wetlands as natural emissions, despite some wetlands being managed and their emissions being partly reported in UNFCCC national communications. The human induced perturbation of climate, atmospheric CO2, and nitrogen and sulfur deposition may cause changes in the sources we classified as natural. Following our definition, emissions from wetlands, inland water or thawing permafrost will be accountable in "natural" emissions, even though, we acknowledge that climate change – a human perturbation – may cause increasing emissions from these sources. Methane emissions from reservoirs are considered as natural even though reservoirs are human man-made, and since the 2019 refinement to the IPCC guidelines (IPCC, 2006; IPCC, 2019) emissions from reservoirs and other flooded lands are considered anthropogenic by UNFCCC.**
"

*Whether or not the authors intend, this reviewer concludes that we face a larger and more urgent geospatial rather chemical challenge. In short terms, we need to fix the wetland inland freshwater distribution and extent uncertainties as our highest priority, with understanding and quantification of atmospheric OH distributions, reactions and impacts as probably a second priority. If authors intend these priorities, they should state them explicitly? They can always revise in a subsequent version. If this reviewer developed a wrong impression, then authors need to reconsider information they presented and how.*

The authors agree with the priorities. However, addressing them involves different communities. Accurate split between wetland and freshwater extent would allow a better estimate of the bottom up budget; quantifying OH burden and distributions would impact the top-down estimate of the methane budget and the growth rate projected from bottom up estimates. Working on both issues should help reconciling bottom-up and top-down estimates. The last section – Section 6, has been revised to better highlights priorities and progresses needed on the methane budget.

*Topical editor suggested that I do a full review. Finding the track-changes version somewhere between daunting and impenetrable, I agreed. After plowing through so much material, however, I almost wish I had tried instead to follow the track changes version. Fundamentally, this is way too much material. Removal of the regional details to separate papers helps a lot, but the present version still represents way too much material. You want strong interest and to serve many users, but at present the too-long,*

*too-disorganized, too-detailed text represents a severe barrier to almost every reader and potential user. Amazing compilation effort, but you need to present it after and on the basis of an equally amazing editorial effort. The latter effort remains sadly missing.*

The authors thank the reviewer for his honesty. We agree that many repetitions occurred throughout the text and some parts suffer from too many details. We have followed the reviewer's suggestions to improve the organization of the manuscript and shorten it by 8%.

*Specific comments - long list and I know I missed some issues*
*Line 304: "the relatively small and variable number of studies" Do the authors mean a small number of studies with variable results? A variable number of studies covering small regions or sectors? Confusing.*

This has been rephrased as follow: **"considering that the number of studies is relatively small for many individual source or sink estimates »**

*Line 317: "processed in the same way" Does this phrase mean that monthly and annual emission data, from any source, were subjected to the identical spatial 1x1 gridding? But some source data exists at higher spatial resolution? E.g. would need some form of disaggregation? Later (lines 320, 321) a reader finds that calculations of monthly and annual means came from (came after?) gridding. Confusion between spatial and temporal averaging here?*

The files were provided at monthly or yearly scale. First a spatial re-gridding was applied to have all fluxes on 1°x1° grid at the original temporal scale. Then, annual and decadal means were computed. It has been rephrased as follows:

**"Gridded emissions from atmospheric inversions, land-surface models for wetland or biomass burning were provided at the monthly scale. Emissions from anthropogenic inventories are usually available as yearly estimates. These monthly or yearly fluxes were provided on a 1°x1° grid or re-gridded to 1°x1°, then converted into units of Tg $CH_4$ per grid cell. Inversions with a resolution coarser than 1° were downscaled to 1° by each modeling group. Land fluxes in coastal pixels were reallocated to the neighbouring land pixel according to our 1° land-sea mask, and vice-versa for ocean fluxes. Annual and decadal means used for this study were computed from the monthly or yearly gridded 1°x1° maps."**

*Line 332: "the regional budget presented in (Stavert, 2019)" should read as 'the regional budget presented by Stavert (2019)'? Proofreaders will not know what to do with these punctuation errors so authors must take care to express them correctly.*

This has been corrected. A thorough reading has been performed to check the punctuation and hyperlinks related to each citation.

*Line 336: In the version I read ("the TransCom inter-comparison map (Gurney et al., 2004)", the Gurney et al reference shows as a non-functional hot link, different to all other references?*

This has been corrected. A thorough verifications has been performed to check the punctuation and hyperlink related to each citation. Hyperlinks were all removed.

*Lines 358, 35: again some references as apparently hot links, others in standard (expected) format. Why?*

This is due to the use of a former list of citation from the previous version completed manually with the new references instead of using the citation software. We have corrected this issue.

*Line 359: "most of the inverse system the" Instead: inverse systems?*

This has been corrected.

*Line 373: "Methane sources and sinks" Because the (long and detailed) section 3 deals entirely with bottom-up (e.g. line 374) emissions, and because section 4 carries the explicit title of top-down (e.g. at line 1677: "Atmospheric observations and top-down inversions"), shouldn't the title of section 3 read as 'Bottom-up sources and sinks'? (By this point one hopes that most readers/users know that the discussion refers to CH4.) If section 3 does NOT deal solely with bottom-up sources and processes (like most readers, this reviewer loses track of bottom-up vs top-down deep in section 3, e.g. in the section on CH4 lifetimes) then the reference to bottom-up at line at 374, 375 needs to change?*

Indeed Section 3 present the sources and sinks from the process perspectives and provide estimate from Bottom-up approaches. The title of Section has been changed to **"Methane sources and sinks: bottom-up estimates".** We keep methane in the title of section 3 but removed it in the sub-section. See the final Table of Content.

*Line 395 to 400: good section on definitions used here vs those used by IPCC. This reviewer agrees with these choices but, more important, thanks authors for explaining the differences.*

We thank the reviewer for the comment. The definition of the categories and how they are named here ("natural" versus "anthropogenic") are critical and challenging in our study. For clarity, we moved the whole paragraph (previously L 380 – L 400) to subsection 2.4 "Definition of source categories". Some explanations were added to explain that other emissions such as "permafrost" are classified in "natural", see response above.

*Line 436: FAOSTAT is annual or every 5 years? Annual, evidently, from Table 1?*

Yes. FAOSTAT provide annual values. This has been added in the text

*Line 457: "Although, country emissions" Delete the comma?*

This has been corrected.

*Line 472: Here the reader finds FAO-CH4 at annual resolution so FAOSTAT (question above) must also provide annual data? Different to FAO land use inventories with 5-year update cycles?*

Yes FAOSTAT, named here FAO-CH4, is at annual resolution.

*Lines 475 to 503: Some deception here? None of the five sources provides actual 2017 data? Instead, authors have, by necessity, applied source-specific extrapolations to each of the five sources to reach 2017. So in fact we really only have valid bottom-up data to 2015 (perhaps, with a stretch, to 2016 for some sources) but for reasons we don't learn, authors advertise (e.g. through the title) coverage through 2017. I applaud the desire to update from Saunier 2016 but the definition of "the most recent available year" (e.g. on line 302) refers only to atmospheric concentrations but falls short on all bottom-up source data? Authors should make explicit mention of this discontinuity? Authors, knowing the UN systems, live with these inherent reporting delays but readers/ users may not understand? Earlier and later the authors refer to differences between top-down total CH4 emissions numbers and bottom-up numbers. This extrapolation represents an additional uncertainty factor? This discontinuity, e.g. 2017 data in some cases but extrapolations from 2015 or earlier in many other cases, impacts Figure 3?*

Indeed, the delay in updating inventories may not be well-known by the reader. We acknowledge that extrapolation of inventories is mentioned a bit late in text. We have added few sentences in the Methodology section to make this clearer. However, the extrapolations based on FAO and BP statistics are the best that could be done, waiting for updated inventories. EDGAR uses FAO data as activity data for agriculture, and data from IEA for energy related emissions. The EDGAR version v5.0 has been recently released but stops in 2015 for $CH_4$.

Besides inventories, top down and biogeochemical models have been run until 2017, and the most recent literature is used for other bottom-up sources and sinks. We have added the following text in the methodology, Sect.2.2 "Period of the budget and availability of data": **"The surface land models were run over the full period 2000-2017 using dynamical wetland areas (Sect. 3.2.1).**

**For the top-down estimates, we use atmospheric inversions with atmospheric measurements covering 2000-2017...."**

*Also, Table 2, biogeochemical models for wetland CH4 emission all "were performed for the whole period 2000-2017" (Table 2) but forced with invariant wetland surface area maps? We know that wetland surface areas have changes substantially but do the data show changes specific to 2008-2017?*

Biogeochemical models have been run over 2000-2017 using dynamic wetland surface area, that include satellite-derived wetland areas, as stated in Section 3.2.1." **The WAD2M dataset provides monthly global wetland areas over 2000-2017**."

*Line 553: "agreement Despite the offset of the SSP scenarios compared to the recent inventories" Need a period between agreement and Despite. What does 'offset' mean in this context? Divergence? Divergence from SSP2.6 or 3.4 but tracking 8.5? Not clear what authors intend with this sentence.*

Indeed, this was not clear, we intended to say that discrepancies in absolute estimates between inventories and SSPs scenarios in the year 2005 should be left aside in order to focus on differences of trends. This has been rephrased to: **"...agreement. In the future, it will be important to monitor trends from year 2015 estimated in inventories and compare them to SSP scenarios'."**

*Line 560: "transport" here but "transportation" already mention above (line 557). In one case you mean the act or process of transportation and in the other case the actual transport of fossil fuel products (e.g. via pipeline)? For readers who do not know IPCC categories, we need clarification here.*

The term "transport" has been changed to "**road or non-road transport",** which should be more obvious.

*Line 606, 607: "applying "Tier 1" approaches for coal mine emissions is not accurate enough" ?are not accurate enough?*

The subject here is "applying", as gerund, so the verb here is "is".

*Line 649: "and, could lead to". Delete the comma?*

This has been corrected

*Lines 665 to 669: studies find emission underestimates due to inability to correctly account for CH4 from fracking? Otherwise, why does reader find this particular sentence in this particular location?*

Indeed, this section was mishandled, and sentences on oil and gas general activities were attributed to the "shale gas" section. This section has been revised by:
- Deleting "conventional and "shale" splitting
- Reorganizing the text, to make clear that most of the discussion -including the underestimation in the inventories- concerns all type of gas production.
- Mentioning shale gas for two aspects: 1. The increase in shale gas production in the US 2. Potential differences in emissions factor between conventional and shale gas.

*Line 672: "abnormal operating conditions" abnormal = fracking? Abnormal = fracking done not according to regulations or best practices? Abnormal = some other type of fossil fuel extraction?*

While revising the text of this section (see response above), some sentences have been deleted to reduce the section. This one in particular.

*Lines 675 to 683: how does this section contribute to the overall or fossil -fuel-specific budget calculations? The sentence following (lines 684 to 685) provides a short sufficient summary which could replace much of the prior text?*

The text has been revised and shorten as explained above.

*Lines 724, 725: "the volatile solids component" This reader does not know the chemical meaning of the term 'volatile solid'?*

A volatile solid is a substance that can easily transform from its solid phase to its vapor phase without going through a liquid phase. This term, whose definition is available in a dictionary, has been kept.

*Lines 723 to 736: highly redundant. Could reduce to the sentence on lines 727 to 729: "Ambient temperature, moisture, and manure storage or residency time affect the amount of CH4 produced because they influence the growth of the microorganisms responsible for CH4 formation". For budget purposes, we don't need more than that?*

We agree, this paragraph has been reduced. Three sentences have been deleted.

*Line 751: Start here while deleting the prior paragraph?*

Yes. Anaerobic conditions in rice have already been mentioned near the beginning of the text. Few sentences have been deleted here.

*Line 775: "the work of (Carlson et al., 2016)" you mean 'the work of Carlson et al. (2016). Please take care with punctuation, as proofreaders will not know what you want.*

Yes. This has been corrected.

*Line 779: "northward shift of rice cultivation". Unless cultivation practices change with latitude, a latitudinal shift will not necessarily change or reduce emissions?*

Indeed, the paper suggest that such a decrease could be partly due to shift of rice cultivated area from southern China (paddies) to northern China (dry rice).

*Lines 837 to 905: way too much information here! Please only include what we need to know to understand this/these budget estimates. Also we seem to have lost the useful end-of-section summary of what this budget effort concludes for this source?*

This section has been greatly reduced and unnecessary details and descriptions, removed. The authors do not include these changes here for the sake of length of the response.

*Line 979: by this point the reader has encountered anaerobic conditions many times (waste water ponds, rice paddies, bovine guts, etc.). Do we really need a new definition and explanation of anaerobic metabolism at every point. Reduce, s.v.p.*

The first sentence has been deleted. The processes related to methane transport in wetlands and non-wetland freshwaters are now mentioned once, at the beginning of Section3.2.

*Line 980: "limit oxygen availability and creates suitable redox" create, not creates?*

This has been corrected, then deleted. See comments above.

*Line 987: are these processes diffusive or advective? This reader does not understand the phrase: "molecular diffusion limited advection"*

This has part has been deleted – after correction, see comments above.

*Line 992: "transportation and are further regulated by" What are 'further regulated'? The land-surface. models? The CH4 emissions? The model parameters? Very confusing, please re-write.*

This has been rephrased to:

**"… and transportation. The models are then forced with inputs accounting for changing environmental factors"**

*Line 995: "(Supplementary Material, Melton et al. (2013); Bohn et al., 2015)" Again, inconsistent and incorrect use of parentheses. Relying on bibliometrics software from various contributors evidently does not work. I have pointed out a few such errors, skipped over many, and anticipate many to come. A waste of this reviewer's time to point them all out. Assign one of your co-authors to search for, evaluate and correct every parenthesis. Authors must take this responsibility. Proofreaders will not find nor know how to correct all of them.*

This has been corrected here and elsewhere.

*Line 1008: "a monthly global wetland area dataset" Does wetland surface area vary only seasonally or does it also evolve with time over several years?*

Here, it is both seasonal variation and dynamic over the 2000-2017 period. This has been rephrased as follows:

**"WAD2M provides year to year varying monthly global wetland areas over 2000-2017."**

*Line 1056 to 1074: a hard-nosed editor could re-write this entire section in two sentences or perhaps even one sentence. How much of this is relevant to this current version of a CH4 budget?*

This part has been reduced substantially as requested.

*Line 1107: "south of 60°N) with. Tan and Zhuang" something wrong with punctuation here?*

This has been corrected.

*Lines 1165 to 1174: here we get expert opinion on what next and what needed for lakes and rivers. Good! But doesn't this section belong elsewhere/later?*

As suggested, these lines have been removed from the budget Section, and used later in the discussion/perspectives.

*Line 1234: why specify exact (and proprietary) GIS software here? Same scale up would have worked in any GIS environment?*

Indeed, the detail on the software is not relevant here. The same scale up would be achieved with any GIS software. This has been removed.

*Line 1341: "which allows very little methane; even from established". Punctuation error?*

The sentence has been rewritten to:

**"Aerobic oxidation is a very efficient sink process, which allows very little methane from reaching the atmosphere even from established and vigorous gas seep areas or below-water gas well blowouts."**

*Line 1376, 1377: Authors already defined marine clathrates earlier (line 392). Reader does not need redundant definitions?*

Indeed, two sentences defining again clathrates have been deleted.

*Line 1398: In earlier sections readers found summaries first, then itemization of individual components following. Here we get the opposite: components first followed by summary. Chose one or the other but do not confuse us with both?*

Indeed, for anthropogenic emissions we discussed emissions by broader to finer categories, while for natural emissions, we do the opposite. We discuss estimate for each fine category and then combine… This way to proceed is more convenient for natural emissions, as few studies assess the broad categories but rather look into some specific source or process. The authors feel that the reader can accommodate to this potential issue.

*Line 1455: Many references in this section appear as (non-functional) hot links. Many other references do not. Please fix this!*

Yes, we worked on correcting this.

*Line 1511: Finding the (strange) unit here of "molec" and having earlier seen reference to mole fractions (strictly, in an SI units sense moles per mole but more casually expressed as ppb), one wishes the authors had provided a conversion table as in the global carbon budget. They have done a very good job of keep flux units consistent as Tg CH4 yr-1, but they could help readers by cross-referencing the various concentration units.*

Methane emissions are given in Tg $CH_4$ yr$^{-1}$

Methane mixing ratios are usually expressed in ppb.

OH concentrations in the atmosphere are commonly given in molecules cm$^{-3}$ in the literature as done here. The authors have not seen any occurrence of OH concentration in mole per mole unit in the literature.

In the CO2 global budget, they provide conversion factors for CO2 to C, and one conversion factor from carbon flux to carbon concentrations in the atmosphere (assuming many hypotheses, not valid here neither for $CH_4$ or OH).

As a result, the authors will keep the initial unit used in the paper. **"molec" has been replaced by "molecules".**

*Lines 1498 to 1558: do we need an extensive discussion of OH concentrations, reactions, hemispheric distributions and lifetimes to understand the magnitude and validity of the present CH4 budget? This review thinks not. One could summarize this entire section in two or three lines?*

Well, the authors understand that the text was missing a direct relationship between OH and methane loss. A sentence has been added to address this. ("**Mass-weighted OH tropospheric concentrations do not directly represent methane loss, as the spatial and vertical distributions of OH affect this loss, through, in particular, the temperature dependency and the distribution of methane. However, estimating OH concentrations and, spatial and vertical distributions is a key step in estimating methane loss through OH**.")

The authors also acknowledge that the discussion was too long and sometimes off-topic (not the right time period, top-down instead of bottom-up…). As a result, many parts of the text have been removed.

*Line 1526: Pinatubo eruption represented a climate variation? Hardly.*

Indeed, Pinatubo eruption is not a climatic variation, but it did impact the climate for the following years. This has been deleted to avoid misinterpretation.

*Line 1530: "enhances CH4 consumption"? You mean 'enhances the CH4 sink' or 'enhances CH4 oxidation?*

Yes. This has been rephrased to "**CH4 oxidation**".

*Line 1548: "consistent, albeit lower, than the value deduced from"? Very sloppy wording, leaves a reader guessing at what you mean here.*

Yes, this has been simplified to **" lower than the value deduced from**".

*Lines 1577 to 1579: This sentence summarizes the entire stratospheric dynamics and photochemistry section. Why do we need all the rest of this?*

The reviewer is right. This section has been substantially reduced to keep only whet is relevant to the stratospheric methane loss estimate.

*Line 1590: Should CH4 sink terms use a different sign than source terms? I suspect carbon budget has already wrestled with this flux direction issue. What did they decide? Does GCP adhere to a consistent sink/source nomenclature convention?*

Here both emissions and sinks are provided as absolute values (positive), as done in the $CO_2$ global budget, where all values given in the tables are positive.

*Line 1606: "the KIE approach" KIE = kinetic isotope effect, but you haven't defined it yet?*
Indeed, KIE is now defined near the beginning of the section, when it is first mentioned.

*Line 1679:  systematic atmospheric?*
This has been corrected.

*Line 1695: A couple TCCON sites, perhaps Australia or NZ or both, have published CH4 data in ESSD?*
Yes? There is a Pollard et al. (2017) paper that has been published in ESSD. This citation has been added. Pollard, D. F., Sherlock, V., Robinson, J., Deutscher, N. M., Connor, B., and Shiona, H.: The Total Carbon Column Observing Network site description for Lauder, New Zealand, Earth Syst. Sci. Data, 9, 977–992, https://doi.org/10.5194/essd-9-977-2017, 2017.

*Lines 1699 to 1711, other data. If not useful or used for this global CH4 budget, why mention it/them?*
Indeed, the previous manuscript has been already amended in this direction but not enough. The text on "other" data has been removed.

*Lines 1713 to 1716: a reader just saw this identical list a few lines earlier. We don't need to see two mentions of the same information?*
Indeed, this has been reduced to:
" **We use globally averaged $CH_4$ mole fractions at the Earth's surface from the four observational networks (NOAA/ESRL, AGAGE, CSIRO and UCI).** "

*Line 1739: And you show GATM for the previous version of global CH4 budget (e.g. in Table 3) but not for this version. Why not?*
We thank the reviewer for this important remark. We did forget to report the values in Table 3. This has been corrected in the revised version of the manuscript.

*Line 1745: By definition, a growth rate can not be both positive and stable. Instead of stable, you mean consistent or persistent?*
This has been corrected to "**positive persistent growth rates since 2007**"

*Line 1751 to 1755: Here a reader finds the recent growth values, nicely documented with appropriate uncertainties. Why do these values not also appear in Table 3?*
Indeed, this was a missing element in Table 3. The values have been integrated to Table 3.

*Lines 1756 and following, Satellite data. Important to show command of this data for for budget purposes but way way too much information here. Why write four or five sentences of description about a source that you - for valid reasons - can not use? Is this a budget discussion or narrative about the history of SWIR sensors? You define SWIR at least twice in this section. Compilation - good - but not followed by tight budget-focussed editing - bad.*
Indeed, such information is not relevant for the budget. This part has been removed.

*Line 1778: "retrieval approaches, Proxy and Full Physics. The proxy method retrieves" ???. A former sub-heading now buried in this text? We do not need all this information?*
This part has been modified to remove extra details. The text is now: "**Different retrievals of methane based on TANSO-FTS/GOSAT products are available: NIES (Yoshida et al., 2013), SRON (Schepers et al., 2012) and University of Leicester (Parker et al., 2011). The three retrievals are used by the top-down systems (Table S6)**."

*Line 1792: "only inversions using GOSAT retrievals are used." After nearly 40 lines of text a reader finally finds this short conclusion. Please can someone apply a sharp red pencil to this manuscript - focus on the CH4 budget!*

The details on instruments other than GOSAT have been removed, and the paragraph now focuses on GOSAT only. The authors have kept the last sentence (though reduced) because on the previous budget SCIAMACHY was used, but not anymore.

*Line 1799: "ensemble of inversions gathering various chemistry transport" Why? To reduce uncertainties or provide independent validation for the CH4 budget terms (e.g. use top-down to constrain bottom-up) or to ensure that a long list of inversion modelers get their work listed and recognized via this global CH4 budget paper. I apologize if this reviewer's questions seem uniformed, irrelevant or even rude. But pity the poor readers/users. They want to use this product to understand CH4 budgets! Instead, they confront pages of tangential descriptions replete with (obscured by) more and more acronyms. Seriously, am I reading the CH4 budget or a narrative recount of every CH4 activity that has occurred?*

The authors understand the reviewer's feeling after such a long review. We might have missed an explanation of why two budgets (bottom-up and then top-down).

From the bottom-up budget, estimates of emissions from the different sources are built independently, without confrontation to other studies, and especially without constraints from the atmosphere. Doing so, the reader understands that summing-up all this estimate brings to too high total global emissions.

Using the top-down estimates is a way to introduce atmospheric constraint. However, inversions have caveats as well (good for the total fluxes, less good for the sectoral partition; good for the total, less good for the regional sources). Confronting bottom-up with top-down has help to improve both approaches, and shows ways for next improvements. The GCP methane budget is also a platform of communication through the many different communities working on methane; communication, which was clearly missing few years ago, is the key to solve the main issues of the methane budget.

We acknowledge that Saunois et al. (2016) reports many activities with probably too much details, lying between a budget and a review. From now on, and to address most of the referees' comments, the "review" part can be set aside and GCP will focus more on the budget, while reducing the review components (and also the co-author list).

*Lines 1801, 1802: "assume that this model range is sufficient to cover the range of transport model errors in the estimate of methane fluxes". You 'assume'? This assembly of world experts on CH4 uses an ensemble approach because they can't (or won't) distinguish useful from not useful inversions? Again, does this approach help quality of the budget? If so, quantify that. Does it help users? If so, tell us. Show us! Did the inversions prove useful in the previous CH4 budget? If so, how? Although these authors claim (lines 1807, 1808) intention to do flux assessment rather than a MIP, the section in its present description looks, smells and sounds like a MIP to this reader.*

First, this is not an inter-comparison study as the model set-ups differ and generally do not even use the same prior fluxes. This has been emphasized with the following sentence:" **This approach corresponds to a flux assessment, but not to a model inter-comparison as the protocol did not enforce the use of harmonized priors and inversion set-up, as well as the set of input atmospheric data.**"

Well, of course, some inverse systems may perform better than other. A model evaluation needs to be done to assess this (an on-going work beyond the scope of this publication). In the future, such evaluation criteria should be defined and evaluation would need to be done before the final analysis and discussed in the next update of the budget.

Waiting for this, including all the models is a conservative approach that allows to cover different unresolved uncertainties: model transport, set-up issues including prior set-up.

The mentioned sentence has been replaced by: "**Including these different systems is a conservative approach that allows to cover different potential uncertainties of the inversion, among them: model transport, set-up issues, and prior dependency.**"

*Lines 1844 to 1846: Here, deep in the manuscript where few readers will notice, we finally read a justification of costs and benefits of time span and time lag of the current product. Good discussion! But we should have read it in the introduction? It sets out temporal goals that guide the entire product!*

The reviewer is right. It has been moved accordingly, to the methodology section 2.2 (Period of the budget and data availability).

*Lines 1850 and following: This is a good useful orderly section! But: a) it repeats a lot of material from earlier; and b) a reader needed to work through - or, more likely, skip through - an unusually large amount of text to get here. Authors could argue that this section depends on all the prior detail. This reader/reviewer asks authors to consider how to make the user's 'life' easier rather than putting priority on including all details. Details can and should go in individual research papers. The budget should not replay, only summarize?*

The aim of the methane budget paper is to produce an update of the different sources and sinks of methane. Despite the feeling of the reviewer, Section 3 is a summary of estimates of methane sources and sinks from bottom-up approaches. The organization of the paper is presented in the introduction, the reader has the possibility to skip Sections 3 and 4 to go directly to the budget, depending of his interest.

*Line 1882: Introductory sentence implies treatment by the five source categories but discussion/summary that follows treats natural vs anthropogenic (with wetlands called out separately) but does not follow the five-category organization that you used earlier, for example. Revise this opening sentence to better convey what will follow?*

The discussions on each anthropogenic sector are quite short here. So we did not follow the five categories organization. The first sentence has been modified accordingly as follows: "**The global methane emissions from natural and anthropogenic sources**…".

*Line 1914; "study derive significant changes in wetland emissions" derive? I think you mean 'identify' or 'reveal' or 'point to'?*

Indeed. This has been changed to "**point to**".

*Line 1919: We just read (lines 1875) that uncertainty ranges for, e.g. ocean, termites, geological sources have reduced (improved) in this version, but here the authors point to those same specific sources as plausible reasons for the top-down bottom-up discontinuity? Uncertainty goes down but discrepancy goes up? Mathematically I think that works but this discussion implies a substantial weakness in our understanding. This reader particularly wonders about permafrost in this regard (see opening comments above), not so much about the permafrost numbers but more about whether assigning permafrost emissions as natural emissions doesn't presently or won't in the future exacerbate this discrepancy?*

This sentence has been deleted and replaced by "**This discrepancy comes from estimates in "other natural" emissions (non-wetland freshwaters, geological sources, termites, oceans, and permafrost)."**
Later in the text, we now include the permafrost, and state:" **Better constraining the estimation and partition of methane emissions between wetlands and non-wetland freshwater systems, including emissions from thawing permafrost, may be the key to reconcile top-down and bottom-up budget on natural sources.**"

*Line 1930: "Improved area estimates …" By this point reader has seen this recommendation at least twice. It really rather belongs in a summary or future work section, not here? This entire paragraph repeats earlier text and belongs instead in a summary section?*

Indeed. This has been partly moved to the Section 6, which has been entirely re-organized…

*Line 1948: "which is about 30% (23%) of global methane" What does the 23% indicate here? About 30% but actually 23%? 23% of global but 30% of anthropogenic? Some plus minus 23% uncertainty? I find the*

*answer to my question in line 1949, 30% of top-down but only 23% of bottom-up. Readers should encounter definition before data, not the other way around?*

This has been rephrased as follows: "**which is about 30% of the top-down global methane emissions, and 23% of the bottom-up total global estimate.**"

*Line 1958: "partition of methane emissions between wetlands and freshwater systems should still receive a high priority" this includes the thawing permafrost freshwater confusion?*

The reviewer is right that the permafrost issue is not fully considered in the discussion, and that, potentially, methane emissions from permafrost are included either in the wetland estimate (if satellite wetland areas cover peatland or thermokarst in permafrost regions) or in non-wetland freshwater emissions, or both. The sentence has been reformulated as follows: "**the estimation and proper partition of methane emissions between wetlands and non-wetland freshwater systems, and emissions from permafrost ecosytems, should still receive a high priority**"

*Lines 1982, 1983: Valid caution but a reader should have encountered it earlier?*

The authors think that this statement needs to be kept here, following the results. However, we have added in Sect. 4.2 the following sentences: "**In poorly observed regions, top-down surface inversions may rely on the prior estimates. Further, inversions bring little or no additional information to constrain (often) spatially overlapping emissions (e.g. in India, China) even in well observed regions. We recall that many top-down systems solve for total fluxes at the surface or for some categories that may differ from the GCP categories.**"

*Lines 1997 to 1999: Very confusing here, again the source/sink flux direction problem. The text says 'model inputs somewhat higher' but what you mean is some models show a higher value for total sink because they include a larger land sink term? For a known atmospheric CH4 concentration, that larger sink term must lead to a correspondingly larger source term! Here, we read about a larger central top-down estimate. Authors intend this phrase to refer to the larger sink term, or to a larger total CH4 emissions term? The authors, no doubt, know various sink, source and net terms, but here they fail to express themselves in clear language, leaving readers therefore confused. This statement occurs within the section on CH4 sinks, so one wonders …*

Well, the paragraph sits in the "sink section". The authors agree that the meaning of the sentence was far from clear. It has been rewritten as: "**These sink estimates used as prior in the inversions are generally higher than the mean estimate of the soil sink calculated by bottom-up models (30 Tg CH$_4$ yr$^{-1}$, Sec. 3.3.4).**"

*Line 2006: "chemical lifetime and to narrow it down in" Authors have addressed the issue of lifetime many times and in great detail prior to this statement. A reader knows at this point a lifetime of 9 years, plus or minus perhaps 0.5 years. How would refinement of this lifetime calculation improve the overall budget calculation, much less mitigation planning? E.g. does a lifetime of 8 years or 10 years really make a difference at this point? This reviewer suspects not. This section represents an(other) example of everyone's scientific priority gaining equal weight in the compiled budget. This (small, to this reviewer) remaining uncertainty has larger overall priority than fixing the distribution and extent uncertainties of freshwater (also mentioned numerous times)? Organizationally, this entire section (lines 2005 to 2012) belongs in a summary / future work section.*

This part has been deleted from the text, and most of the ideas have been inserted, more clearly, in Section 6.

*Line 2024: A reader has now seen this statement at least four times?*

This sentence has been removed. The following sentence has also been removed to shorten the text.

*Line 2033: Readers already know, at least twice, why SCIAMACHY data not used.*

This has been removed.

*Lines 2024 to 2053: Good knowledgeable expert discussion of impact of satellite vs in situ data on inversions and emissions. But, of what relevance, in specific plus/minus Tg CH4 terms, to the global CH4 budget? A reader sees "clearly show" but at the same time "not systematically consistent". I almost used a profane shorthand here: what do the authors want readers to learn, and how does it matter to the CH4 budget? No clue in this section. More "further investigation" ….*

The text means to say: At the global scale, satellite and surface-based inversions give approximately the same global emission (consistent growth rate between surface and satellite observations). However, the regional distributions differ depending on the nature of the observations used (satellite or surface). Also the regional patterns of these differences are not consistent through the different inverse systems. Indeed, some systems suggest higher emissions in the tropics when using GOSAT instead of surface observations, while other suggest the opposite.

The "further investigation" part, has been removed from this section.

The text has been modified as follows: "**As expected, the regional distributions differ depending on the nature of the observations used (satellite or surface). The largest differences (satellite-based minus surface-based inversions) are observed over the tropical region, ranging between -13 and +26 Tg CH$_4$ yr$^{-1}$ below 30°N, and over the northern mid-latitudes (between -20 and +15 Tg CH$_4$ yr$^{-1}$). Satellite data provide stronger constraints on fluxes in tropical regions than surface data, due to a much larger observational coverage. It is therefore not surprising that differences between these two types of observations are found in the tropical band, and consequently in the northern mid-latitudes to balance total emissions, thus affecting north-south gradient of emissions. However, the regional patterns of these differences are not consistent through the different inverse systems. Some systems suggest higher emissions in the tropics when using GOSAT instead of surface observations, while other suggest the opposite. This difference may depend on whether or not a bias correction is applied to the satellite data based on surface observations, and the transport**."

*Line 2082: Section 6, developments, missing elements, remaining uncertainties. Important section. Readers will not disagree with topics raised here. But the section reads more as a recitation of the previous CH4 budget issues while omitting issues raised in this version. Ebullition, for example, mentioned repeatedly in the text, does not emerge here. The entire tension between spatial uncertainties and chemical uncertainties, a theme of this paper, does not emerge here, at least not in those recognizable terms. Pages of text on the need to improve distribution or lifetime of OH, but only one faint mention here, that basically repeats what we already read? We could get a list: previous but now resolved, previous but not yet resolved, new. Here a reader finds no sense of priority, just a large wish list not much changed from the prior budget. Use the GWP of CH4, find the largest or most tractable uncertainties, convert those to climate urgency: where should we put our efforts? If we know new satellites have recently or will soon come on line, what do we need for validation? Where will that data have most impact in this budget? More, more, more. More systematic. More integrated. Just another wish list. After all the work to compile this information, the world-wide experts then throw up their hands - they don't know what they need next? They need everything? Not helpful.*

*After working to assimilate all the details provided to this point, this reader basically glazed over this section. It seemed only vaguely related to what we just learned, a white -paper level of what we should do to better understand CH4 but not well connected to all that we just read about efforts and information need to assemble this budget.*

Section 6 has been fully re-written in order to highlight the priorities and the time line of the needed measures to improve the budget and our understanding. The first item is on the wetland and freshwater issue, which emphasizes this point as number 1 priority. The authors feel that the newly written Section 6 resemble more a road-map than a wish-list from a white paper.

*Line 2089: "Knox and al., 2019)" ??*

This has been corrected.

*Lines 2209 to 2219: a short clear paragraph of budget outcomes - thank you! Notable, unfortunately, because it differs so much in style, brevity and clarity from most of the preceding text.*

This is a comment.

*Lines 2226 to 2231: weaknesses and cautions about extracting regional information from satellites. Important, clear, useful. But link this to specific recommendations in Section 6, where now we just find a wish list for more of everything?*

Section 6 now better presents the recommendations for this issue; the authors refer to Sect. 6.

*Line 2237: a "clear priority"! Finally! First one? Does permafrost fit into this inland freshwater priority?*
*Line 2242: "also place importance in" Should a reader conclude here that authors consider improvements in OH distributions as second in importance to the freshwater extent problem?*
*Line 2246 to 2253: Short clear paragraph, contains a bit of the unhelpful 'need everything' philosophy, but if we have these topics listed here we could get rid of much of Section 6?*
*Lines 2254 to 2257: Good statement, good motivation, good summary of forward plans, but repeats almost verbatim what readers saw earlier (lines 1844 to 1846)? Some repeat expected in a good summary but not clear here why we need this detail twice?*

Response to the 4 above comments:
Indeed, the conclusion included many repeats from Section 6. This part has been highly reduced to avoid this.

*Line 2268: "a negative contribution (from biomass burning" You really do not want punctuation errors in such an important paragraph?*

This has been corrected.

*Line 2299 and following, Acknowledgements. Interesting section. I think global carbon budget does something like this but through a table instead? Nothing wrong with this approach but text includes several (perhaps many) punctuation and tense errors. Someone of the author team should read this carefully?*

For the $CO_2$, budget the funding acknowledgement have been gathered in an appendix Table. We have done the same and put the acknowledgement to funding and support in Table A1.

*Page 126, 127, Table 3. Cells need reformatting, presumably will happen during typesetting. Authors will need to check text wraps. Repeat the question raised above about why no GATM rate information for current version(s)?*

GATM has been added. Table 3 formatting will be check during proofreading.

**References:**
Nakazawa, T., Machida, T., Tanaka, M., Fujii, Y., Aoki, S., and Watanabe, O.: Differences of the atmospheric CH4 concentration between the Arctic and Antarctic regions in pre-industrial/pre-agricultural era, Geophys. Res. Lett., 20, 10, doi:10.1029/93GL00776, 1993

---

## Author Comment (AC2) · 2 Mar 2020

**The Global Methane Budget: 2000-2017**

Saunois et al., ESSDD, 2019

**Detailed Response to Tonatiuh Guillermo Nuñez Ramirez**

The authors thank Tonatiuh Guillermo Nunez Ramirez for his comments on some specifics part of the text that needed some clarification.

Below are the responses to his comments (in italics, blue). Changes in the text follow each response in bold font.

*The emission estimates for the decade of 2000-2009 in Saunois 2016 have larger ranges than in Saunois 2019. I think, if the GCP condenses all information that exists, there should be at least a table explaining which studies were left out and why as supplementary information.*

We thank Dr Nunez Ramirez for the suggestion. We will probably consider to do this for the next release to highlight the methodology changes between the different budget releases.

*A major problem with the bottom-up budget is that it is so much larger than the top-down estimate. The border between wetlands and other fresh water systems is very fuzzy and more discussion is required. Historically wetlands have been classified as bogs, fens, swamps, floodplains, and shallow lakes (Bartlett 1993). For example lake Chapala, Mexico's largest lake, has a maximum depth of 2 m, is it a wetland or a lake? Is a floodplain to be considered a wetland or a freshwater system? For example, in the Amazon inundation can vary for several meters.*

The definition of wetland as well as the boundaries between the different non-wetland inland water systems (lakes, reservoirs, ponds, …) is critical. The different definitions used within the scientific community make difficult the budget assessment and favors double counting issues. The issues raised by Dr Ramirez are emphasized in the wetland subsection as well as in the inland water subsection. Section 6 presents some recommendations to overcome these issues.

*Furthermore, in the Eastern Amazon, emissions tend to be larger at as river flow starts to decrease in August and September (Devol 1988, Beck 2012, Ringeval 2014, Basso 2016). This seasonal maximum is not capture by any of the WETCHIMP models, which instead show a maximum between January and April (Ringeval 2014). Ringeval (2014) were able to reproduce a seasonal cycle of CH4 emissions from the Amazon mainstream that was more similar to observations by using output from a hydrology model to identify floodplains.*

Specific behaviors of surface land models are not investigated in this budget and none of the models used included a hydrology model for floodplains, as they used satellite-based wetland (flooded area) data which may not capture inundated areas covered by trees such as the Amazon floodplains (see Hastie et al. 2019). However, further details will be provided in a side paper led by Ben Poulter.

*Furthermore, from your description it seems you classify as wetlands as saturated soils and fresh water systems can be lakes, rivers, reservoirs. Early studies, e.g. Devol (1988, 1990), Bartlett (1988, 1990), Tathy 1992, Keller 1994 and Melack (2004) made measurements both over saturated soils, emergent plants and open water. These early studies were used to calibrate many models, for example, in Spahni (2011), the LPJ-Bern model was calibrated to match the seasonal cycle from an inverse modeling estimate.*

The calibration (and evaluation) of wetland models remains difficult due the lack of appropriate observation data sets. As stated by Melton et al. (2013), model comparison to observations requires observations compatible with the spatial scale of the models (usually 0.5 degree). Some models use flux measurements to adjust their flux density, and others use atmospheric top-down estimate to calibrate their model –for example by latitudinal band in Sphani et al. (2011).

*Furthermore, DelSontro et al. (2018) has very high emissions but is stratification and transport within a lake were not taken in consideration. For example, lakes in East Africa are highly stratified and anoxic below the mixed layer but the amount of emissions estimated by DelSontro (2018) is difficult to bring in agreement with satellite $CH_4$ cartographies (e.g. Frankenberg, 2011). they do not emit high quantities of methane continuously due to the same stratification of the water column. Is the future of CH4 emission modeling the merging of dynamic vegetation models with hydrology models?*

The DelSontro et al. (2018) study does not consider seasonal variations in stratification and transport, rather they use sample from the warm season and satellite-based product for Chl-a based on the growing season for phytoplankton apparently (see Sayers et al., 2015).

Yes, future developments need to integrate both wetlands and other inland waters systems together. This would avoid double counting and integrate flux transfer between systems. Recommendations for future development are suggested in Section 6, newly reformatted.

*With respect to the soil sink, your estimates are based on published model estimates. However, in these models, the sink strength depends on atmospheric mixing ratio (of-ten a global constant value). For example, in the Curry (2007) model, the flux j is*

*$j = C_0 * g_0 * r_w * r_c * \qquad D * k$*

*where $C_0$ [ppm] is the $CH_4$ mixing ratio and $g_0$ is a conversion factor from ppm to mass units. Taking this into account, the sink becomes much larger may become much larger and changes in time in proportion to the atmospheric abundance. Furthermore, both Ridgwell (1999) and Curry (2007) had use the ideal gas law to set the g0 parameter to 610 and 578 assuming a pressure of 100 kPa and temperatures of $0°C$ and $15°C$ respectively. By determine the g0 per gridcell based on monthly temperatures and pressure, the g0 ranges between 320 and 750.*

Indeed, the soil sink depends on atmospheric methane concentration. Changes in the soil sink needs to be further investigated and in particular, included in the priors of top-down simulations, which is not systematically the case presently.

Regarding the conversion factor g0, this factor is inversely proportional to T (expressed in K), so that even for a temperature range of -10 C to 30 C (typical of a continental mid-latitude location), g0 changes by only 16%. g0 does not depend on pressure. Reviewer SC1 may be confounding the temperature dependence of g0 with the T and P dependence of $CH_4$ diffusivity in air ($D_{air}$); but this too is only weakly dependent on T, and in the opposite sense to g0—see, e.g., CRC Handbook of Chemistry & Physics, 20th Ed. (2016), p 6-259. More importantly, the error involved in assuming a global constant value of g0 and $D_{air}$ in this parameterization is dwarfed by the uncertainty in the oxidation coefficient k0 which, as emphasized by Curry (2007), could change the derived uptake by a factor ~3 or more in either direction.

*Additionally, there are important contribution from Hackstein (1994, 1996, 2006) concerning potentially large emissions from wild terrestrial vertebrates and three arthropod taxa apart from termites.*

We thank Dr Nunez Ramirez for pointing this to us. We understand from this literature that the uncertainty is quite large.

*In the future, it would be useful to also have estimates of the year-to-year variability for wetlands and OH in order to understand what drove the observed year-to-year variability of the growth rate.*

Further study on the year to year variability from wetlands emissions or OH will be conducted in another paper as in Saunois et al. (2017). For the future budgets, we will consider if and how the IAV and changes in emissions and atmospheric methane can be discussed in the same paper without lengthening to much the manuscript.

*References:*

Hastie, A, Lauerwald, R, Ciais, P, Regnier, P. Aquatic carbon fluxes dampen the overall variation of net ecosystem productivity in the Amazon basin: An analysis of the interannual variability in the boundless carbon cycle. Glob Change Biol., 25, 2094– 2111, https://doi.org/10.1111/gcb.14620, 2019

Melton, J. R., Wania, R., Hodson, E. L., Poulter, B., Ringeval, B., Spahni, R., Bohn, T., Avis, C. A., Beerling, D. J., Chen, G., Eliseev, A. V., Denisov, S. N., Hopcroft, P. O., Lettenmaier, D. P., Riley, W. J., Singarayer, J. S., Subin, Z. M., Tian, H., Zürcher, S., Brovkin, V., van Bodegom, P. M., Kleinen, T., Yu, Z. C., and Kaplan, J. O.: Present state of global wetland extent and wetland methane modelling: conclusions from a model inter-comparison project (WETCHIMP), Biogeosciences, 10, 753–788, https://doi.org/10.5194/bg-10-753-2013, 2013.

Sayers, M.J., Amanda G. Grimm, Robert A. Shuchman, Andrew M. Deines, David B. Bunnell, Zachary B. Raymer, Mark W. Rogers, Whitney Woelmer, David H. Bennion, Colin N. Brooks, Matthew A. Whitley, David M. Warner & Justin Mychek-Londer: A new method to generate a high-resolution global distribution map of lake chlorophyll, International Journal of Remote Sensing, 36:7, 1942-1964, DOI: 10.1080/01431161.2015.1029099, 2015.

---

## Author Comment (AC3) · 5 Mar 2020

**The Global Methane Budget: 2000-2017**

Saunois et al., ESSDD, 2019

**Detailed Response to Michael Prather (Referee #2)**

We acknowledge the work from Michael Prather for the time spent on reading and commenting on the paper. We thank him for his useful corrections and suggestions on the paper, which have helped clarifying and improving the manuscript. Below are the responses to his comments (in *italics, blue*). Changes in the text follow each response in **bold font**. Following both reviewers advices, the text has been shortened by 5 pages (8%), from 61 to 56 pages in the ESSDD format.

*This is a very well written and thorough review of the methane budget. The abstract and introduction read easily and layout the scope of the work. The remaining 2000 lines are a bit harder to get through, but this paper is meant as a reference work and not a "beach read." For a paper on methane, it is great to see Ehhalt's original work as one of the prime references. I have two major suggestions and then several minor/editorial comments on my read-through.*

We thank M. Prather for the compliment.

*#1 As a reference work, it would be very useful to have a Table of Contents up front. The TOC should be as detailed as possible (e.g., add another level: 3.3.2.1. Lakes and Ponds) and possibly include a list of key words for each section. This would greatly help those looking for specific discussions.*

A table of content has been added. Unfortunately, ESSD allows only 3 levels for the sectioning. We will add as much as possible details in the TOC, that fit ESSD style requirement.

*#2 The Section 3.3.5 on CH4 Lifetime contains some serious errors in understanding. I am not sure how to fix it, but the authors should think about the purpose and what they want to get across. See details below.*

Section 3.3.5 has been revised accordingly to the detailed comments. We have kept only the calculation of lifetime based on CCMI models and using the "budget" definitions: Total_burden/Tropo_loss for the tropospheric lifetime and Total_burden/total_loss for the "total lifetime". The other definitions were removed from this section to avoid confusion.

*L217: The Prather 10% is only 68% confidence interval, perhaps if the Saunois 15% is 90% confidence, they are consistent.*

The Saunois et al. (2016) uncertainty is based on a min-max range using the different ACCMIP models. Some of the models have difficulties in well representing methane loss due to uncomplete chemical scheme. The uncertainty is larger but its conservative and includes all models…

*L225: Probably more important here is not 'credible scenarios,' but credible mitigation strategies.*

Yes. Mitigation strategies are more important. Though the future climate scenario should help in defining mitigation strategies… We have replaced "credible future climate scenario" by "**appropriate mitigation strate**gies".

*L231: do you want to mention the Paris Accord's "stocktaking"?*

This has been added in the sentence as: "**In order to verify such reductions, for example to help conducting Paris Agreement's stocktake,…**"

*L243: This is a slight mis-direct. It is not the lack of some direct observations, but by my calculation, the number of OH "measurements" needed to integrate the loss of methane (1 km x 100 m parcels, every $\frac{1}{2}$ hour (clouds)) is of order 2x1014 per yr. I would put this as an impossible task without some modeling and other tricks.*

Yes, several ideas were mixed there. This part has been reformulated to:" **The spatial and temporal distributions of OH are highly variable. Although OH can be measured locally, calculating CH₄ loss through OH measurements would require OH measurements every half on hour (to integrate cloud cover), in each small parcel (below 1km spatially to consider OH high reactivity and heterogeneity) of the atmosphere. As a result, such a calculation is currently possible only through modelling. However, simulated OH concentrations from chemistry climate models still show uncertain spatio-temporal distribution at regional to global scales (Zhao et al., 2019)."**

*L290: Here and later, you cannot reference a work in prep as a real reference. You can describe ongoing work by Staevert and colleagues that will follow up this paper and come up with a shorthand notation for this work.*

This paper is in preparation and should be submitted before the final proof-reading of our paper. In case it will not be submitted, we will change the phrasing.

*L318-324: If you are being precise, please define where the edges of the 1-degree grid fall, one the 0.0, or 0.5, or? Also does what you describe mean that coastal boxes (<100% Land and <100% Ocean) have no emissions – you should definitely want to warn people as it will look funny when plotted. Also what do you do about large lakes (resolved at 1 degree)?*

We have added information on the edges: "**The edges of the 1°x1° grid fall in 0.0°.**" The land/sea (or lake) mask is an issue when estimating regional budget but not the global one. The reallocation of land emissions is based on the percentage of sea/land in the initial and final grid cells. Coastal boxes have then non-zero fluxes. The same treatment is reserved to the big lakes.

*L383: This discussion of natural vs. anthropogenic intrigued me. We have a long history of trying to break emissions cleanly in these two camps. With natural often being ignored in the scenarios (RCP, SRES), such that in the IPCC SAR & TAR, I had to make up the natural sources to ensure a consistent budget and initial trajectory for the future scenarios. The other problem is that attribution is still not an exact science and thus declaring something like wildfires or wetland loss as "anthropogenic" is not trivial.*

This issue has been raised by referee#1, and also by several co-authors, which would like specific treatment for some sources at the frontier between these two categories.

*Let me propose that the sources be split into "pre-agricultural" and new "anthropogenic" types of sources (such as ag or industry). This allows us to work our best science on how wetland methane emissions have changed, without trying to ascribe cause. The changes in the "pre-ag" sources probably can be attributed in many cases to direct human intervention.*

Thank you very much for this suggestion. This fits the definition the lead author has in mind, and the response provided to referee#1. Such a definition allows to sit permafrost emissions in the "natural" sources, while for sure, thawing permafrost will release more in the future due to climate warming, being an indirect anthropogenic source. The text in the Methodology section has been changed to:

"**In the following, we present the different methane sources classified from anthropogenic or natural origin. "Natural sources" refer to pre-agricultural emissions even if they are perturbed by anthropogenic climate change, and "anthropogenic sources" are caused by direct human activities since pre-industrial/pre-agricultural time (3000-2000 BP, Nakasawa et al. (1993)) including agriculture, waste management and fossil fuel related activities. Natural emissions are split between "wetland" and "other natural" emissions (e.g., non-wetland inland waters, wild animals, termites, land geological sources, oceanic geological and biogenic sources, and terrestrial permafrost). Anthropogenic emissions contain: "agriculture and waste emissions", "fossil fuel emissions", "biomass and biofuel burning emissions", assuming that all types of fires cause anthropogenic sources, although they are partly of natural origin (Fig. 6, see also Table 3 and 6).**
**Our definition of natural/anthropogenic sources does not correspond exactly to the definition used by UNFCCC following the IPCC guidelines (IPCC, 2006), where, for pragmatic reasons, all emissions from managed land are reported as anthropogenic, which is not the case here. For instance, we consider all wetlands as natural emissions, despite some wetlands being managed and their emissions being partly reported in UNFCCC national communications. The human induced perturbation of climate, atmospheric CO2, and nitrogen and sulfur deposition may cause changes in**

the sources we classified as natural. Following our definition, emissions from wetlands, inland water or thawing permafrost will be accountable in "natural" emissions, even though, we acknowledge that climate change – a human perturbation – may cause increasing emissions from these sources. Methane emissions from reservoirs are considered as natural even though reservoirs are human man-made, and since the 2019 refinement to the IPCC guidelines (IPCC, 2006; IPCC, 2019) emissions from reservoirs and other flooded lands are considered anthropogenic by UNFCCC."

*L529: can you make this simpler to read: "perhaps because AMAP analysed data from a wider range of inventories and projections, plus it was referenced to one year only (2005) rather than averaged over a decade, as done here."*
Thank you very much for the suggestion. This has been corrected accordingly.

*L552: "realized" in what? the scenarios or real life?*
In the real life... This has been changed to: "… **current emissions appear to follow the higher-emission trajectories over the next decade**".

*L660ff: In terms of shale gas emissions, is the DeCarlo work on the Marcellus Shale consistent with these? Goetz, J. D., et al., Analysis of local-scale background concentrations of methane and other gas-phase species in the Marcellus Shale, Elementa Science of the Anthropocene, 1–20, doi:10.1525/journal.elementa 182, 2017.*
The "oil and gas" section has been revised accordingly to Referee#1's comments. The discussion focuses on the budget of oil and gas emissions. Shale gas emissions are included in the "oil and gas" estimate from inventories. Few sentences have been kept to discuss the uncertainty regarding shale gas emissions – mainly through their different emissions factors. The authors looked at the Goetz et al. paper, they did not specifically discuss methane emission to compare to inventories such as EPA.

*L1026: Is this really true? If we have all positive values but some far outliers, then you would reject a positive flux because the std dev exceeded the mean?*
Here we use a simple diagnostic to roughly estimate the area over which land surface models agree on the emissions. Indeed, we did not do the calculation using a bootstrapping method (considering all but one model and looping over the models). This would indeed identify outlier model and provide an average area. However, we use the method as in the previous budget for consistency.

*L1174: I am unsure what the "up scaling issues" means.*
This includes the spatial and temporal uncertainty of the flux density used for a single class of ecosystems for example. This paragraph has been removed, belonging now to Section 6, following Referee#1's comment.

*L1205ff: This sentence on the Petrenko work is a bit perjorative and full of inuendo ("If it is correct", "which is questionable"). There is nothing obviously wrong with the work, unless the 14C community is worried about it, and if so state why. The sentence on L1208 about the discrepancy is fine.*
Indeed, we agree the wording was judgmental, and useless for the budget. The sentence has been removed.

*L1233: I am not sure what 'termite' as a unit is?*
We understand why the referee as difficulty with the unit. The density flux is given per biomass of termite, that is per 'gram of termite', which should be "(g termite)$^{-1}$" instead of "g$^{-1}$ termite"
The unit has been corrected.

*L1500-1507: I am a little worried about this section, but have no simple fix to recommend.*
*(1) the effect of CH4 oxidation on HOx depends on the pathway for H2CO, if it photolyzes on one path you get 2 HOx, otherwise, if OH is high, then OH-destruction it yields 0 net HOx. NOx plays a role in this, but HOx levels are also very important.*
*(2)The short lifetime of OH should go back to Levy or Logan, not a 2004 reference.*
*(3) "estimate"? One can estimate OH from models without observations.*

*(4) Many, many OH measurements are made in the free troposphere by Brune and others. I am beginning to think it best to stop at 'compounds.' on L1501 and jump to L1507 'Following. . .'*

Indeed, we agree that a long review would be needed to discuss OH. And this is not the point of the budget paper. This part has been removed as suggested.

*L1511: I think you have to recognize here that mass weighted OH concentrations are NOT a good measure of the methane loss. Since this is a methane paper, you should focus on the average methane loss rate from these models. I know this is a hobby of mine, but please look at the table in my 1990 paper (Prather & Spivakovsky, Tropospheric OH and the lifetimes of HCFCs, JGR: 95, 18723-18729, 1990), also reproduced by Mark Lawrence in 2000, and one can see that the average OH varies by 20-40% depending on how it is weighted. What is relevant here is when OH is weighted by mass and exp(-1800/T). In fact, the OH values do not tell you the methane lifetime unless you know the temperature weighting.*

It is true that the mass weighted OH concentrations do not represent a measure of the methane loss directly. Even though, the discussion should focus on ethane loss directly, some explanations on OH range in concentrations and horizontal and vertical distributions are needed to explain – partly – the range in methane loss. The following text has been added prior the discussion on OH:

"**Mass-weighted OH tropospheric concentrations do not directly represent methane loss, as the spatial and vertical distributions of OH affect methane loss, through, in particular, the reaction temperature dependency and the distribution of methane. However, estimating OH concentrations and spatial and vertical distributions is a key step in estimating methane loss through OH.** »

*L1527-1560: The Holmes et al paper (2013, Future methane, hydroxyl, and their un-certainties: key climate and emission parameters for future predictions, Atmos. Chem. Phys., 13, 285–302, doi:10.5194/acp-13-285-2013) really addresses recent OH variability and should be included with this discussion. (Sorry to push my own papers again, but it is a balanced survey of OH variations from cause [all those listed] to methylchloroform-derived variability.)*

The Holmes et al. (2013) paper has been included as a reference in this discussion.

*L1579: It is unclear if your 60 Tg is irreversible mixing (i.e., loss in strat) or the cross tropopause flux. The amount of CH4 entering the stratosphere is 10x or more larger than this number, most of which crosses into the lower stratosphere inn the sub-tropics and is then transported into the troposphere with little chemistry. I would drop this whole sentence as it does not say much.*

This part has been removed, in agreement with referee#1 comment.

*L1590: We have a much more accurate measure of the stratospheric loss from the Plumb & Ko relationship and the observed N2O-CH4 tracer slopes in the lower stratosphere. I do not know when this was last revisited, however.*

The authors have searched for updated estimates using Plumb and Ko relationship, but without success, to the best of their knowledge…

*L1654ff: This section 3.3.5 on CH4 Lifetime has some major problems. Excuse my didactic diversion here. "Lifetime" is a budgetary number since the standards and notation in the 1995/96 IPCC SAR through AR5. It is the burden divided any loss rate. Note that the total burden in the system must be used to take advantage of adding inverse lifetimes. Hence you will see the "lifetime of CH4 against trop OH loss" is the Tg (including stratosphere) divided by the Tg/yr lost to OH in the troposphere. If you use this definition carefully then inverse lifetimes can added and we can think of the lifetime due to stratospheric loss being about 160 yr as is recommended here. The OH lifetime should be noted and taken from the MIPs, it should be about 11 yr.*

*"Perturbation time, response time, e-fold time" are used to define the decay of a perturbation about any atmospheric state (again, steady-state does not matter for these definitions). Since 1994, for CH4 we have known that this time scale is about 1.4 times the total lifetime because of chemical feedbacks whereby CH4 suppresses OH. This is well known, consistently modeled and increases all the integrated impacts by this factor. With a budget lifetime of 9.1 yr, the time scale for CH4 perturbations is about 12 yr. Any perturbation to a chemically reactive species will excite a large number of chemical modes – each with its own pattern of species and its own decay time. Hence CO is an indirect greenhouse gas because it generates a CH4 perturbation that decays with a 12-yr e-fold. "Steady state" is not required*

*for these numbers, but the steady-state lifetime does have some magic properties. It is the effective average over all the different chemical modes (their amplitudes and time scales) excited by a perturbation.*

*The rest of the lifetime numbers from the recent MIPS look to be OK, but make sure that these are calculated using the full burden.*

We removed the sentence mentioning "Perturbation time, response time, e-fold time" from this section (bottom-up) and keep only the definition of lifetime as burden over loss.

The lifetimes were calculated using the total burden of methane from CCMI models – associated to the tropospheric methane loss to calculate lifetime with respect to OH removal. As a result, the numbers were correct. In this Section (bottom-up) we have removed the couples of sentences discussing lifetime from observation – considered as top-down estimates.

*L1708: The references to Rigby 2017 and Turner 2017 really should include also the accompanying Prather and Holmes paper (Over-explaining or under-explaining methane's role in climate change, PNAS 114(21) 5324-5326, doi: 10.1073/pnas.1704884114, 2017) that points out the fundamental error in modeling CH4 perturbations for both papers (i.e., they did not include the chemical feedbacks).*

This part of the text has been removed from the revised manuscript, following referee#1 comments – discussing with non-used data set in this budget…

*L1750. Based on notes above, I might expect the time scale here to be 12 yr rather than 9 yr, but I think it is probably close enough for this simple analysis.*

The commented approach is based on atmospheric methane concentrations recorded before and during the stabilization period. In this approach, we assumed no trend in emissions and in lifetime (so no trend in sinks). Such assumption can be criticized in regard with some literature explaining the stabilization period by decreasing emissions associated to increasing sink. We acknowledge that there are uncertainties associated with this calculation. Nevertheless, the result seems consistent with lifetime values from the literature. We have changed the text as follows:

**"When a constant atmospheric lifetime is assumed, the decreasing growth rate from 1983 through 2006 may imply that atmospheric $CH_4$ was approaching steady state, with no trend in emissions. The NOAA global mean $CH_4$ concentration was fitted with a function that describes the approach to a first-order steady state ($_{ss}$ index): $[CH_4](t) = [CH_4]_{ss}-([CH_4]_{ss}-[CH_4]_0)e^{-t/\tau}$; solving for the lifetime, $\tau$, gives 9.3 years, which is very close to current literature values (e.g., Prather et al., 2012, 9.1 ± 0.9 years). Such an approach includes uncertainties, especially due to the strong assumption of no trend in emissions and sinks, which does not agree with some study explaining the stabilization period by decreasing emissions associated to increasing sink (e.g. Bousquet et al., 2006). However, this value seems consistent albeit higher than the chemistry climate estimates (8.2 years, see Sect. 3.3.5)"**

*L1795ff: This is a good discussion of results, but I wonder how much these inversions depend on the a priori's for lifetime, etc. I do not see how you can cover this here, but can you comment? OK, see L1988.*
*L1988: Yes, if everyone uses the same trop OH, then the budget total is pretty well fixed (except for T biases, and maybe ITCZ gradients).*

Indeed, this is one of the main caveats of the top-down budget.

*L2154ff & L2200ff: Agreed. We continue to produce models with the same range in CH4 OH-lifetime and no means of differentiating them. I would not emphasize the vertical as our model failings include horizontal as well. We need to develop some key observational tests that reflect the reactivity of the air parcels. To start, we need to run the MIPS with some more focused diagnostics that tell us why the models CH4 lifetimes are so different. Even more interesting is that the historical trends and the scenario projections often go in different directions across the model ensemble. And we should not forget that our goal is accurately simulate CH4 loss, not just get OH correct.*

We thank Michael Prather for his comments and point of view of the future directions to better estimate methane loss. This comment has been used to feed Section 6.

*L2273: This is where you might want to look at the Holmes (2013 ACP) effort to model OH variability and match it to CH3CCL3 observations and CH4 lifetimes. It is multi-model and more physically based than these inversions.*

We thank the reviewer for the reference. This has been included and we will look at this study carefully for the further study on methane change.